# High-dimensional Asymptotics of Feature Learning: How One Gradient Step Improves the Representation

**Jimmy Ba**[1], **Murat A. Erdogdu**[1], **Taiji Suzuki**[2], **Zhichao Wang**[3], **Denny Wu**[1], **Greg Yang**[4]

[1]University of Toronto and Vector Institute,   [2]University of Tokyo and RIKEN AIP,
[3]University of California, San Diego,   [4]Microsoft Research AI

{jba,erdogdu,dennywu}@cs.toronto.edu, taiji@mist.i.u-tokyo.ac.jp,
zhw036@ucsd.edu, gregyang@microsoft.com

## Abstract

We study the first gradient descent step on the first-layer parameters $\boldsymbol{W}$ in a two-layer neural network: $f(\boldsymbol{x}) = \frac{1}{\sqrt{N}}\boldsymbol{a}^\top\sigma(\boldsymbol{W}^\top\boldsymbol{x})$, where $\boldsymbol{W} \in \mathbb{R}^{d \times N}, \boldsymbol{a} \in \mathbb{R}^N$ are randomly initialized, and the training objective is the empirical MSE loss: $\frac{1}{n}\sum_{i=1}^n (f(\boldsymbol{x}_i) - y_i)^2$. In the proportional asymptotic limit where $n, d, N \to \infty$ at the same rate, and an idealized student-teacher setting where the teacher $f^*$ is a single-index model, we compute the prediction risk of ridge regression on the conjugate kernel after one gradient step on $\boldsymbol{W}$ with learning rate $\eta$. We consider two scalings of the first step learning rate $\eta$. For small $\eta$, we establish a Gaussian equivalence property for the trained feature map, and prove that the learned kernel improves upon the initial random feature model, but cannot defeat the best linear model on the input. Whereas for sufficiently large $\eta$, we prove that for certain $f^*$, the same ridge estimator on trained features can go beyond this "linear regime" and outperform a wide range of (fixed) kernels. Our results demonstrate that even one gradient step can lead to a considerable advantage over random features, and highlight the role of learning rate scaling in the initial phase of training.

## 1 Introduction

We consider the training of a fully-connected two-layer neural network (NN) with $N$ neurons,

$$f_{\text{NN}}(\boldsymbol{x}) = \frac{1}{\sqrt{N}}\sum_{i=1}^N a_i\sigma(\langle\boldsymbol{x},\boldsymbol{w}_i\rangle) = \frac{1}{\sqrt{N}}\boldsymbol{a}^\top\sigma(\boldsymbol{W}^\top\boldsymbol{x}), \tag{1.1}$$

where $\boldsymbol{x} \in \mathbb{R}^d, \boldsymbol{W} \in \mathbb{R}^{d \times N}, \boldsymbol{a} \in \mathbb{R}^N$, $\sigma$ is the nonlinear activation function applied entry-wise, and the training objective is to minimize the empirical risk. Our analysis will be made in the *proportional asymptotic limit*, i.e., the number of training data $n$, the input dimensionality $d$, and the number of neurons $N$ jointly tend to infinity. Intuitively, this regime reflects the setting where the network width and data size are comparable, which is consistent with practical choices of model scaling.

When the first layer $\boldsymbol{W}$ is fixed and the second layer $\boldsymbol{a}$ is optimized, we arrive at a kernel model, where the kernel defined by features $\boldsymbol{x} \mapsto \sigma(\boldsymbol{W}^\top\boldsymbol{x})$ (often called the *hidden representation*) is referred to as the *conjugate kernel* (CK) [Nea95]. When $\boldsymbol{W}$ is randomly initialized, this model is an example of the *random features* (RF) model [RR08], the training and test performance of which has been extensively studied in the proportional limit [LLC18, MM22]. These precise characterizations reveal interesting phenomena also present in practical deep learning [BHMM19].

However, RF models do not fully explain the empirical success of NNs: one crucial advantage of deep learning is the *ability to learn useful features* [GDDM14, DCLT18] that "adapt" to the learning problem [Suz18]. In fact, recent works have shown that such adaptivity enables NNs optimized by

gradient descent to outperform a wide range of linear/kernel estimators [AZL19, GMMM19]. While many explanations of this separation have been proposed, our starting point is the empirical finding that "non-kernel" behavior often occurs in the *early phase* of NN optimization, especially under large learning rates [JSF$^+$20, FDP$^+$20]. The goal of this work is to answer the following question:

*Can we precisely capture the emergence of feature learning in the early phase of gradient descent, and demonstrate its improvement over the initial (fixed) kernel in the proportional limit?*

## 1.1 Contributions

Motivated by the above observations, we investigate a simplified scenario of the "early phase" of learning: how *the first gradient step* on the first-layer parameters $\boldsymbol{W}$ impacts the representation of the two-layer NN (1.1). Specifically, we consider regression with the squared loss (MSE), and a student-teacher setting in the proportional asymptotic limit; we aim to characterize the prediction risk of the kernel ridge regression estimator on top of the first-layer CK feature $\boldsymbol{x} \mapsto \sigma(\boldsymbol{W}^\top \boldsymbol{x})$, before and after one gradient descent step on the empirical risk (starting from Gaussian initialization).

Following prior works on the precise asymptotics of RF regression [GLK$^+$20, DL20], we focus on the setting where the input $\boldsymbol{x}$ is Gaussian and the teacher $f^*$ is a single-index model. In this case, the prediction risk of a large class of RF/kernel ridge regression estimators is lower-bounded by the $L^2$-norm of the "nonlinear" component of the teacher $\|\mathsf{P}_{>1}f^*\|_{L^2}^2$, i.e., they only learn *linear* functions on the input. After one gradient step on $\boldsymbol{W}$, we compute the CK ridge estimator using separate training data, and compare its prediction risk against this linear lower bound. Our analysis will be made under two choices of learning rate scalings:

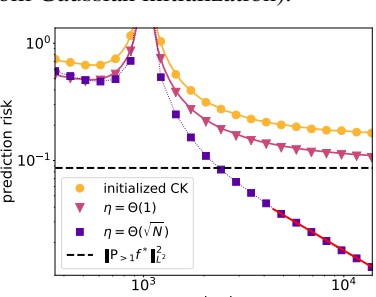

Figure 1: Prediction risk of ridge regression on trained CK features (erf) after one feature learning step. Markers represent empirical simulations and solid curves are predicted asymptotic values; red line indicates $\Theta(d/n)$ rate.

- **Small lr:** $\eta = \Theta(1)$. In Section 4, we extend the *Gaussian Equivalence Theorem* (GET) in [HL20] to the updated feature map after one gradient descent step on $\boldsymbol{W}$ with learning rate $\eta = \Theta(1)$; this allows us to precisely characterize the prediction risk using random matrix theoretical tools. We prove that after one gradient step, the ridge regression estimator on the learned CK features already exhibits nontrivial improvement over the initial RF ridge regression model (see pink curve in Figure 1), but it remains in the "linear regime" and cannot outperform the best linear estimator on the input (black dashed line).

- **Large lr:** $\eta = \Theta(\sqrt{N})$. In Section 5, we analyze a larger learning rate that coincides with the *maximal update parameterization* in [YH20]. For certain target functions $f^*$, we prove that kernel ridge regression after one feature learning step can achieve lower risk than the lower bound $\|\mathsf{P}_{>1}f^*\|_{L^2}^2$; thus, it outperforms a wide range of kernel estimators (see purple curve in Figure 1).

## 1.2 Related works

**Asymptotics of kernel regression.** Recent works provided precise analysis of RF and kernel models in the proportional limit [GLK$^+$20, DL20, LCM20, AP20, MM22]. These results typically build upon analyses of the spectrum of kernel matrices, a key ingredient in which is the "linearization" of nonlinear random matrices via Taylor expansion [EK10] or orthogonal polynomials [CS13, PW17].

Consequently, a large class of kernel models are essentially linear in the proportional asymptotic limit [LR20, BMR21]. In the case of RF models, a similar property is captured by the Gaussian Equivalence Theorem [GMKZ20, HL20, GLR$^+$21], which roughly states that RF estimators achieve the same prediction risk as a (noisy) linear model. For inputs with unit norm, [GMMM21, MMM21] showed that sample size $n = \Omega(d^2)$ is required to go beyond this "linear" regime. As we will see in certain settings, such a limitation can also be overcome (in the $n \asymp d$ scaling) by training the feature map for one gradient step with a sufficiently large learning rate.

**Advantage of NNs over fixed kernels.** It is well-known that under a specific initialization, the learning dynamics of overparameterized NNs can be described by the neural tangent kernel (NTK) [JGH18]. However, the NTK description essentially "freezes" the model around its initialization [COB19], and thus does not explain the presence of *feature learning* in NNs [YH20].

In fact, various works have shown that deep learning is more powerful than kernel methods in terms of approximation and estimation ability [Bac17, Suz18, IF19, SH20, GMMM20]. Moreover, in some specialized settings, NNs optimized with gradient-based methods can outperform the NTK (or more generally any kernel estimators) in terms of generalization error [AZL19, WLLM19, GMMM19, LMZ20, DM20, SA20, AZL20, RGKZ21, KWLS21, ABAB+21] (see [MKAS21, Table 2] for a survey). These results often require a careful analysis of the landscape (e.g., properties of global optimum) or optimization dynamics; in contrast, our goal is to precisely characterize the first gradient step and demonstrate a similar separation.

**Early phase of NN optimization.** Recent empirical studies suggest that properties of the final trained model is strongly influenced by the early stages of optimization [GAS19, LM20, PPVF21], and the NTK evolves most rapidly in the first few epochs [FDP+20]. Large learning rate in the initial steps can impact the conditioning of loss surface [JSF+20, CKL+21] and potentially improve the generalization performance [LWM19, LBD+20]. Under structural assumptions on the data, it has been proved that one gradient step with sufficiently large learning rate can drastically decrease the training loss [CLB21], extract task-relevant features [DM20, FCB22], or escape the trivial stationary point at initialization [HCG21]. While these works also highlight the benefit of one feature learning step, to our knowledge this advantage has not been *precisely* characterized in the proportional regime (where the performance of RF models has been extensively studied).

## 2  Problem setup and assumptions

**Notations.** Throughout this paper, $\|\cdot\|$ denotes the $\ell_2$-norm for vectors and the $\ell_2 \to \ell_2$ operator norm for matrices, and $\|\cdot\|_F$ is the Frobenius norm. For matrix $M \in \mathbb{R}^{n \times n}$, $\operatorname{tr}(M) = \frac{1}{n}\operatorname{Tr}(M)$ is the normalized trace. $\mathcal{O}_d(\cdot)$ and $o_d(\cdot)$ stand for the standard big-O and little-o notations, where the subscript highlights the asymptotic variable; we write $\tilde{\mathcal{O}}(\cdot)$ when the (poly-)logarithmic factors are ignored. $\mathcal{O}_{d,\mathbb{P}}(\cdot)$ (resp. $o_{d,\mathbb{P}}(\cdot)$) represents big-O (resp. little-o) in probability as $d \to \infty$. $\Omega(\cdot), \Theta(\cdot)$ are defined analogously. $\Gamma$ is the standard Gaussian distribution in $\mathbb{R}^d$. Given $f : \mathbb{R}^d \to \mathbb{R}$, we denote its $L^p$-norm w.r.t. $\Gamma$ as $\|f\|_{L^p(\mathbb{R}^d, \Gamma)}$, which we abbreviate as $\|f\|_{L^p}$ when the context is clear.

### 2.1  Training procedure

**Gradient descent on the 1st layer.** Given training examples $\{(x_i, y_i)\}_{i=1}^n$, we learn the two-layer NN (1.1) by minimizing the empirical risk: $\mathcal{L}(f) = \frac{1}{n}\sum_{i=1}^n \ell(f(x_i), y_i)$, where $\ell$ is the squared loss $\ell(x, y) = \frac{1}{2}(x - y)^2$. As previously remarked, fixing the first layer $W$ at random initialization and learning the second layer $a$ yields an RF model, which is a convex problem with closed-form solution. In contrast, we are interested in *learning the feature map (representation)*; hence we first fix $a$ (at initialization) and perform gradient descent on $W$. We write the initialized first-layer as $W_0$, and the weights after one gradient step as $W_1$. The gradient update, which we refer to as the *feature learning step*, with learning rate $\eta$ is given as: $W_1 = W_0 + \eta\sqrt{N} \cdot G_0$ where

$$G_0 := \frac{1}{n}X^\top \left[ \left( \frac{1}{\sqrt{N}}\left( y - \frac{1}{\sqrt{N}}\sigma(XW_0)a \right)a^\top \right) \odot \sigma'(XW_0) \right], \qquad (2.1)$$

in which $\odot$ is the Hadamard product, $\sigma'$ is the derivative of $\sigma$ (acting entry-wise), and we denoted the input feature matrix $X \in \mathbb{R}^{n \times d}$, and the corresponding label vector $y \in \mathbb{R}^n$. We remark that the $\sqrt{N}$-scaling in front of $\eta$ accounts for the $\frac{1}{\sqrt{N}}$-prefactor in our definition of two-layer NN (1.1).

**Ridge regression for the 2nd layer.** After obtaining the updated weights $W_1$, we evaluate the quality of the new CK features by computing the prediction risk of the *kernel ridge regression* estimator on top of the first-layer representation. Note that if ridge regression is performed on the same data $X$, then after one feature learning step, $W_1$ is no longer independent of $X$, which significantly complicates the analysis. To circumvent this difficulty, we estimate the regression coefficients $\hat{a}$ using *a new set of training data* $\{\tilde{x}_i, \tilde{y}_i\}_{i=1}^n$, which for simplicity we assume to have the same size as the original dataset. This can be interpreted as the representation being "pretrained" on separate data before the ridge regression estimator is learned.

Denoting the feature matrix on the fresh training set $\{\tilde{X}, \tilde{y}\}$ as $\Phi := \frac{1}{\sqrt{N}}\sigma(\tilde{X}W_1) \in \mathbb{R}^{n \times N}$, the CK ridge regression estimator can be obtained by solving $\hat{a} = \operatorname{argmin}_a \left\{ \frac{1}{n}\|\tilde{y} - \Phi a\|^2 + \frac{\lambda}{N}\|a\|^2 \right\}$.

## 2.2 Student-teacher setting and main assumptions

Given a target function (teacher model) $f^*$ and a learned model $\hat{f}$, we evaluate the model performance using the prediction risk: $\mathcal{R}(\hat{f}) = \mathbb{E}_{\boldsymbol{x}}(\hat{f}(\boldsymbol{x}) - f^*(\boldsymbol{x}))^2 = \|\hat{f} - f^*\|_{L^2}^2$, where the expectation is taken over the test data from the same training distribution.

We utilize the orthogonal decomposition of the activation function $\sigma$. Define the coefficients

$$\mu_0 = \mathbb{E}[\sigma(z)], \quad \mu_1 = \mathbb{E}[z\sigma(z)], \quad \mu_2 = \sqrt{\mathbb{E}[\sigma(z)^2] - \mu_0^2 - \mu_1^2}, \quad \text{where } z \sim \mathcal{N}(0,1). \quad (2.2)$$

This implies $\sigma(z) = \mu_0 + \mu_1 z + \sigma_\perp(z)$, where $\mathbb{E}[\sigma_\perp(z)] = \mathbb{E}[z\sigma_\perp(z)] = 0$, and $\mathbb{E}[\sigma_\perp(z)^2] = \mu_2^2$.

Similarly, for square integrable target function $f^*$, we have the orthogonal decomposition

$$f^*(\boldsymbol{x}) = \mu_0^* + \mu_1^* \langle \boldsymbol{x}, \boldsymbol{\beta}_* \rangle + \mathsf{P}_{>1} f^*(\boldsymbol{x}), \quad \mu_1^* \boldsymbol{\beta}_* = \mathbb{E}[\boldsymbol{x} f^*(\boldsymbol{x})], \quad (2.3)$$

where $\mathsf{P}_{>1}$ is the projector orthogonal to constant and linear functions in $L^2(\mathbb{R}^d, \Gamma)$, which implies that $\mathbb{E}[\mathsf{P}_{>1}f^*(\boldsymbol{x})] = 0, \mathbb{E}[\boldsymbol{x}\mathsf{P}_{>1}f^*(\boldsymbol{x})] = \boldsymbol{0}$. As $d \to \infty$, quantities defined in (2.3) satisfy $\|\boldsymbol{\beta}_*\| = 1$, $\|\mathsf{P}_{>1}f^*\|_{L^2} \to \mu_2^*$, where $\mu_0^*, \mu_1^*, \mu_2^*$ are bounded constants. Intuitively, $\mu_0^*, \mu_1^*$, and $\mu_2^*$ can be interpreted as the "magnitude" of the constant, linear, and nonlinear components of $f^*$, respectively.

**Assumption 1.**

1. **Proportional limit.** $n, d, N \to \infty$, $n/d \to \psi_1$, $N/d \to \psi_2$, where $\psi_1, \psi_2 \in (0, \infty)$.

2. **Gaussian initialization.** $\sqrt{d} \cdot [\boldsymbol{W}_0]_{ij} \overset{\text{i.i.d.}}{\sim} \mathcal{N}(0,1)$, $\sqrt{N} \cdot [\boldsymbol{a}]_j \overset{\text{i.i.d.}}{\sim} \mathcal{N}(0,1)$, for $i \in [d], j \in [N]$.

3. **Normalized activation.** The activation function $\sigma$ has $\lambda_\sigma$-bounded first three derivatives almost surely. In addition, $\sigma$ satisfies $\mu_0 = 0$ and $\mu_1, \mu_2 \neq 0$ defined in (2.2).

4. **Single-index teacher.** Labels are generated as $y_i = f^*(\boldsymbol{x}_i) + \varepsilon_i$, where $\boldsymbol{x}_i \overset{\text{i.i.d.}}{\sim} \mathcal{N}(0, \boldsymbol{I})$, and $\varepsilon_i$ is i.i.d. sub-Gaussian noise with mean $0$ and variance $\sigma_\varepsilon^2$. The teacher $f^*(\boldsymbol{x}) = \sigma^*(\langle \boldsymbol{x}, \boldsymbol{\beta}_* \rangle)$, where $\boldsymbol{\beta}_* \in \mathbb{R}^d$ with $\|\boldsymbol{\beta}_*\| = 1$, and $\sigma^*$ is Lipschitz with $\mu_0^* = 0, \mu_1^* \neq 0$ as defined in (2.3).

**Remark.** *We make the following comments on the above assumptions.*

- *Following [HL20], we assume smooth centered activation to simplify the computation; empirical evidence suggests that similar result holds beyond this condition (e.g. [LGC$^+$21]). We also expect the Gaussian input assumption may be replaced by weaker orthogonality conditions as in [FW20].*

- *The single-index setting has been extensively studied in the proportional regime [GLK$^+$20, DL20, HL20]. However, prior works only considered training the coefficients $\boldsymbol{a}$ on top of fixed feature map, and such RF models cannot efficiently learn a single-index $f^*$ in high dimensions [YS19].*

Under Assumption 1, a relatively large sample size corresponds to larger $\psi_1$, and a relatively large network width corresponds to larger $\psi_2$. The proportional scaling of $n, d, N$ implies that the model width is not significantly larger than the training set size, in contrast to the polynomial overparameterization often required in NTK analyses [DZPS19], which may be less realistic in practical settings.

Importantly, the initialization of our two-layer NN (1.1) resembles the *mean-field* parameterization [MMN18, CB18]: the second layer is divided by an additional $\sqrt{N}$-factor compared to the kernel (NTK) scaling — this ensures that $f_{\text{NN}}(\boldsymbol{x}) = o_{d,\mathbb{P}}(1)$ at initialization and enables feature learning (see [YH20, Corollary 3.10]). As an illustrative example in Figure 2, we plot the gradient descent trajectory of the first-layer parameters $\boldsymbol{W}$ in two coordinates. Observe that under the mean-field parameterization (main figure), the neurons travel away from the initialization and align with the target function (black dashed lines), whereas in the NTK parameterization (subfigure, which omits the $\frac{1}{\sqrt{N}}$-prefactor), the parameters remain close to their initialization and hence do not learn useful features.

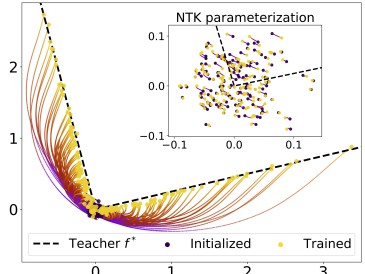

Figure 2: 2D visualization of optimization trajectory under mean-field (main) and NTK (subfigure) parameterizations. $f^*$ consists of two ReLU neurons and the student is a two-layer ReLU neural network. Darker color indicates earlier in training, and vice versa. We set $d = 512$, $\psi_1 = \psi_2 = 10$; both models are optimized until training losses are below $10^{-3}$.

# 3 Preliminary results

## 3.1 Lower bound for kernel ridge regression

To illustrate the benefits of *feature learning*, we compare the prediction risk of ridge regression on the trained CK (after one gradient step) against that on the initial RF and fixed kernels. Specifically, given training data $\{\boldsymbol{x}_i, y_i\}_{i=1}^n$, we consider the following classes of kernel models for comparison.

- **Random features model.** We introduce two RF kernels associated with (1.1) at initialization: the conjugate kernel (CK) defined by features $\phi_{\mathrm{CK}}(\boldsymbol{x}) = \frac{1}{\sqrt{N}}\sigma(\boldsymbol{W}_0^\top \boldsymbol{x}) \in \mathbb{R}^N$, and the neural tangent kernel (NTK) [JGH18] defined by features $\phi_{\mathrm{NTK}}(\boldsymbol{x}) = \frac{1}{\sqrt{Nd}}\mathrm{Vec}\big(\sigma'(\boldsymbol{W}_0^\top \boldsymbol{x})\boldsymbol{x}^\top\big) \in \mathbb{R}^{Nd}$. Given a feature map $\mathrm{RF} \in \{\mathrm{CK,NTK}\}$, the RF ridge regression estimator can be written as

$$\hat{f}_{\mathrm{RF}}(\boldsymbol{x}) = \langle \phi_{\mathrm{RF}}(\boldsymbol{x}), \hat{\boldsymbol{a}}\rangle, \quad \hat{\boldsymbol{a}} = \operatorname*{argmin}_{\boldsymbol{a}\in\mathbb{R}^N}\left\{\frac{1}{n}\sum_{i=1}^n(y_i - \langle\phi_{\mathrm{RF}}(\boldsymbol{x}_i),\boldsymbol{a}\rangle)^2 + \frac{\lambda}{N}\|\boldsymbol{a}\|^2\right\}. \quad (3.1)$$

- **Rotation invariant kernel model.** Consider the inner-product kernel: $k(\boldsymbol{x},\boldsymbol{y}) = g\left(\frac{\langle\boldsymbol{x},\boldsymbol{y}\rangle}{d}\right)$, and the Euclidean distance kernel: $k(\boldsymbol{x},\boldsymbol{y}) = g\left(\frac{\|\boldsymbol{x}-\boldsymbol{y}\|^2}{d}\right)$, where $g$ satisfies the smoothness conditions in [EK10]. Denoting the associated RKHS with $\mathcal{H}$, the kernel ridge estimator is given by

$$\hat{f}_{\mathrm{ker}} = \operatorname*{argmin}_{f\in\mathcal{H}}\left\{\frac{1}{n}\sum_{i=1}^n(y_i - f(\boldsymbol{x}_i))^2 + \lambda\|f\|_{\mathcal{H}}^2\right\} \Rightarrow \hat{f}_{\mathrm{ker}}(\boldsymbol{x}) = k(\boldsymbol{x},\boldsymbol{X})^\top(\boldsymbol{K}+\lambda\boldsymbol{I})^{-1}\boldsymbol{y}. \quad (3.2)$$

We write the prediction risk of the above kernel estimators as $\mathcal{R}_{\mathrm{CK}}(\lambda), \mathcal{R}_{\mathrm{NTK}}(\lambda), \mathcal{R}_{\mathrm{ker}}(\lambda)$, respectively. The following lower bound on the prediction risk is a simple combination of existing results.

**Proposition 1** ([HL20, MZ20, BMR21]). *Under Assumption 1, we have*

$$\inf_{\lambda>0}\min\{\mathcal{R}_{\mathrm{CK}}(\lambda),\mathcal{R}_{\mathrm{NTK}}(\lambda),\mathcal{R}_{\mathrm{ker}}(\lambda)\} \geq \|\mathsf{P}_{>1}f^*\|_{L^2}^2 + o_{d,\mathbb{P}}(1), \quad (3.3)$$

*where $\mathsf{P}_{>1}$ denotes the projector orthogonal to constant and linear functions in $L^2(\mathbb{R}^d,\Gamma)$.*

This proposition implies that in the proportional limit, ridge regression on the RF or rotationally invariant kernels defined above does not outperform the best linear estimator on the input – it cannot achieve vanishing risk unless the target function is linear ($\|\mathsf{P}_{>1}f^*\|_{L^2} = 0$). In the following, we compare the prediction risk of the ridge estimator on trained features against this lower bound.

## 3.2 Almost rank-1 property of the gradient matrix

Before we analyze the prediction risk of the ridge regression estimator on the trained CK, we first need to understand the gradient matrix $\boldsymbol{G}_0$ in (2.1). The following proposition shows that the first gradient step on $\boldsymbol{W}$ can be approximated in operator norm by a rank-1 matrix under Assumption 1.

**Proposition 2.** *Define $\boldsymbol{G}_0 := \frac{1}{\eta\sqrt{N}}(\boldsymbol{W}_1 - \boldsymbol{W}_0)$ and a rank-1 matrix $\boldsymbol{A} := \frac{\mu_1}{n\sqrt{N}}\boldsymbol{X}^\top\boldsymbol{y}\boldsymbol{a}^\top$. Given Assumption 1, there exist some constants $c, C > 0$ such that for all large $n, N$, and $d$, we have*

$$\|\boldsymbol{G}_0 - \boldsymbol{A}\| \leq \frac{C\log^2 n}{\sqrt{n}}\cdot\|\boldsymbol{G}_0\|,$$

*with probability at least $1 - ne^{-c\log^2 n}$.*

**Scaling of learning rate $\eta$.** Based on the above proposition, we can now specify an appropriate learning rate $\eta$ such that the change in the first-layer weights after one gradient descent step is neither insignificant nor unreasonably large. Assumption 1 implies that, for proportional $n, d, N$, the initial weight matrix satisfies $\|\boldsymbol{W}_0\| = \Theta_{d,\mathbb{P}}(1)$, $\|\boldsymbol{W}_0\|_F = \Theta_{d,\mathbb{P}}(\sqrt{d})$, and due to Proposition 2, the first gradient step satisfies $\sqrt{N}\|\boldsymbol{G}_0\| = \Theta_{d,\mathbb{P}}(1)$, $\sqrt{N}\|\boldsymbol{G}_0\|_F = \Theta_{d,\mathbb{P}}(1)$.

In light of the above scaling, if we write $\eta = \Theta(N^\alpha)$, then $\alpha \geq 0$ is required so that the change in the weight matrix is non-negligible (one may verify that for $\eta = o_d(1)$, the test performance of kernel

ridge regression remains unchanged after one GD step). On the other hand, when $\alpha > 1/2$, the gradient update "overwhelms" the initialized parameters $\boldsymbol{W}_0$, and the preactivation feature $\langle \boldsymbol{x}, \boldsymbol{w}_i \rangle$ in the NN (1.1) becomes unbounded as $N \to \infty$. This motivates us to consider the following two regimes of learning rate scaling.

$$\textbf{Small lr}: \eta = \Theta(1) \;\Rightarrow\; \|\boldsymbol{W}_1 - \boldsymbol{W}_0\| \asymp \|\boldsymbol{W}_0\|$$

$$\textbf{Large lr}: \eta = \Theta(\sqrt{N}) \;\Rightarrow\; \|\boldsymbol{W}_1 - \boldsymbol{W}_0\|_F \asymp \|\boldsymbol{W}_0\|_F$$

In Section 4, we consider small step size $\eta = \Theta(1)$, which is parallel to common practice in NN optimization[1]. Whereas in Section 5, we analyze the larger step size $\eta = \Theta(\sqrt{N})$, which resembles the learning rate scaling in the maximal update parameterization in [YH20]; in particular, from Lemma 10 in Appendix B.1, one can easily verify that given data point $\boldsymbol{x} \sim \mathcal{N}(0, \boldsymbol{I})$, the change in each coordinate of the feature vector is roughly of the same order as its initialized magnitude, that is, for $i \in [N]$, $\left| \sigma(\boldsymbol{W}_1^\top \boldsymbol{x}) - \sigma(\boldsymbol{W}_0^\top \boldsymbol{x}) \right|_i \asymp \left| \sigma(\boldsymbol{W}_0^\top \boldsymbol{x}) \right|_i = \tilde{\Theta}(1)$ with probability 1 as $N \to \infty$.

# 4 $\eta = \Theta(1)$: improvement over the initial CK

From Proposition 2, we observe that the dominant rank-1 direction in the first-step gradient matrix $\boldsymbol{G}_0$ contains information of the teacher model $f^*$ (through label vector $\boldsymbol{y}$). Intuitively, this indicates that the learned feature map after one GD step $\boldsymbol{x} \mapsto \sigma(\boldsymbol{W}_1^\top \boldsymbol{x})$ can "adapt" to $f^*$, and hence we may expect the ridge regression estimator on the trained CK to achieve better performance. In this section, we precisely characterize the CK prediction risk under the small learning rate $\eta = \Theta(1)$. We first introduce the Gaussian equivalence property which will be useful in the risk computation.

## 4.1 The Gaussian equivalence property

The Gaussian Equivalence Theorem (GET) states that the performance of a nonlinear kernel model is the same as that of a noisy linear model. Specifically, for the ridge regression estimator, define

$$\mathcal{R}_{\mathrm{F}}(\lambda) = \mathbb{E}_{\boldsymbol{x}}\big( \langle \boldsymbol{\phi}_{\mathrm{F}}(\boldsymbol{x}), \hat{\boldsymbol{a}}_\lambda \rangle - f^*(\boldsymbol{x}) \big)^2,$$

$$\hat{\boldsymbol{a}}_\lambda = \mathrm{argmin}_{\boldsymbol{a}} \Big\{ \frac{1}{n} \sum_{i=1}^{n} (y_i - \langle \boldsymbol{\phi}_{\mathrm{F}}(\boldsymbol{x}_i), \boldsymbol{a} \rangle)^2 + \frac{\lambda}{N} \|\boldsymbol{a}\|^2 \Big\}, \tag{4.1}$$

where $\mathrm{F} \in \{\mathrm{CK}, \mathrm{GE}\}$ indicates the choice of feature map, which can be either the nonlinear CK feature $\boldsymbol{\phi}_{\mathrm{CK}}(\boldsymbol{x}) = \frac{1}{\sqrt{N}} \sigma(\boldsymbol{W}^\top \boldsymbol{x})$, or the linear Gaussian equivalent (GE) feature $\boldsymbol{\phi}_{\mathrm{GE}}(\boldsymbol{x}) = \frac{1}{\sqrt{N}} \big( \mu_1 \boldsymbol{W}^\top \boldsymbol{x} + \mu_2 \boldsymbol{z} \big)$ where $\boldsymbol{z} \sim \mathcal{N}(0, \boldsymbol{I})$ is independent of $\boldsymbol{x}, \boldsymbol{W}$. In the following, for both $\boldsymbol{\phi}_{\mathrm{CK}}$ and $\boldsymbol{\phi}_{\mathrm{GE}}$, we take $\boldsymbol{W}$ to be the updated weight matrix $\boldsymbol{W}_1$ after one GD step.

The Gaussian equivalence refers to the universality phenomenon $\mathcal{R}_{\mathrm{CK}}(\lambda) \approx \mathcal{R}_{\mathrm{GE}}(\lambda)$. For RF models (3.1), the GET has been rigorously proved in [HL20, MS22, MM22]. Furthermore, [GLR+21, LGC+21] provided empirical evidence that such equivalence holds for more general feature maps, including the representation of certain pretrained NNs (e.g., see [LGC+21, Figure 4]). Since our setting goes beyond RF models and cannot be covered by the prior results, we establish the GET for our *trained* feature map under small learning rate.

**Theorem 3.** *Suppose that Assumption 1 holds and the activation $\sigma$ is an odd function. If the learning of $\boldsymbol{W}_1$ in (2.1) and estimation of $\hat{\boldsymbol{a}}_\lambda$ in (4.1) are performed on independent training data $\boldsymbol{X}$ and $\tilde{\boldsymbol{X}}$, respectively, then the GET holds after the first-layer weight is trained for one gradient step with learning rate $\eta = \Theta(1)$; that is, for the CK feature $\boldsymbol{\phi}_{\mathrm{CK}}(\boldsymbol{x}) = \frac{1}{\sqrt{N}} \sigma(\boldsymbol{W}_1^\top \boldsymbol{x})$, and $\lambda > 0$,*

$$|\mathcal{R}_{\mathrm{CK}}(\lambda) - \mathcal{R}_{\mathrm{GE}}(\lambda)| = o_{d,\mathbb{P}}(1). \tag{4.2}$$

This is to say, for learning rate $\eta = \Theta(1)$, the Gaussian equivalent model provides an accurate description of the prediction risk of CK ridge regression after one feature learning step. The important observation is that even though the trained parameters in $\boldsymbol{W}_1$ are no longer i.i.d., the Gaussian equivalence property can still hold when $\boldsymbol{W}_1 - \boldsymbol{W}_0$ remains "small" (in some norm, see (C.3) in Appendix C.1 for details), which entails that the neurons remain nearly orthogonal to one another.

---

[1]Heuristically, the updated NN under $\eta = \Theta(1)$ remains close to the "kernel regime" in the sense that each neuron is close to initialization, i.e., as $N \to \infty$, $\left| [\boldsymbol{W}_1 - \boldsymbol{W}_0]_{ij} \right| \ll \left| [\boldsymbol{W}_0]_{ij} \right|$ with high probability.

**Implications of Gaussian equivalence.**   Under the GET, we can alternatively compute $\mathcal{R}_{\mathrm{GE}}(\lambda)$, the prediction risk of ridge regression on noisy Gaussian features $\phi_{\mathrm{GE}}$, which is much easier to analyze. Theorem 3 is empirically validated in Figure 3(a)(b), where we observe an agreement between the experimental values and the analytic predictions[2] from Section 4.2. On the other hand, the GET also implies that the kernel estimator is essentially "linear" in high dimensions. For the squared loss, it is straightforward to verify that the Gaussian equivalent model cannot learn the nonlinear component of the target function $\mathsf{P}_{>1} f^*$ as follows.

**Fact 4.** *Under the same assumptions as Theorem 3, $\mathcal{R}_{\mathrm{GE}}(\lambda) \geq \|\mathsf{P}_{>1} f^*\|_{L^2}^2$ for any $\psi_1, \psi_2, \lambda > 0$.*

Hence when $\eta = \Theta(1)$, even though training the first-layer $\boldsymbol{W}$ for one step can lead to non-trivial improvement over the initial RF model (which we precisely quantify in Section 4.2), the learned CK cannot outperform the best linear model on the input features. In other words, to (possibly) learn a nonlinear $f^*$, the trained feature map needs to violate the GET. In the case of one gradient step on $\boldsymbol{W}$, this amounts to using a sufficiently large step size, which we analyze in Section 5.

## 4.2   Precise asymptotics of CK ridge regression

Having established the Gaussian equivalence property for the CK ridge estimator after one gradient step with $\eta = \Theta(1)$, we can now compute the asymptotic prediction risk for the trained kernel and compare with the initialized RF. To quantify the discrepancy in the prediction risk (4.1), we write $\mathcal{R}_0(\lambda)$ as the prediction risk of the initialized RF ridge regression estimator (on the feature map $\boldsymbol{x} \mapsto \sigma(\boldsymbol{W}_0^\top \boldsymbol{x})$), and $\mathcal{R}_1(\lambda)$ as the prediction risk of the ridge estimator on the trained feature map after one feature learning step $\boldsymbol{x} \mapsto \sigma(\boldsymbol{W}_1^\top \boldsymbol{x})$.

Importantly, because of the dependency between the trained weights $\boldsymbol{W}_1$ and the teacher model $f^*$ (due to the gradient update (2.1)), we cannot simply apply a rotation invariance argument (e.g., [MM22, Lemma 9.2]) to remove the dependency on the true parameters $\boldsymbol{\beta}_*$ and reduce the prediction risk to the trace of certain rational functions of the kernel matrix. In other words, knowing the spectrum (or the Stieltjes transform) of the CK is not sufficient for these purposes. Instead, we utilize the GET and the almost rank-1 property of $\boldsymbol{G}_0$ in Proposition 2, which, in combination with techniques from operator-valued free probability theory [MS17, AP20], enables us to obtain the asymptotic expression of the difference in the prediction risk before and after one gradient step.

**Theorem 5.** *Under the same assumptions as Theorem 3 and $\eta = \Theta(1)$, we have*

$$\mathcal{R}_0(\lambda) - \mathcal{R}_1(\lambda) \xrightarrow{\mathbb{P}} \delta(\eta, \lambda, \psi_1, \psi_2) \geq 0,$$

*where $\delta(\eta, \lambda, \psi_1, \psi_2)$ is defined by (C.19) in Appendix C.3. Here, $\delta$ is a non-negative function of $\eta, \lambda, \psi_1, \psi_2 \in (0, +\infty)$ with parameters $\mu_1^*, \mu_1, \mu_2$, and it vanishes if and only if (at least) one of $\mu_1^*, \mu_1$ and $\eta$ is equal to zero.*

**Remark.** *Performance of the initial RF ridge estimator $\mathcal{R}_0(\lambda)$ has been characterized by the prior works [GLK+20, MM22]; hence, the precise asymptotics of $\delta$ provided in Theorem 5 allows us to explicitly compute the asymptotic prediction risk of the CK model after one gradient step, i.e. $\mathcal{R}_1(\lambda)$.*

Theorem 5 confirms our intuition that training the first-layer parameters improves the CK model, as shown in Figure 3(a)(b). Remarkably, this improvement (when $\delta > 0$) holds for any $\psi_1, \psi_2 \in (0, \infty)$, that is, taking one gradient step (with learning rate $\eta = \Theta(1)$) is *always* beneficial, even when the training set size $n$ is small. Moreover, we do not require the student and teacher models to have the same nonlinearity — a non-vanishing decrease in the prediction risk is present as long as $\mu_1, \mu_1^* \neq 0$. On the other hand, the GET also implies an upper bound on the possible improvement: $\delta \leq \mathcal{R}_0(\lambda) - \mu_2^{*2}$ as $n, d, N \to \infty$; this is to say, the trained CK remains in the "linear" regime.

Fore the details of Theorem 5, see Appendix C.3.2. Additionally, from inspecting the asymptotic risk formulae (C.19), we can arrive at the following characterization of two special cases of interest.

- **Large sample regime ($\psi_1 \to \infty$):** $\delta$ is increasing with respect to the learning rate $\eta$; that is, taking a larger step results in greater decrease in the prediction risk, as shown in Figure 3(a).

- **Large width regime ($\psi_2 \to \infty$):** In this case $\delta \to 0$; thus, the benefit of one-step feature learning (with $\eta = \Theta(1)$) becomes less significant as the width increases, as shown in Figure 3(b).

---

[2]We note that when the first-layer weights are trained for more gradient steps, the GET (Theorem 3) will likely fail eventually, as empirically demonstrated in Appendix A.1.

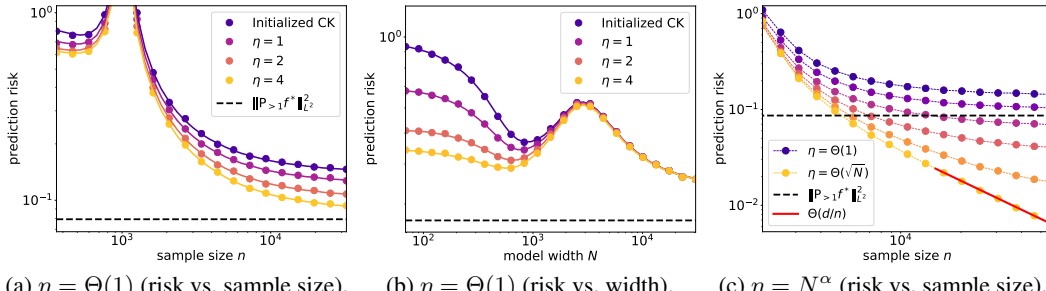

(a) $\eta = \Theta(1)$ (risk vs. sample size).  (b) $\eta = \Theta(1)$ (risk vs. width).  (c) $\eta = N^\alpha$ (risk vs. sample size).

Figure 3: Prediction risk of CK ridge regression on trained features: dots represent empirical simulations ($d = 512$, averaged over 50 runs) and solid curves are asymptotic predictions; dashed black line corresponds to the kernel lower bound (3.3). (a) $\eta = \Theta(1)$, $\sigma = \tanh$, $\sigma^* = \text{SoftPlus}$; we set $\psi_2 = 2$, $\lambda = 10^{-4}$, $\sigma_\varepsilon = 0.25$. (b) $\eta = \Theta(1)$, $\sigma = \tanh$, $\sigma^* = \text{ReLU}$; we set $\psi_1 = 5$, $\lambda = 10^{-2}$, $\sigma_\varepsilon = 0.1$. (c) $\eta = N^\alpha$ for $\alpha \in [0, 1/2]$; brighter color represents larger step size. We choose $\sigma = \sigma^* = \text{erf}$, $\psi_2 = 2$, $\lambda = 10^{-3}$, and $\sigma_\varepsilon = 0.1$.

## 5 $\eta = \Theta(\sqrt{N})$: improvement over the kernel lower bound

In this section, we consider a gradient step with large learning rate $\eta = \Theta(\sqrt{N})$, which matches the asymptotic order of the Frobenius norm of the gradient $\boldsymbol{G}_0$ and that of the initialized weight matrix $\boldsymbol{W}_0$. Note that after absorbing the prefactors, this learning rate scaling is analogous to the maximal update parameterization [YH20], which admits a feature learning limit. More specifically, the change in each coordinate of the feature vector $[\sigma(\boldsymbol{W}^\top \boldsymbol{x})]_i$ is $\tilde{\Theta}_{d,\mathbb{P}}(1)$, which has roughly the same order of magnitude as its value at initialization.

Due to the large step size, columns of the updated weight matrix $\boldsymbol{W}_1$ are no longer near-orthogonal, which is an important property in existing analyses of the Gaussian equivalence (e.g., see Proposition 13 in Appendix C.1 or [HL20, Equation (66)]). Indeed, we will see that in this regime, the ridge regression estimator on the trained CK features is no longer "linear" and can potentially outperform the kernel lower bound (3.3) in the proportional limit. However, in the absence of GET, it is difficult to derive the precise asymptotics of the CK model. As an alternative, we establish an *upper bound* on the prediction risk $\mathcal{R}_1(\lambda)$, which we then compare against the kernel ridge lower bound.

**Existence of a "good" solution.** Given the trained first-layer weights $\boldsymbol{W}_1$, we first construct a second-layer $\tilde{\boldsymbol{a}}$ for which the prediction risk can be upper-bounded. For a pair of nonlinearities $(\sigma, \sigma^*)$, we introduce a scalar $\tau^*$ which is the optimum of the following minimization problem:

$$\tau^* := \inf_{\kappa \in \mathbb{R}} \mathbb{E}_{\xi_1} \left[ \left( \sigma^*(\xi_1) - \mathbb{E}_{\xi_2} \sigma(\kappa \xi_1 + \xi_2) \right)^2 \right], \tag{5.1}$$

where $\xi_1, \xi_2 \overset{\text{i.i.d.}}{\sim} \mathcal{N}(0, 1)$. We write $\kappa^*$ as an optimal value at which $\tau^*$ is attained (when $\tau^*$ is not achieved by a finite $\kappa$, the same argument holds by introducing a small tolerance factor $\epsilon > 0$ in $\tau^*$; see Appendix D.2). Roughly speaking, $\tau^*$ approximates the prediction risk of a specific student model which takes the form of an average over a *subset* of neurons (after one feature learning step). In particular, the first term on the RHS of (5.1) containing $\sigma^*$ corresponds to the teacher $f^*$, and the second term $\mathbb{E}_{\xi_2}$ represents the constructed student model. The following lemma shows that we can find some $\tilde{\boldsymbol{a}}$ on the trained CK features whose prediction risk is approximately $\tau^*$, under the additional assumption that the activation function $\sigma$ is bounded. For more details, see Appendix D.

**Lemma 6** (Informal). *Suppose that Assumption 1 holds and $\sigma$ is bounded. Then, after one gradient step on $\boldsymbol{W}$ with $\eta = \Theta(\sqrt{N})$, there exist some second-layer coefficients $\tilde{\boldsymbol{a}}$ such that the constructed student model $\tilde{f}(\boldsymbol{x}) = \frac{1}{\sqrt{N}} \tilde{\boldsymbol{a}}^\top \sigma(\boldsymbol{W}_1^\top \boldsymbol{x})$ achieves a prediction risk which is "close" to $\tau^*$.*

It is worth noting that the definition of $\tau^*$ does not involve the specific value of the learning rate $\eta$. This is because for any choice of $\eta = \Theta(\sqrt{N})$, due to the Gaussian initialization of $a_i$, we can find a subset of weights that receive a "good" learning rate (with high probability) such that the corresponding neurons are useful for learning the teacher model. In addition, observe that $\tau^*$ is a simple Gaussian integral which can be numerically or analytically computed (see Appendix D.2 for more examples). For instance, when $\sigma = \sigma^* = \text{erf}$, one can easily verify that $\kappa^* = \sqrt{3}$ and $\tau^* = 0$.

**Prediction risk of ridge regression.** Since we have established the existence of a "good" student model $\tilde{f}$ that can achieve a prediction risk close to $\tau^*$ (as defined in (5.1)), in what follows, we prove an upper bound for the prediction risk of the ridge regression estimator on the trained CK features $\mathcal{R}_1(\lambda)$ in terms of the scalar $\tau^*$. The proof of the following result is shown in Appendix D.3.

**Theorem 7.** *Under the same assumptions as Lemma 6, after one gradient step on $W$ with $\eta = \Theta(\sqrt{N})$, there exist constants $C, \psi_1^* > 0$ such that for any $n/d > \psi_1^*$, the ridge regression estimator (4.1) with regularization parameter $n^{\varepsilon-1} < N^{-1}\lambda < n^{-\varepsilon}$ for some small $\varepsilon > 0$ satisfies*

$$\mathcal{R}_1(\lambda) \leq 10\tau^* + C\left(\sqrt{\tau^*} \cdot \sqrt{\frac{d}{n}} + \frac{d}{n}\right),$$

*with probability 1 as $n, d, N \to \infty$ proportionally.*

While Theorem 7 does not provide exact expression of the prediction risk, the upper bound still allows us to compare the prediction risk of the CK ridge regression before and after one large gradient step. In particular, if $\|\mathsf{P}_{>1} f^*\|_{L^2}^2 \geq 10\tau^*$ (the constant 10 is not optimized), we know that the trained CK can outperform the kernel lower bound (3.3) (and also the initialized CK) in the proportional limit, when the ratio $\psi_1 = n/d$ is sufficiently large. The following corollary provides two examples of this separation (see Figure 3(c)).

**Corollary 8.** *Under the same conditions as Theorem 7, there exists a constant $\psi_1^*$ such that for any $\psi_1 > \psi_1^*$, the following holds with probability 1 when $n, d, N \to \infty$ proportionally:*
- *For $\sigma = \sigma^* = \mathrm{erf}$, we have $\mathcal{R}_1(\lambda) = \mathcal{O}(d/n)$.* • *For $\sigma = \sigma^* = \tanh$, we have $\mathcal{R}_1(\lambda) < \|\mathsf{P}_{>1} f^*\|_{L^2}^2$.*

In the two examples outlined above, training the features by taking one large gradient step on the first-layer parameters can lead to substantial improvement in the performance of the CK model. In fact, the new ridge regression estimator may outperform a wide range of kernel models as described in Section 3.1, and as shown in Figure 3(c). However, we emphasize that this separation is only present in specific pairs of $(\sigma, \sigma^*)$ for which the scalar $\tau^*$ is sufficiently small. In general settings, learning a good representation would likely require a training procedure that takes more than one gradient step (even if $f^*$ is as simple as a single-index model, see Figure 4(c) in Appendix A.1).

## 6 Conclusion

We investigated how the conjugate kernel of a two-layer neural network (1.1) benefits from feature learning in an idealized student-teacher setting, where the first-layer parameters $W$ are updated by one gradient descent step on the empirical risk. Based on the approximate low-rank property of the gradient matrix, we quantified the improvement in the prediction risk of conjugate kernel ridge regression under two different scalings of first-step learning rate $\eta$. To the best of our knowledge, this is the first work that rigorously characterizes the precise asymptotics of kernel models (defined by neural networks) in the presence of feature learning.

We outline a few limitations of our current analysis as well as future directions.

- **Dependence between $W_1$ and $X$.** One crucial assumption that we make is that the trained weight matrix $W_1$ is *independent* of the data $\tilde{X}$ on which the CK is computed. While this does not cover the important scenario where feature learning and kernel evaluation are performed on the same data, our setting is very natural in the analysis of pretrained models or transfer learning, which would be an interesting extension.

- **Scaling of learning rate.** Our findings illustrate that different learning rate scalings such as $\eta = \Theta(1)$ and $\eta = \Theta(\sqrt{N})$ result in drastically different behavior. One natural question to ask is whether there exists a "phase transition" in between the two regimes that dictates whether the GET holds. Interestingly, [RGKZ21] showed that instead of breaking the near-orthogonality of the weights $W$ (via large gradient step), one can also introduce sufficiently large low-rank shifts to the input $X$ to enable the initial RF model to fit a nonlinear $f^*$. Intuitively, this may be due to the "dual" relation of the inputs $X$ and the weights $W$ in the CK model.

**Acknowledgement**

The authors would like to thank (in alphabetical order) Konstantin Donhauser, Zhou Fan, Hong Hu, Masaaki Imaizumi, Ryo Karakida, Bruno Loureiro, Yue M. Lu, Atsushi Nitanda, Sejun Park, Ji Xu, Yiqiao Zhong for discussions and feedback on the manuscript.

JB was supported by NSERC Grant [2020-06904], CIFAR AI Chairs program, Google Research Scholar Program and Amazon Research Award. MAE was supported by NSERC Grant [2019-06167], Connaught New Researcher Award, CIFAR AI Chairs program, and CIFAR AI Catalyst grant. TS was partially supported by JSPS KAKENHI (20H00576) and JST CREST. ZW was supported by NSF Grant DMS-2055340. DW was partially supported by a Borealis AI Fellowship. Part of this work was completed when DW interned at Microsoft Research (hosted by GY).

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
