# Table of Contents

# A Background and additional results

## A.1 Additional experiments

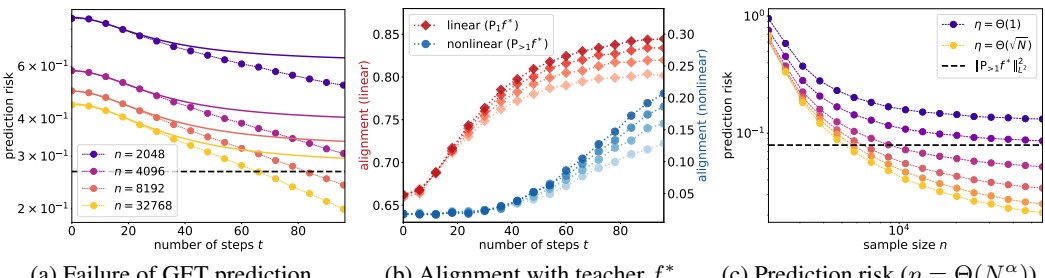

(a) Failure of GET prediction.     (b) Alignment with teacher $f^*$.     (c) Prediction risk ($\eta = \Theta(N^\alpha)$).

Figure 4: (a) Prediction risk of ridge regression on trained CK features after multiple GD steps with $\eta = \Theta(1)$: dots represent empirical simulations (averaged over 50 runs) and solid curves are asymptotic values predicted by the GET. We set $\sigma = \sigma^* = \text{ReLU}$, $\psi_2 = 2$, $d = 512$, $\lambda = 10^{-4}$, and $\eta = 0.1$. (b) Alignment between the student $f_\lambda^t$ and the linear (red) and nonlinear (blue) components of the teacher model (A.1). Darker colors correspond to larger sample size $n = \{2^{11}, 2^{12}, 2^{13}, 2^{15}\}$. (c) Prediction risk of ridge regression on trained CK features after one gradient step (empirical simulation, $d = 1024$): brighter color represents larger step size scaled as $\eta = N^\alpha$ for $\alpha \in [0, 1/2]$. We set $\sigma = \sigma^* = \text{SoftPlus}$, $\psi_2 = 2$, $\lambda = 10^{-3}$, and $\sigma_\varepsilon = 0.1$.

**Failure cases of GET.** It is worth noting that the Gaussian equivalence property (Theorem 3) may no longer hold if we train the features longer. In particular, because of our mean-field parameterization, the first-layer weight $W$ needs to travel sufficiently far away from initialization to achieve small training loss (see Figure 2). Hence in our experimental simulations (where $n, d, N$ are large but finite), as the number of steps $t$ increases, we expect the Gaussian equivalence predictions to become inaccurate at some point. This transition is empirically demonstrated in Figure 4(a). Observe that for larger $t$, the GET predictions *overestimate* the test loss; one possible explanation is that the trained kernel can learn nonlinear functions (which we show in Section 5 for one gradient step with $\eta = \Theta(\sqrt{N})$ and specific choices of $f^*$), which the GET cannot capture.

We provide additional empirical evidence on this explanation in Figure 4(b). To track the learning of the linear and nonlinear components of $f^*$, we recall the orthogonal decomposition:

$$f^*(\boldsymbol{x}) = \underbrace{\mu_0^* + \mu_1^* \langle \boldsymbol{x}, \boldsymbol{\beta}_* \rangle}_{f_L^*(\boldsymbol{x})} + \underbrace{\mathsf{P}_{>1} f^*(\boldsymbol{x})}_{f_{NL}^*(\boldsymbol{x})}.$$

Denote the CK ridge estimator on the feature map after $t$ gradient steps $\boldsymbol{x} \to \sigma(\boldsymbol{W}_t^\top \boldsymbol{x})$ as $f_\lambda^t$. We estimate the following alignment quantities (we normalize $f_L^*$ and $f_{NL}^*$ to have unit $L^2$-norm):

$$\textit{Linear} \text{ component: } \langle f_L^*, f_\lambda^t \rangle_{L^2(\mathbb{R}^d, \Gamma)}. \quad \textit{Nonlinear} \text{ component: } \langle f_{NL}^*, f_\lambda^t \rangle_{L^2(\mathbb{R}^d, \Gamma)}. \quad (A.1)$$

In Figure 4(b), we observe that the student model $f_\lambda^t$ first aligns with the linear component of the teacher $f_L^*$; on the other hand, when the student model begins to learn the nonlinear component $f_{NL}^*$ (at $\sim 30$ gradient steps), the GET predictions (Figure 4(a)) overestimate the prediction risk.

**Large learning rate (SoftPlus).** In Figure 4(c) we repeat the large learning rate experiment in Section 5 for different nonlinearity $\sigma = \sigma^* = \text{SoftPlus}$, for which $\tau^* \approx 0.03 > 0$, and hence the upper bound in Theorem 7 is *non-vanishing*. In this case, we observe that the prediction risk of the CK ridge regression model (after one feature learning step) is also non-vanishing even when the step size is large; this indicates that although we do not compute the exact risk in Theorem 7, the upper-bounding quantity $\tau^*$ in (5.1) has predictive power on the actual prediction risk.

**Impact of data splitting.** As previously mentioned, our current theoretical analysis requires the first-layer $W$ and second-layer $a$ to be learned from independent training data, i.e., a "data-splitting" procedure. Note that from Lemmas 10,11, and 22 one can easily deduce that when the sample size becomes large ($\psi_1 = n/d \to \infty$), the gradient at each neuron becomes proportional to $\boldsymbol{\beta}_*$ which no longer depends on $X$. In other words, if we denote $\tilde{\mathcal{R}}(\lambda)$ as the CK ridge prediction risk *without*

the data-splitting procedure, then $\tilde{\mathcal{R}}(\lambda) = \mathcal{R}(\lambda) + o_{\psi_1, \mathbb{P}}(1)$. Consequently, as we gradually increase the sample size $n$ (or ratio $\psi_1$), we expect the behavior of training $\boldsymbol{W}$ and $\boldsymbol{a}$ on the same data $\boldsymbol{X}$ to become more aligned with our theoretical results.

This intuition is confirmed in Figure 5, where we plotted our theoretical prediction (for the data-splitting setting) against empirical simulations where $\boldsymbol{W}$ and $\boldsymbol{a}$ are trained on the same data. In Figure 5(a) we observe that our asymptotic formula for the small learning rate regime (Theorem 5) accurately tracks the risk curve for $\psi_1 > 2$ but deviates when $\psi_1$ is small (subfigure). Moreover, Figure 5(b) shows that without data-splitting, the trained CK under large learning rate may also outperform the kernel lower bound for large $\psi_1$. To sum up, in both cases, we observe similar learning behavior with or without the data-splitting procedure (especially in the large $\psi_1$ regime).

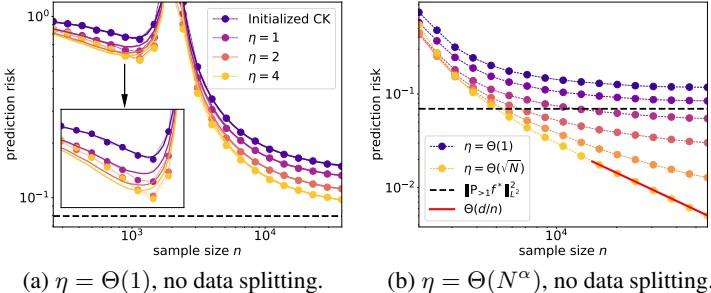

(a) $\eta = \Theta(1)$, no data splitting.     (b) $\eta = \Theta(N^\alpha)$, no data splitting.

Figure 5: Prediction risk of ridge regression on trained CK. Dots represent empirical simulations *without data-splitting*, i.e., $\boldsymbol{W}$, $\boldsymbol{a}$ trained on the same data (d=1024), and solid lines are theoretical predictions from the data-splitting setting. (a) $\eta = \Theta(1)$, $\sigma = \tanh$, $\sigma^* = $ SoftPlus, $\psi_2 = 2$, $\lambda = 10^{-4}$, $\sigma_\varepsilon = 0.25$. (b) $\eta = N^\alpha$ for $\alpha \in [0, 1/2]$; brighter color represents larger $\alpha$. We set $\sigma = \sigma^* = \tanh$, $\psi_2 = 2$, $\lambda = 10^{-3}$, $\sigma_\varepsilon = 0.1$.

## A.2 Additional related works

**The kernel regime and beyond.** The neural tangent kernel (NTK) [JGH18] describes the learning dynamics of wide neural network under specific parameter scaling. Such description is based on *linearizing* the NN around its initialization, and the limiting kernel can be computed for various architectures [ADH+19, Yan20]. Thanks to strong convexity of the kernel objective, global convergence rate guarantees of gradient descent can be established [DZPS19, JT20]. As mentioned in Section 1.2, this first-order Taylor expansion fails to explain the *adaptivity* of NNs; therefore, recent works also analyzed higher-order approximations of the training dynamics [DGA20, HY20]. Noticeably, a quadratic model (i.e., second-order approximation) can outperform kernel (NTK) estimators in certain settings [AZLL19, BL20].

In contrast to the aforementioned local approximations (via Taylor expansion and truncation), the *mean-field* regime (e.g., [NS17, MMN18, CB18]) deals with a different scaling limit under which the evolution of parameters can be described by some partial differential equation (for comparison between regimes see [WGL+20, GSJW20]). While the mean-field limit can capture the presence of feature learning [CB20, Ngu21], *quantitative* guarantees often require additional conditions such as KL regularization [NWS22, Chi22]. Note that our parameterization (1.1) mirrors the mean-field scaling, but we circumvent the difficulty of analyzing the nonlinear PDE because only the "early phase" (one gradient step) is considered.

Finally, we highlight two concurrent papers that studied the mean-field dynamics of two-layer NNs (under one-pass SGD) in the high-dimensional asymptotic regime, and showed learnability results for certain target functions. [ABAM22] established a separation between NNs and kernel methods in learning "staircase-like" functions on hypercube; [VSL+22] analyzed how the model width and step size impact the learning of a well-specified two-layer NN teacher model.

**Spectrum of kernel random matrices.** Kernel matrices in the proportional regime was first analyzed by [EK10] through Taylor expansion, and later their limiting spectra were fully described by [CS13, DV13, FM19]. As an extension of kernel random matrices, the CK matrix has also been studied in [PW17, Péc19, BP21, BP22] and [LLC18, FW20, WZ21], using the moment method and the Stieltjes transform method, respectively. In addition, the spectrum and concentration behavior of the NTK matrix were elaborated in [MZ20, FW20, WZ21]. We remark that based on these prior

results on the NTK of two-layer NNs, one can check our large learning rate $\eta = \Theta(\sqrt{N})$ satisfies $\sqrt{N}\eta \cdot \lambda_{\max}(\boldsymbol{F}) = \Theta_{d,\mathbb{P}}(1)$, where $\boldsymbol{F}$ is the *Fisher information matrix*; heuristically speaking, this means that the chosen step size is not unreasonably large (under first-order approximation).

### A.3 Linearity of kernel ridge regression

As previously mentioned, our kernel ridge regression lower bound (Proposition 1) is a simple combination of existing results, which we briefly outline below.

**Linear regression on input.** We first discuss the prediction risk of the ridge regression estimator on the input features. Recall that under Assumptions 1, we may write: $f^*(\boldsymbol{x}) = \mu_1^*\langle \boldsymbol{x}, \boldsymbol{\beta}_* \rangle + \mathsf{P}_{>1}f^*(\boldsymbol{x})$. Given the ridge regression estimator on the input features: $\hat{\boldsymbol{\theta}}_{\mathrm{Lin}} \triangleq (\boldsymbol{X}^\top \boldsymbol{X} + \lambda n \boldsymbol{I}_d)^{-1} \boldsymbol{X}^\top \boldsymbol{y}$, we have the following bias-variance decomposition,

$$\mathcal{R}_{\mathrm{Lin}}(\lambda) = \underbrace{\mathbb{E}_{\boldsymbol{x}}\Big( f^*(\boldsymbol{x}) - \boldsymbol{x}^\top (\boldsymbol{X}^\top \boldsymbol{X} + \lambda n \boldsymbol{I}_d)^{-1} \boldsymbol{X}^\top f^*(\boldsymbol{X}) \Big)^2}_{\text{Bias}}$$
$$+ \underbrace{\sigma_\varepsilon^2 \operatorname{Tr}\Big( (\boldsymbol{X}^\top \boldsymbol{X} + \lambda n \boldsymbol{I}_d)^{-2} \boldsymbol{X}^\top \boldsymbol{X} \Big)}_{\text{Variance}} + o_{d,\mathbb{P}}(1).$$

Following a similar computation as [BMR21, Theorem 4.13] and using the asymptotic formulae in [DW18, WX20], we can derive the following expression,

$$\mathcal{R}_{\mathrm{Lin}}(\lambda) \xrightarrow{\mathbb{P}} \frac{\bar{m}'(-\lambda)}{\bar{m}^2(-\lambda)} \cdot \frac{\mu_1^{*2}}{(1+\bar{m}(-\lambda))^2} + (\sigma_\varepsilon^2 + \mu_2^{*2}) \cdot \left( \frac{\bar{m}'(-\lambda)}{\bar{m}^2(-\lambda)} - 1 \right) + \mu_2^{*2}, \qquad \text{(A.2)}$$

where $\bar{m}(-\lambda) > 0$ is the Stieltjes transform of the limiting eigenvalue distribution of $\frac{1}{n}\boldsymbol{X}\boldsymbol{X}^\top$. Observe that $\mathcal{R}_{\mathrm{Lin}}(\lambda) \geq \mu_2^{*2}$. In addition, as shown in [DW18, WX20], the optimal ridge regularization and the corresponding prediction risk can be written as

$$\lambda_{\mathrm{opt}} = \frac{\sigma_\varepsilon^2 + \mu_2^{*2}}{\psi_1 \mu_1^{*2}}, \quad \mathcal{R}_{\mathrm{Lin}}(\lambda_{\mathrm{opt}}) \xrightarrow{\mathbb{P}} \frac{\sigma_\varepsilon^2 + \mu_2^{*2}}{\lambda_{\mathrm{opt}} \bar{m}(-\lambda_{\mathrm{opt}})} - \sigma_\varepsilon^2. \qquad \text{(A.3)}$$

**Lower bound for RF/kernel ridge regression.** First note that for RF models (3.1), the lower bound $\mu_2^{*2}$ is directly implied by the GET [HL20] under Assumption 1 (see Fact 4). For inner-product kernels[3] in (3.2), if $g : \mathbb{R} \to \mathbb{R}$ is a smooth function in a neighborhood of 0, then the same lower bound can be obtained from [BMR21, Theorem 4.13] (observe that the bias is lower bounded by $\|\mathsf{P}_{>1}f^*\|_{L^2}^2$). Finally, for the (first-layer) NTK, the ridge regression estimator is given as

$$\hat{f}_{\mathrm{NTK}}(\boldsymbol{x}) = \boldsymbol{g}^\top (\boldsymbol{K} + \lambda \boldsymbol{I})^{-1} \boldsymbol{y},$$
$$\text{where } \boldsymbol{g}_i = \frac{1}{Nd} \sum_{k=1}^N \langle \boldsymbol{x}, \boldsymbol{x}_i \rangle \sigma'(\langle \boldsymbol{x}, \boldsymbol{w}_k \rangle) \sigma'(\langle \boldsymbol{x}_i, \boldsymbol{w}_k \rangle),$$
$$\text{and } \boldsymbol{K}_{ij} = \frac{1}{Nd} \sum_{k=1}^N \langle \boldsymbol{x}_i, \boldsymbol{x}_j \rangle \sigma'(\langle \boldsymbol{x}_i, \boldsymbol{w}_k \rangle) \sigma'(\langle \boldsymbol{x}_j, \boldsymbol{w}_k \rangle).$$

Define the orthogonal decomposition $\sigma'(z) = b_0 + \sigma'_\perp(z)$, where $b_0 = \mu_1 = \mathbb{E}[\sigma'(z)]$, $b_1^2 = \mathbb{E}[\sigma'(z)^2] - b_0^2$, for $z \sim \mathcal{N}(0,1)$. Similar to [AP20, MZ20], we make the following linearization:

$$\boldsymbol{g} \approx \bar{\boldsymbol{g}} \triangleq \frac{1}{d} \cdot b_0^2 \boldsymbol{X}\boldsymbol{x}, \quad \boldsymbol{K} \approx \bar{\boldsymbol{K}} \triangleq \frac{1}{d} \cdot b_0^2 \boldsymbol{X}\boldsymbol{X}^\top + b_1^2 \boldsymbol{I}.$$

The error of this substitution has been studied in [MZ20, Lemma B.8] [WZ21, Theorem 2.7], which, together with [BMR21, Theorem 4.13], entail the following equivalence under Assumption 1,

$$\mathcal{R}_{\mathrm{NTK}}(\lambda) = \mathcal{R}_{\mathrm{Lin}}\left( \frac{\lambda + b_1^2}{b_0^2 \psi_1} \right) + o_{d,\mathbb{P}}(1),$$

where $\mathcal{R}_{\mathrm{Lin}}$ is the prediction risk of the ridge regression estimator on the input features defined in (A.2). Hence, the linear lower bound (3.3) directly applies; in fact, the prediction risk is lower-bounded by the optimal ridge regression estimator on the input (A.3).

---

[3] Similar result can also be shown for Euclidean distance kernels following the analysis in [EK10, Thm. 2.2].

**Kernel lower bound under polynomial scaling.** For high-dimensional input $x$ uniform on sphere or hypercube, [GMMM21, MMM21] showed that RF and kernel ridge estimators can learn at most a degree-$k$ polynomial when $n = \mathcal{O}(d^{k+1-\varepsilon})$ for small $\varepsilon > 0$; for the proportional scaling $n \asymp d$, this implies our lower bound $\|\mathsf{P}_{>1}f^*\|_{L^2}^2$ (but under different input assumptions). [DWY21] provided a similar result for more general data distributions and a class of rotation invariant kernels based on power series expansion, but the dependence on $k$ is not sharp enough to recover the linear lower bound in Proposition 1.

# B  Matrix concentration properties

## B.1  Norm control of gradient matrix

In this section we establish a few important properties of the gradient matrix defined in (2.1). For our later analysis, a key quantity to control is the entry-wise 2-$\infty$ matrix norm defined as

$$\|\boldsymbol{M}\|_{2,\infty} := \max_{1 \le i \le N} \|\boldsymbol{m}_i\|,$$

for any matrix $\boldsymbol{M} \in \mathbb{R}^{d \times N}$ with the $i$-th column $\boldsymbol{m}_i \in \mathbb{R}^d$ and $1 \le i \le N$. It is easy to verify that

$$\|\boldsymbol{M}\|_{2,\infty} \le \|\boldsymbol{M}\| \le \|\boldsymbol{M}\|_F \le \sqrt{N}\|\boldsymbol{M}\|_{2,\infty}.$$

In addition, for the Hadamard product with rank-1 matrix, we have the following property.

**Fact 9.** *Given* $\boldsymbol{mn}^\top \odot \boldsymbol{M} = \mathrm{diag}(\boldsymbol{m})\boldsymbol{M}\,\mathrm{diag}(\boldsymbol{n})$ *for* $\boldsymbol{m} \in \mathbb{R}^m, \boldsymbol{n} \in \mathbb{R}^n, \boldsymbol{M} \in \mathbb{R}^{m \times n}$, *we have*

$$\left\|\boldsymbol{mn}^\top \odot \boldsymbol{M}\right\| \le \|\mathrm{diag}(\boldsymbol{m})\| \cdot \|\boldsymbol{M}\| \cdot \|\mathrm{diag}(\boldsymbol{n})\| = \|\boldsymbol{m}\|_\infty \|\boldsymbol{M}\|\|\boldsymbol{n}\|_\infty.$$

### B.1.1  Norm bounds for the first gradient step

We begin with the first gradient step. Recall the definition of the gradient matrix under the squared loss (we omit the learning rate $\eta$ and prefactor $\sqrt{N}$):

$$
\begin{aligned}
\boldsymbol{G}_0 &= -\frac{1}{n}\boldsymbol{X}^\top\left[\left(\frac{1}{\sqrt{N}}\left(\frac{1}{\sqrt{N}}\sigma(\boldsymbol{X}\boldsymbol{W}_0)\boldsymbol{a} - \boldsymbol{y}\right)\boldsymbol{a}^\top\right) \odot \sigma'(\boldsymbol{X}\boldsymbol{W}_0)\right] \\
&= \underbrace{\frac{1}{n} \cdot \frac{\mu_1}{\sqrt{N}}\boldsymbol{X}^\top \boldsymbol{y}\boldsymbol{a}^\top}_{A} + \underbrace{\frac{1}{n} \cdot \frac{1}{\sqrt{N}}\boldsymbol{X}^\top\left(\boldsymbol{y}\boldsymbol{a}^\top \odot \sigma'_\perp(\boldsymbol{X}\boldsymbol{W}_0)\right)}_{B} \\
&\quad \underbrace{-\frac{1}{n} \cdot \frac{1}{N}\boldsymbol{X}^\top\left(\sigma(\boldsymbol{X}\boldsymbol{W}_0)\boldsymbol{a}\boldsymbol{a}^\top \odot \sigma'(\boldsymbol{X}\boldsymbol{W}_0)\right)}_{C},
\end{aligned}
\tag{B.1}
$$

where we utilized the orthogonal decomposition: $\sigma'(z) = \mu_1 + \sigma'_\perp(z)$. Due to Stein's lemma, we know that $\mathbb{E}[z\sigma(z)] = \mathbb{E}[\sigma'(z)] = \mu_1$, and hence $\mathbb{E}[\sigma'_\perp(z)] = 0$ for $z \sim \mathcal{N}(0,1)$. The following lemma provides norm control for the above decomposition under the same *Gaussian initialization* for $\boldsymbol{W}_0$, $\boldsymbol{a}$ and $\boldsymbol{X}$ as Assumption 1.

**Lemma 10.** *Assume that* $f^* \in L^2(\mathbb{R}^d, \Gamma)$, *and both* $f^*$ *and* $\sigma$ *are Lipschitz functions. Then*

(i) $\mathbb{E}\|\boldsymbol{A}\|_{2,\infty} \le \mathbb{E}\|\boldsymbol{A}\| \le \mathbb{E}\|\boldsymbol{A}\|_F \le C\sqrt{\frac{d}{nN} + \frac{1}{N}}$,

(iii) $\mathbb{E}\|\boldsymbol{C}\| \le \mathbb{E}\|\boldsymbol{C}\|_F \le \frac{C}{N}\sqrt{1 + \frac{d}{n}}$.

*Furthermore, we have the following probability bounds.*

(i) $\mathbb{P}\left(\|\boldsymbol{A}\|_F \ge C\left(\sqrt{\frac{d}{nN}} + \sqrt{\frac{1}{N}}\right)\right) \le C'\left(e^{-cn} + e^{-cN}\right)$,

$\mathbb{P}\left(\|\boldsymbol{A}\|_F \le C\sqrt{\frac{d}{nN}}\right) \le C'\left(e^{-c\min\left\{\frac{nd^2}{(n^2+d^2)}, \frac{nd}{n+d}\right\}} + e^{-cN} + e^{-cn}\right)$, *and*

$\mathbb{P}\left(\|\boldsymbol{A}\|_{2,\infty} \ge C\frac{(\sqrt{n}+\sqrt{d})\log n}{N\sqrt{n}}\right) \le C'\left(e^{-c\frac{(\sqrt{n}+\sqrt{d})^2}{n}\log^2 n} + e^{-cn} + Ne^{-c\log^2 n}\right)$.

(ii) $\mathbb{P}\Big(\|\boldsymbol{B}\| \geq C \frac{(\sqrt{n}+\sqrt{d})(\sqrt{n}+\sqrt{N})\log^2 n}{n\sqrt{Nd}}\Big) \leq C'\Big((n+N)e^{-c\log^2 n} + e^{-(\sqrt{n}+\sqrt{d})^2} + e^{-cN} + e^{-cd}\Big),$

$\mathbb{P}\Big(\|\boldsymbol{B}\|_F \geq C \frac{\sqrt{n}+\sqrt{d}}{\sqrt{nN}}\Big) \leq C'\Big(e^{-cn} + e^{-cN} + e^{-c(\sqrt{n}+\sqrt{d})^2}\Big).$

(iii) $\mathbb{P}\Big(\|\boldsymbol{C}\|_F \geq C \frac{(\sqrt{d}+\sqrt{n})\log n \log N}{\sqrt{nN}}\Big) \leq C'\Big(Ne^{-cN} + ne^{-cd} + ne^{-c\log^2 n} + Ne^{-c\log^2 N}\Big).$

*Here all constants $C, C', c > 0$ only depend on $\lambda_\sigma$, $\mu_1$, $\sigma_\varepsilon$ and $\|f^*\|_{L^2(\mathbb{R}^d,\Gamma)}$.*

**Remark.** *In Lemma 10, we do not use the proportional scaling to simplify the expressions. This is because the dependence on $n, d, N$ needs to be tracked separately in some of our calculations.*

**Proof.** We analyze the three matrices of interest separately.

**Part** $(i)$**.** We first upper-bound $\|\boldsymbol{A}\|_F^2$. Notice that

$$\frac{n\sqrt{N}}{\mu_1}\|\boldsymbol{A}\|_F \leq \|\boldsymbol{X}^\top f^*(\boldsymbol{X})\boldsymbol{a}^\top\|_F + \|\boldsymbol{X}^\top \boldsymbol{\varepsilon}\boldsymbol{a}^\top\|_F$$
$$\leq \|\boldsymbol{X}\|(\|f^*(\boldsymbol{X})\| + \|\boldsymbol{\varepsilon}\|)\|\boldsymbol{a}\|. \tag{B.2}$$

We know that Gaussian random matrices and vectors satisfy

$$\mathbb{E}\|\boldsymbol{\varepsilon}\|^2 = \sigma_\varepsilon^2 n, \quad \mathbb{E}\|f^*(\boldsymbol{X})\|^2 = n\|f^*\|_{L^2(\mathbb{R}^d,\Gamma)}^2, \tag{B.3}$$

$$\mathbb{E}\|\boldsymbol{a}\|^2 = 1, \quad \mathbb{E}\|\boldsymbol{X}\|^2 \leq C_0(n+d), \tag{B.4}$$

where the last inequality is from [Ver18, Exercise 4.6.2]. Based on Cauchy-Schwarz inequality, we can employ (B.3) and (B.4) to obtain

$$\mathbb{E}\|\boldsymbol{A}\|_{2,\infty} \leq \mathbb{E}\|\boldsymbol{A}\| \leq \mathbb{E}\|\boldsymbol{A}\|_F \leq C_1\sqrt{\frac{d}{nN} + \frac{1}{N}},$$

where constant $C_1 > 0$ only depends on $\mu_1$, $\sigma_\varepsilon$ and $\|f^*\|_{L^2(\mathbb{R}^d,\Gamma)}$. As for the probability bound, we use the Lipschitz concentration property (e.g., see [Ver18, Theorem 5.2.2]) of $\|\boldsymbol{a}\|$, $\|\boldsymbol{\varepsilon}\|$ and $\|f^*(\boldsymbol{X})\|$, and apply [Ver18, Corollary 7.3.3] for $\|\boldsymbol{X}\|$ to obtain

$$\mathbb{P}\big(\|\boldsymbol{\varepsilon}\| \geq \sigma_\varepsilon\sqrt{n}\big) \leq 2e^{-cn}, \quad \mathbb{P}\Big(\big|\|\boldsymbol{a}\| - 1\big| \geq \frac{1}{2}\Big) \leq 2e^{-cN}, \tag{B.5}$$

$$\mathbb{P}\Big(\big|\|f^*(\boldsymbol{X})\| - \|f^*\|_{L^2(\mathbb{R}^d,\Gamma)}\sqrt{n}\big| \geq \frac{1}{2}\|f^*\|_{L^2(\mathbb{R}^d,\Gamma)}\sqrt{n}\Big) \leq 2e^{-cn}, \tag{B.6}$$

$$\mathbb{P}\Big(\|\boldsymbol{X}\| \geq \sqrt{n} + \sqrt{d} + t\Big) \leq 2e^{-ct^2}, \tag{B.7}$$

for any $t \geq 0$. Hence, from (B.2), we arrive at

$$\mathbb{P}\left(\|\boldsymbol{A}\|_F \geq \sqrt{\frac{d}{nN}} + \sqrt{\frac{1}{N}} + t\right) \leq 4\Big(e^{-cn} + e^{-cN} + e^{-ct^2 nN}\Big).$$

Note that the same probability bounds also applies to $\|\boldsymbol{A}\|$ and $\|\boldsymbol{A}\|_{2,\infty}$. Thus, we may take $t = \sqrt{\frac{1}{N}}$ to obtain the desired result. The lower bound on $\|\boldsymbol{A}\|_F$ follows from a similar computation, the details of which can be found in [BES⁺22, Appendix B.3]. As for the last inequality on $\|\boldsymbol{A}\|_{2,\infty}$, by definition we know that

$$\|\boldsymbol{A}\|_{2,\infty} \leq \frac{\mu_1}{n\sqrt{N}}\|\boldsymbol{X}\|(\|f^*(\boldsymbol{X})\| + \|\boldsymbol{\varepsilon}\|)\|\boldsymbol{a}\|_\infty.$$

The desired result can be obtained from the tail bound on the sup-norm of Gaussian random vector, $\mathbb{P}\Big(\|\boldsymbol{a}\|_\infty \leq t/\sqrt{N}\Big) \geq 1 - 2Ne^{-ct^2}$, in combination with (B.5), (B.6) and (B.7).

**Part** $(ii)$**.** As a result of Fact 9, we have

$$\|\boldsymbol{B}\| \leq \frac{1}{n\sqrt{N}}\|\boldsymbol{X}\|\|\boldsymbol{a}\|_\infty(\|f^*(\boldsymbol{X})\|_\infty + \|\boldsymbol{\varepsilon}\|_\infty)\|\sigma'_\perp(\boldsymbol{X}\boldsymbol{W}_0)\|. \tag{B.8}$$

We first control the operator norm of the random feature matrix $\sigma'_{\perp}(\boldsymbol{X}\boldsymbol{W}_0)$. Since $\sigma'_{\perp}$ is centered, [FW20, Lemma D.4] implies that

$$\mathbb{P}\Big(\|\sigma'_{\perp}(\boldsymbol{X}\boldsymbol{W}_0)\| \geq C(\sqrt{n}+\sqrt{N})\lambda_\sigma B, \mathcal{A}_B\Big) \leq 2e^{-cN},$$

where event $\mathcal{A}_B$ is defined by

$$\mathcal{A}_B := \left\{ \|\boldsymbol{W}_0\| \leq B, \sum_{i=1}^{N}(\|\boldsymbol{w}_i^0\|^2 - 1)^2 \leq B^2 \right\},$$

given any constant $B > 0$. Following the proof of Proposition 3.3 in [FW20], we can obtain that

$$\mathbb{P}\left( \sum_{i=1}^{N}(\|\boldsymbol{w}_i^0\|^2 - 1)^2 \geq 4t^2 \right) \leq 2e^{N\log 5 - cd\min\{t^2, t\}}, \tag{B.9}$$

for any $t \geq 0$. Besides, inequality (B.7) implies that for any $t \geq 0$,

$$\mathbb{P}\left( \|\boldsymbol{W}_0\| \leq c'\sqrt{\frac{N}{d}} \right) \geq 1 - 2e^{-cd}.$$

By choosing $t = c'\sqrt{\frac{N}{d}}$ in (B.9) and $B := c'\sqrt{\frac{N}{d}}$ for sufficient large $c' > 0$, we can claim that there exists sufficient large constant $c > 0$ such that

$$\mathbb{P}(\mathcal{A}_B^c) \leq 2e^{-cd} + 2e^{-cN}.$$

Combining the above inequalities, we have

$$\mathbb{P}\left( \|\sigma'_{\perp}(\boldsymbol{X}\boldsymbol{W}_0)\| \geq C(\sqrt{n}+\sqrt{N})\sqrt{\frac{N}{d}} \right) \leq 4e^{-cN} + 2e^{-cd}. \tag{B.10}$$

In addition, the following tail bound is due to property of (sub-)Gaussian random variables:

$$\mathbb{P}\Big(\|\boldsymbol{a}\|_\infty \leq t_1/\sqrt{N}\Big) \geq 1 - 2Ne^{-ct_1^2}, \quad \mathbb{P}(\|\boldsymbol{\varepsilon}\|_\infty \leq t_2) \geq 1 - 2ne^{-ct_2^2}, \tag{B.11}$$

for any $t_1, t_2 \geq 0$. Because $f^*$ is Lipschitz, $f^*(\boldsymbol{X})$ is a sub-Gaussian random vector satisfying

$$\mathbb{P}(\|f^*(\boldsymbol{X})\|_\infty \leq t_2) \geq 1 - 2ne^{-ct_2^2}.$$

Let $t_1 = t_2 = \log n$. Applying all these three tail bounds (B.10) and (B.7), (B.8) gives us the first part of the probability bound in $(ii)$. As for the second part, following the observation

$$\|\boldsymbol{B}\|_F \leq \frac{\mu_1}{n\sqrt{N}}\|\boldsymbol{X}\|\|\boldsymbol{y}\boldsymbol{a}^\top \odot \sigma'_{\perp}(\boldsymbol{X}\boldsymbol{W}_0)\|_F \leq \frac{\mu_1\lambda_\sigma}{n\sqrt{N}}\|\boldsymbol{X}\|\|\boldsymbol{y}\|\|\boldsymbol{a}\|,$$

we can adopt (B.5), (B.6) and (B.7) to conclude the second probability bound.

**Part $(iii)$.** Finally, we analyze the lower-order term $\boldsymbol{C}$. Recall the definitions $\boldsymbol{X} = [\tilde{\boldsymbol{X}}, \ldots, \boldsymbol{x}_n]^\top$, $\boldsymbol{W}_0 = [\boldsymbol{w}_1^0, \ldots, \boldsymbol{w}_N^0]$ and $\boldsymbol{a} = [a_1, \ldots, a_N]^\top$. We first observe that

$$\mathbb{E}\|\sigma(\boldsymbol{X}\boldsymbol{W}_0)\boldsymbol{a}\boldsymbol{a}^\top \odot \sigma'(\boldsymbol{X}\boldsymbol{W}_0)\|_F^2 \leq \lambda_\sigma^2 \sum_{j=1}^{n}\sum_{k=1}^{N}\mathbb{E}\left( \sum_{i=1}^{N}a_ia_k\sigma(\boldsymbol{x}_j^\top\boldsymbol{w}_i^0) \right)^2,$$

$$=\lambda_\sigma^2 \sum_{j=1}^{n}\sum_{k=1}^{N}\sum_{i,l=1}^{N}\mathbb{E}[a_la_ia_k^2\sigma(\boldsymbol{x}_j^\top\boldsymbol{w}_i^0)\sigma(\boldsymbol{x}_j^\top\boldsymbol{w}_l^0)] = \lambda_\sigma^2 \sum_{j=1}^{n}\sum_{k=1}^{N}\sum_{i=1}^{N}\mathbb{E}[a_i^2a_k^2\sigma(\boldsymbol{x}_j^\top\boldsymbol{w}_i^0)^2],$$

$$\leq \frac{C'}{N^2}\sum_{j=1}^{n}\sum_{k=1}^{N}\sum_{i=1}^{N}\mathbb{E}[\sigma(\boldsymbol{x}_j^\top\boldsymbol{w}_i^0)^2] \leq C''n, \tag{B.12}$$

where the last inequality can be deduced by

$$\mathbb{E}[\sigma(\boldsymbol{x}^\top\boldsymbol{w})^2] = \mathbb{E}_{\boldsymbol{w}}\big[\mathbb{E}_{\boldsymbol{x}}[\sigma(\boldsymbol{x}^\top\boldsymbol{w})^2]\big] = \mathbb{E}_{\boldsymbol{w}}\big[\mathbb{E}_z[\sigma(\|\boldsymbol{w}\|z)^2]\big]$$

$$\leq 2\mathbb{E}_{\boldsymbol{w}}\big[\mathbb{E}_z(\sigma(\|\boldsymbol{w}\|z) - \sigma(z))^2\big] + 2\mathbb{E}_{\boldsymbol{w}}\big[\mathbb{E}_z[\sigma(z)^2]\big]$$
$$\leq 2\lambda_\sigma^2 \mathbb{E}_{\boldsymbol{w}}\big[(\|\boldsymbol{w}\| - 1)^2\big] + 2\mathbb{E}_{\boldsymbol{w}}\big[\mathbb{E}_z[\sigma(z)^2]\big] \leq 4\lambda_\sigma^2 + \mathbb{E}_z[\sigma(z)^2],$$

which is uniformly bounded by a constant. Therefore, by (B.4) and (B.12), we get

$$\mathbb{E}\|\boldsymbol{C}\|] \leq \mathbb{E}\|\boldsymbol{C}\|_F \leq \frac{1}{nN}\mathbb{E}[\|\boldsymbol{X}\|^2]^{\frac{1}{2}}\mathbb{E}[\|\sigma(\boldsymbol{X}\boldsymbol{W}_0)\boldsymbol{a}\boldsymbol{a}^\top \odot \sigma'(\boldsymbol{X}\boldsymbol{W}_0)\|_F^2]^{\frac{1}{2}} \leq \frac{C_3}{N}\sqrt{1 + \frac{d}{n}}.$$

As for the tail control, because of Fact 9, we consider the following upper-bound,

$$\|\boldsymbol{C}\| \leq \|\boldsymbol{C}\|_F \leq \frac{1}{nN}\|\boldsymbol{X}\|\|\sigma(\boldsymbol{X}\boldsymbol{W}_0)\boldsymbol{a}\|_\infty\|\boldsymbol{a}\|_\infty\|\sigma'(\boldsymbol{X}\boldsymbol{W}_0)\|_F \leq \frac{\lambda_\sigma}{\sqrt{nN}}\|\boldsymbol{X}\|\|\sigma(\boldsymbol{X}\boldsymbol{W}_0)\boldsymbol{a}\|_\infty\|\boldsymbol{a}\|_\infty,$$
(B.13)

where the last inequality is due to $|\sigma'|$ being upper-bounded by $\lambda_\sigma$.

To control $\|\sigma(\boldsymbol{X}\boldsymbol{W}_0)\boldsymbol{a}\|_\infty$, note that since $\boldsymbol{a}$ is centered by Assumption 1, we can apply Bernstein inequality for $\boldsymbol{a}$ and $\boldsymbol{W}$ conditioned on the event $\mathcal{M} := \big\{\big|\|\boldsymbol{x}_i\|/\sqrt{d} - 1\big| \leq 1/2, \ i \in [n]\big\}$. Conventionally, we denote $\|\cdot\|_{\psi_2}$ as the sub-Gaussian norm. Since $\big\|\|\boldsymbol{x}_i\| - \sqrt{d}\big\|_{\psi_2}$ is bounded by some absolute constant ([Ver18, Theorem 3.1.1]), we know that
$$\mathbb{P}(\mathcal{M}) \geq 1 - ne^{-cd}.$$

Notice that for any $j \in [n]$, $\sigma(\boldsymbol{x}_j^\top \boldsymbol{W}_0)\boldsymbol{a} = \sum_{i=1}^N a_i\sigma(\boldsymbol{x}_j^\top \boldsymbol{w}_i^0)$ is the sum of $N$ independent and centered sub-Exponential random variables, where, in terms of [FW20, Lemma D.5], the sub-Exponential norm $\|\cdot\|_{\psi_1}$ of each term is bounded by the sub-Gaussian norm of the entries as follows,

$$\|a_i\sigma(\boldsymbol{x}_j^\top \boldsymbol{w}_i^0)\|_{\psi_1} \leq \|a_i\|_{\psi_2}\|\sigma(\boldsymbol{x}_j^\top \boldsymbol{w}_i^0)\|_{\psi_2} \leq \frac{C\lambda_\sigma}{\sqrt{N}}\frac{\|\boldsymbol{x}_j\|}{\sqrt{d}} \leq \frac{3C\lambda_\sigma}{2\sqrt{N}},$$

for constant $C > 0$. Thus, by Bernstein inequality [Ver18, Theorem 2.8.1], for each $j \in [n]$,

$$\mathbb{P}\big(|\sigma(\boldsymbol{x}_j^\top \boldsymbol{W}_0)\boldsymbol{a}| \geq \log n\big) \leq 2e^{-c(\log n)^2}.$$

Then we take the union over all $\boldsymbol{x}_j$ and obtain $\|\sigma(\boldsymbol{X}\boldsymbol{W}_0)\boldsymbol{a}\|_\infty \leq \log n$ with probability at least $1 - 2ne^{-c(\log n)^2}$. Hence, by (B.7), (B.11) and (B.13), we get $\|\boldsymbol{C}\|_F \geq \frac{(\sqrt{d}+\sqrt{n}+t)\log n \log N}{\sqrt{n}N}$ with probability at most $2ne^{-c(\log n)^2} + 2Ne^{-c(\log N)^2} + ne^{-cd} + 2e^{-ct^2} + 2Ne^{-cN}$. Part $(iii)$ is established by choosing $t = \sqrt{d}$. This concludes the proof of the lemma. $\square$

Proposition 2 is a direct consequence of the above norm bounds.

**Proof of Proposition 2.** Notice that $\boldsymbol{G}_0 - \boldsymbol{A} = \boldsymbol{B} + \boldsymbol{C}$. In the proportional regime, by Lemma 10, there exist universal constants $C, c > 0$ such that

$$\mathbb{P}\bigg(\|\boldsymbol{G}_0 - \boldsymbol{A}\| \leq C\frac{\log^2 n}{n}\bigg) \geq 1 - ne^{-c\log^2 n}.$$

On the other hand, part $(i)$ in Lemma 10 implies that

$$\mathbb{P}\bigg(\|\boldsymbol{A}\| \geq \frac{C}{\sqrt{n}}\bigg) \geq 1 - e^{-cn},$$

for some constant $c, C > 0$. Here we used the fact $\|\boldsymbol{A}\| = \|\boldsymbol{A}\|_F$ because it is a rank-one matrix. Conditioning on the two events stated above, we have

$$\|\boldsymbol{G}_0 - \boldsymbol{A}\| \leq \frac{C}{\sqrt{n}}\frac{\log^2 n}{\sqrt{n}} \leq \frac{\log^2 n}{\sqrt{n}}\|\boldsymbol{A}\| \leq \frac{\log^2 n}{\sqrt{n}}(\|\boldsymbol{G}_0\| + \|\boldsymbol{G}_0 - \boldsymbol{A}\|).$$

As long as $n$ is sufficiently large such that $\frac{\log^2 n}{\sqrt{n}} < \frac{1}{2}$, we can obtain

$$\mathbb{P}\bigg(\|\boldsymbol{G}_0 - \boldsymbol{A}\| \leq \frac{2\log^2 n}{\sqrt{n}}\|\boldsymbol{G}_0\|\bigg) \geq 1 - ne^{-c\log^2 n} - e^{-cn},$$

which completes the proof. $\square$

### B.1.2 Decomposition of matrix A

Using the orthogonal decomposition (2.3), we can further decompose the rank-1 matrix $\boldsymbol{A}$ as follows

$$\boldsymbol{A} = \underbrace{\frac{1}{n} \cdot \frac{\mu_1 \mu_1^*}{\sqrt{N}} \boldsymbol{X}^\top \boldsymbol{X} \boldsymbol{\beta}_* \boldsymbol{a}^\top}_{\boldsymbol{A}_1} + \underbrace{\frac{1}{n} \cdot \frac{\mu_1}{\sqrt{N}} \boldsymbol{X}^\top (\mu_0^* \mathbf{1} + \mathsf{P}_{>1} f^*(\boldsymbol{X}) + \boldsymbol{\varepsilon}) \boldsymbol{a}^\top}_{\boldsymbol{A}_2}, \tag{B.14}$$

where we denote $\mathsf{P}_{>1} f^*(\boldsymbol{X}) := [\mathsf{P}_{>1} f^*(\boldsymbol{x}_1), \ldots, \mathsf{P}_{>1} f^*(\boldsymbol{x}_n)]^\top \in \mathbb{R}^n$. Similar to the previous Lemma 10, we have the following norm bound.

**Lemma 11.** *Assume that target function $f^* \in L^4(\mathbb{R}^d, \Gamma)$ is a Lipschitz function. We have*

*(i)* $\mathbb{E}\|\boldsymbol{A}_1\|_F \leq \frac{C}{\sqrt{N}}\left(1 + \frac{d}{n}\right)$ *and* $\mathbb{P}\left(\|\boldsymbol{A}_1\|_F \geq C\left(\frac{1}{\sqrt{N}} + \frac{d}{n\sqrt{N}}\right)\right) \leq C'(e^{-cN} + e^{-cn})$;

*(ii)* $\mathbb{E}\|\boldsymbol{A}_2\|_F \leq C\sqrt{\frac{d}{Nn}}$, *and when $n \geq d$,*

$$\mathbb{P}\left(\|\boldsymbol{A}_2\|_F^2 \geq \frac{Cd}{nN}\right) \leq C'(e^{-c\sqrt{n}} + e^{-cN} + ne^{-cd} + d^{-1}), \tag{B.15}$$

*for some constants $C, C', c > 0$ that only depend on $\mu_1$, $\sigma_\varepsilon$ and $f^*$.*

**Proof.** For simplicity, we denote $\mathsf{P}_{>1} f^*(\boldsymbol{x})$ by $f_{\mathrm{NL}}^*(\boldsymbol{x})$ and $\mathsf{P}_{>1} f^*(\boldsymbol{X})$ by $\boldsymbol{f}_{\mathrm{NL}}^* \in \mathbb{R}^n$.

**Part $(i)$.** The expectation follows from (B.4) and the following inequality,

$$\|\boldsymbol{A}_1\|_F \leq \frac{\mu_1 \mu_1^*}{n\sqrt{N}} \|\boldsymbol{X}\|^2 \|\boldsymbol{\beta}_*\| \|\boldsymbol{a}\| = \frac{\mu_1 \mu_1^*}{n\sqrt{N}} \|\boldsymbol{X}\|^2 \|\boldsymbol{a}\|.$$

The probability bound also follows from the same argument as Lemma 10.

**Part $(ii)$.** Following the proof of part $(i)$ in Lemma 10, we can further decompose $\|\boldsymbol{A}\|_F$ into

$$\|\boldsymbol{A}_2\|_F \leq \frac{\mu_1}{n\sqrt{N}} \|\boldsymbol{X}\| \|\boldsymbol{a}\| \left(\mu_0^* \sqrt{n} + \|\boldsymbol{f}_{\mathrm{NL}}^*\| + \|\boldsymbol{\varepsilon}\|\right).$$

Since $\mathsf{P}_{>1} f^*$ is a Lipschitz function as well, we can again apply the Lipschitz concentration (B.6). Hence, combining (B.7), (B.5) and (B.6), one can conclude the bound on the expectation of $\|\boldsymbol{A}_2\|_F$.

For the tail bound, we consider matrices

$$\boldsymbol{A}_2' := \frac{\mu_1}{n\sqrt{N}} \boldsymbol{X}^\top \boldsymbol{f}_{\mathrm{NL}}^* \boldsymbol{a}^\top, \quad \boldsymbol{A}_2'' := \frac{\mu_1}{n\sqrt{N}} \boldsymbol{X}^\top \boldsymbol{\varepsilon} \boldsymbol{a}^\top, \quad \boldsymbol{A}_2''' := \frac{\mu_0^* \mu_1}{n\sqrt{N}} \boldsymbol{X}^\top \mathbf{1} \boldsymbol{a}^\top,$$

whose squared Frobenius norms are given by $\|\boldsymbol{A}_2'\|_F^2 = \frac{\mu_1^2}{n^2 N} \boldsymbol{a}^\top \boldsymbol{a} \boldsymbol{f}_{\mathrm{NL}}^{*\top} \boldsymbol{X} \boldsymbol{X}^\top \boldsymbol{f}_{\mathrm{NL}}^*$, and

$$\|\boldsymbol{A}_2''\|_F^2 = \frac{\mu_1^2}{n^2 N} \boldsymbol{a}^\top \boldsymbol{a} \boldsymbol{\varepsilon}^\top \boldsymbol{X} \boldsymbol{X}^\top \boldsymbol{\varepsilon}, \quad \|\boldsymbol{A}_2'''\|_F^2 = \frac{\mu_0^{*2} \mu_1^2}{n^2 N} \boldsymbol{a}^\top \boldsymbol{a} \mathbf{1}^\top \boldsymbol{X} \boldsymbol{X}^\top \mathbf{1}.$$

Recall that (B.5) implies

$$\mathbb{P}\left(\|\boldsymbol{a}\|^2 \geq 4\right) \leq 2e^{-cN}. \tag{B.16}$$

Let us first address $\boldsymbol{A}_2''$. Due to (B.16), it suffices to control $\boldsymbol{\varepsilon}^\top \boldsymbol{X} \boldsymbol{X}^\top \boldsymbol{\varepsilon}$, whose expectation with respect to $\boldsymbol{\varepsilon}$ is $\sigma_\varepsilon^2 \operatorname{Tr}(\boldsymbol{X} \boldsymbol{X}^\top)$, and $\mathbb{E}[\operatorname{Tr}(\boldsymbol{X} \boldsymbol{X}^\top)] = nd$. Recalling the Lipschitz Gaussian concentration for $\|\boldsymbol{X}\|_F$ and (B.5), we know that for some constant $c > 0$, $\mathbb{P}(\mathcal{A}_\varepsilon) \geq 1 - 4e^{-cd}$, where $\mathcal{A}_\varepsilon := \{\|\boldsymbol{X}\|_F \leq \sqrt{nd}, \|\boldsymbol{X}\| \leq \sqrt{n} + \sqrt{d}\}$. This directly implies that

$$\|\boldsymbol{X} \boldsymbol{X}^\top\|_F \leq \|\boldsymbol{X}\|_F \|\boldsymbol{X}\| \leq \sqrt{nd}\left(\sqrt{n} + \sqrt{d}\right),$$

conditioned on event $\mathcal{A}_\varepsilon$. Thus, the Hanson-Wright inequality (Theorem 6.2.1 [Ver18]) indicates

$$\mathbb{P}\left(\boldsymbol{\varepsilon}^\top \boldsymbol{X} \boldsymbol{X}^\top \boldsymbol{\varepsilon} \geq t + 4\sigma_\varepsilon^2 nd\right) \leq \mathbb{P}\left(\boldsymbol{\varepsilon}^\top \boldsymbol{X} \boldsymbol{X}^\top \boldsymbol{\varepsilon} \geq t + \sigma_\varepsilon^2 nd \,\Big|\, \mathcal{A}_\varepsilon\right) + \mathbb{P}(\mathcal{A}_\varepsilon^c)$$

$$\leq \mathbb{P}\Big(\Big|\boldsymbol{\varepsilon}^\top \boldsymbol{X}\boldsymbol{X}^\top \boldsymbol{\varepsilon} - \sigma_\varepsilon^2 \|\boldsymbol{X}\|_F^2\Big| \geq t \Big| \mathcal{A}_\varepsilon\Big) + 4e^{-cd} \leq 2e^{-c\min\left\{\frac{t^2}{nd(n+d)}, \frac{t}{(\sqrt{d}+\sqrt{n})^2}\right\}} + 4e^{-cd}.$$

Thus, by choosing $t = nd$ and employing (B.16), we have

$$\mathbb{P}\Big(\|\boldsymbol{A}_2''\|_F^2 \geq \frac{Cd}{nN}\Big) \leq 6e^{-cd} + 2e^{-cN},$$

where we simplified the expression using the assumption that $n \geq d$.

As for $\|\boldsymbol{A}_2'\|_F^2$, $\|\boldsymbol{A}_2'''\|_F^2$, the moment computation in [BES$^+$22, Appendix B.3] yields

$$\mathbb{P}\Big(\|\boldsymbol{A}_2'\|_F^2 + \|\boldsymbol{A}_2'''\|_F^2 \geq \frac{Cd}{nN}\Big) \leq 2e^{-c\sqrt{n}} + ne^{-cd} + 2e^{-cN} + \frac{c}{d},$$

for some constant $C, c > 0$. The proof of (B.15) is completed by combining the above calculations.

$\square$

## B.2 Concentration of quadratic forms

In this section we establish a quadratic form concentration result which will be useful in the later analysis. Given $\eta = \Theta(1)$, we define $\bar{\mu} = \lim_{d\to\infty}\|f^*\|_{L^2(\mathbb{R}^d,\Gamma)}$, and

$$\theta_1 := \sqrt{\bar{\mu}^2 \psi_1^{-1} + \mu_1^{*2}} \cdot \mu_1 \eta, \quad \theta_2 := \mu_1 \mu_1^* \eta. \tag{B.17}$$

These two constants will appear in Theorem 5 when defining $\delta(\eta, \lambda, \psi_1, \psi_2)$. Notice that by (2.3) and Assumption 1, we have $\bar{\mu}^2 = \mu_1^{*2} + \mu_2^{*2}$.

The following quadratic concentration lemma is an adaptation from Lemma 2.7 and Lemma A.1 in [BS98]. We also refer readers to section B.5 in [BS10] for more details.

**Lemma 12.** *Define* $\boldsymbol{u} := \frac{\eta\mu_1}{n}\boldsymbol{X}^\top \boldsymbol{y}$ *where* $\boldsymbol{y} = f^*(\boldsymbol{X}) + \boldsymbol{\varepsilon}$. *Under the Assumption 1, consider any deterministic matrix* $\boldsymbol{D} \in \mathbb{R}^{d\times d}$ *with* $\|\boldsymbol{D}\| \leq C$ *uniformly for some constant* $C > 0$. *Then, as* $n/d \to \psi_1$ *proportionally, we have that*

$$\Big|\boldsymbol{u}^\top \boldsymbol{D}\boldsymbol{u} - (\theta_1^2 - \theta_2^2)\operatorname{tr}\boldsymbol{D} - \theta_2^2 \boldsymbol{\beta}_*^\top \boldsymbol{D}\boldsymbol{\beta}_*\Big|, \quad \Big|\boldsymbol{\beta}_*^\top \boldsymbol{D}\boldsymbol{u} - \theta_2 \boldsymbol{\beta}_*^\top \boldsymbol{D}\boldsymbol{\beta}_*\Big| \xrightarrow{\mathbb{P}} 0,$$

*where* $\theta_1$ *and* $\theta_2$ *are defined in* (B.17). *In addition, recalling that the nonlinear part of the target function is given as* $f_{\mathrm{NL}}^*(\boldsymbol{x}) := f^*(\boldsymbol{x}) - \mu_0^* - \mu_1^*\langle \boldsymbol{x}, \boldsymbol{\beta}_*\rangle$, *we have that*

$$\Big|\frac{1}{n}\boldsymbol{\beta}_*^\top \boldsymbol{D}\boldsymbol{X}^\top f_{\mathrm{NL}}^*(\boldsymbol{X})\Big| \xrightarrow{\mathbb{P}} 0.$$

We refer to Appendix B.4 in [BES$^+$22] for the detailed proof of this lemma.

## C  Proof for small learning rate ($\eta = \Theta(1)$)

### C.1  Gaussian equivalence for trained feature map

**The Gaussian equivalence property.** To validate Theorem 3, we follow the proof strategy of [HL20], which established the GET for RF models using the Lindeberg approach and leave-one-out arguments [EK18]. We remark that concurrent to our work, [MS22] proved the Gaussian equivalence property for a larger model class under an assumed central limit theorem, which is verified for two-layer RF or NTK models, and thus cannot directly imply our results on the trained features.

We first introduce the notations used in this section. Given weight matrix $\boldsymbol{W}$ and input $\boldsymbol{x}$, we define the feature vector $\boldsymbol{\phi}_{\boldsymbol{x}} = \frac{1}{\sqrt{N}}\sigma(\boldsymbol{W}^\top \boldsymbol{x}) \in \mathbb{R}^N$; similarly, given training data matrix $\tilde{\boldsymbol{X}} \in \mathbb{R}^{n\times d}$, the kernel feature matrix is given as $\boldsymbol{\Phi} = \frac{1}{\sqrt{N}}\sigma(\tilde{\boldsymbol{X}}\boldsymbol{W}) \in \mathbb{R}^{n\times N}$. Also, the Gaussian feature can be written as: $\bar{\boldsymbol{\phi}}_{\boldsymbol{x}} = \frac{1}{\sqrt{N}}\Big(\mu_1 \boldsymbol{W}^\top \boldsymbol{x} + \mu_2 \boldsymbol{z}\Big)$, and the corresponding matrix $\bar{\boldsymbol{\Phi}} = \frac{1}{\sqrt{N}}\Big(\mu_1 \tilde{\boldsymbol{X}}\boldsymbol{W} + \mu_2 \boldsymbol{Z}\Big)$,

where $\boldsymbol{z}, [\boldsymbol{Z}]_i \overset{\text{i.i.d.}}{\sim} \mathcal{N}(0, \boldsymbol{I})$ for $i \in [n]$. We emphasize that in our analysis $\boldsymbol{W}$ does not depend on $\tilde{\boldsymbol{X}}$; for notational simplicity, in this subsection we omit the accent in $\tilde{\boldsymbol{X}}$.

We establish the Gaussian equivalence property (Theorem 3) for kernel regression with respect to certain trained feature map under general convex loss $\ell$ satisfying Assumption (A.4) in [HL20]. Consider the estimators obtained from $\ell_2$-regularized empirical risk minimization:

$$\hat{\boldsymbol{a}} \triangleq \arg\min_{\boldsymbol{a}} \left\{ \frac{1}{n} \sum_{i=1}^n \ell(y_i, \langle \boldsymbol{a}, \boldsymbol{\phi}_i \rangle) + \frac{\lambda}{N} \|\boldsymbol{a}\|_2^2 \right\}, \tag{C.1}$$

$$\bar{\boldsymbol{a}} \triangleq \arg\min_{\boldsymbol{a}} \left\{ \frac{1}{n} \sum_{i=1}^n \ell(y_i, \langle \boldsymbol{a}, \bar{\boldsymbol{\phi}}_i \rangle) + \frac{\lambda}{N} \|\boldsymbol{a}\|_2^2 \right\}, \tag{C.2}$$

where we abbreviated the feature vector $\boldsymbol{\phi}_i = \boldsymbol{\phi}_{\boldsymbol{x}_i} = \frac{1}{\sqrt{N}} \sigma(\boldsymbol{W}^\top \boldsymbol{x}_i), \bar{\boldsymbol{\phi}}_i = \bar{\boldsymbol{\phi}}_{\boldsymbol{x}_i} = \frac{1}{\sqrt{N}} \left( \mu_1 \boldsymbol{W}^\top \boldsymbol{x}_i + \mu_2 \boldsymbol{z}_i \right)$ for $i \in [n]$.

In our setting, the first-layer weight $\boldsymbol{W}$ is no longer the initialized random matrix $\boldsymbol{W}_0$. However, we can still write the weight matrix as a perturbed version of $\boldsymbol{W}_0$, i.e., $\boldsymbol{W} = \boldsymbol{W}_0 + \boldsymbol{\Delta}$, where $\boldsymbol{\Delta} \in \mathbb{R}^{d \times N}$ corresponds to the update to the weights that is *independent of* the training data $\boldsymbol{X}$ for ridge regression (e.g., the weight matrix and the ridge regression estimator are trained on separate data). We aim to show that under suitable conditions on $\boldsymbol{\Delta}$, the Gaussian equivalence theorem holds for the kernel model defined by the perturbed features $\boldsymbol{x} \to \frac{1}{\sqrt{N}} \sigma(\boldsymbol{x}^\top \boldsymbol{W})$.

Define the set of weight matrices perturbed from the Gaussian initialization $\boldsymbol{W}_0$ as

$$\mathcal{W} := \left\{ \boldsymbol{W} = \boldsymbol{W}_0 + \boldsymbol{\Delta} \in \mathbb{R}^{d \times N} : \|\boldsymbol{\Delta}\| = \mathcal{O}(1), \|\boldsymbol{\Delta}\|_{2,\infty} = \mathcal{O}\left( \frac{\text{polylog}\, d}{\sqrt{d}} \right) \right\}. \tag{C.3}$$

Note that for learning rate $\eta = \Theta(1)$, we can verify that $\mathcal{W}$ is a high-probability event after one gradient step, as characterized in Lemma 10. The following proposition is a reformulation and extension of [HL20, Theorem 1], stating that the Gaussian equivalence property holds as long as $\boldsymbol{W}$ remains "close" to the initialization $\boldsymbol{W}_0$.

**Proposition 13.** *Under Assumption 1, and $\mathbb{P}(\mathcal{W}) \geq 1 - \exp\left(-c \log^2 N\right)$ for some $c > 0$, we have that as $n, d, N \to \infty$ proportionally,*

$$\mathbb{E}_{\boldsymbol{x}}\big(f^*(\boldsymbol{x}) - \langle \boldsymbol{\phi}_{\boldsymbol{x}}, \hat{\boldsymbol{a}} \rangle\big)^2 = (1 + o_{d,\mathbb{P}}(1)) \cdot \mathbb{E}_{\boldsymbol{x}}\big(f^*(\boldsymbol{x}) - \langle \bar{\boldsymbol{\phi}}_{\boldsymbol{x}}, \bar{\boldsymbol{a}} \rangle\big)^2,$$

*where $\hat{\boldsymbol{a}}$ and $\bar{\boldsymbol{a}}$ are defined in (C.1) and (C.2).*

From Proposition 13 we know that Theorem 3 holds if the optimized weight matrix $\boldsymbol{W}$ falls into the set $\mathcal{W}$ with sufficiently high probability. This condition is in turn verified by Lemma 10. Also note that in our setting of MSE loss and $\lambda > 0$, the RHS of the above equation is bounded in probability.

**Central limit theorem for trained features.** Recall the single-index teacher assumption: $y_i = \sigma^*(\langle \boldsymbol{x}_i, \boldsymbol{\beta}^* \rangle) + \varepsilon_i$ for $i \in [n]$. Observe that for $\boldsymbol{W} \in \mathcal{W}$, the following near-orthogonality condition between the neurons holds with high probability

$$\|\boldsymbol{W}\| = \mathcal{O}(1), \quad \text{and} \quad \max_{i \neq j} \{\langle \boldsymbol{w}_i, \boldsymbol{w}_j \rangle, \langle \boldsymbol{w}_i, \boldsymbol{\beta}_* \rangle\} = \mathcal{O}\left( \frac{\text{polylog}\, d}{\sqrt{d}} \right). \tag{C.4}$$

Importantly, for $\boldsymbol{W}$ satisfying the near-orthogonality condition (C.4), we can utilize the following central limit theorem from [HL20] derived via Stein's method.

**Proposition 14** (Theorem 2 in [HL20]). *Given Assumption 1, suppose that the activation $\sigma$ is an odd function. Let $\{\varphi_d(x; y)\}$ be a sequence of two-dimensional test functions, where $|\varphi_d(x; y)|, |\varphi'_d(x; y)| \leq B_d(y)(1 + |x|)^K$ for some function $B_d$ and constant $K \geq 1$, then for $\boldsymbol{W}$ satisfying (C.4), and fixed vectors $\boldsymbol{\alpha} \in \mathbb{R}^N, \boldsymbol{\beta} \in \mathbb{R}^d$ with $\|\boldsymbol{\beta}\| = 1$, we have*

$$\left| \mathbb{E}\varphi_d\left( \boldsymbol{\phi}_{\boldsymbol{x}}^\top \boldsymbol{\alpha}; \boldsymbol{x}^\top \boldsymbol{\beta} \right) - \mathbb{E}\varphi_d\left( \bar{\boldsymbol{\phi}}_{\boldsymbol{x}}^\top \boldsymbol{\alpha}; \boldsymbol{x}^\top \boldsymbol{\beta} \right) \right|$$

$$= \mathcal{O}\left( \frac{\text{polylog}\, N}{\sqrt{N}} \mathbb{E}[B_d(z)^4]^{1/4} \left( 1 + \|\boldsymbol{\alpha}\|_\infty^2 + \left( \frac{1}{\sqrt{N}} \|\boldsymbol{\alpha}\| \right)^{K'} \right) \right),$$

*where $z \sim \mathcal{N}(0, 1)$, and $K'$ only depends on constant $K$.*

We remark that our assumption of odd activation in Theorem 3 is required by the above Proposition 14, and we believe it could be removed with some extra work. Also, to verify the GET, we take $\varphi_d$ to be the test function defined in [HL20, Equation (50)]. In our case, by [HL20, Lemma 25] we know that there exists a function $B$ satisfying the growth condition such that $\mathbb{E}[B(z)^4]$ is bounded. Therefore, in order to apply Proposition 14 and obtain the Gaussian equivalence theorem (see derivation in [HL20, Section 2]), we only need to control the $\ell_2$-norm and $\ell_\infty$-norm of certain vector $\boldsymbol{\alpha}$ of interest. The following subsection establishes the required norm bound.

**Norm control along the interpolation path.** Following [HL20], we construct an interpolating sequence between the nonlinear and linear features model. For any $0 \leq k \leq n$, we define

$$\boldsymbol{g}_k^* \triangleq \arg\min_{\boldsymbol{g} \in \mathbb{R}^N} \left\{ \sum_{i=1}^{k} \ell(y_i, \langle \boldsymbol{g}, \bar{\boldsymbol{\phi}}_i \rangle) + \sum_{j=k+1}^{n} \ell(y_j, \langle \boldsymbol{g}, \boldsymbol{\phi}_j \rangle) + \frac{n}{N} \Big( \lambda \|\boldsymbol{g}\|_2^2 + Q(\boldsymbol{g}) \Big) \right\}, \quad \text{(C.5)}$$

where we introduce a perturbation term

$$Q(\boldsymbol{g}) \triangleq \gamma_1 \boldsymbol{g}^\top \Big( \mu_1^2 \boldsymbol{W}^\top \boldsymbol{W} + \mu_2^2 \boldsymbol{I} \Big) \boldsymbol{g} + \gamma_2 \mu_1 \sqrt{N} \boldsymbol{\beta}_*^\top \boldsymbol{W} \boldsymbol{g}.$$

Note that when $\gamma_1 = \gamma_2 = 0$, setting $k = 0$ recovers the estimator on nonlinear features $\hat{\boldsymbol{a}}$, and similarly, setting $k = n$ gives the estimator on the linear Gaussian features $\bar{\boldsymbol{a}}$.

We remark that the perturbation $Q(\boldsymbol{g})$ allows us to compute the prediction risk by taking the derivative of the objective w.r.t. $\gamma_1, \gamma_2$ around 0 — see [HL20, Proposition 1] for details. Note that when $\|\boldsymbol{W}\| = \Theta(1)$, we may choose $\gamma^* = \frac{N}{n} \cdot \frac{\lambda/4}{\mu_1^2 \|\boldsymbol{W}\|^2 + \mu_2^2} > 0$ such that for $|\gamma_1| \leq \gamma^*, |\gamma_2| \leq 1$, the overall objective (C.5) is $\frac{\lambda}{2}$-strongly convex (i.e., the strongly-convex regularizer dominates the concave part of $Q(\boldsymbol{g})$ when $\gamma_1 < 0$).

While most of the statements in [HL20] hold for deterministic weight matrices satisfying (C.4), the $\ell_\infty$-norm bound relies on the (sub-)Gaussian property of $\boldsymbol{W}$ and thus only applies to RF models. The following lemma establishes a high probability upper bound on the $\ell_\infty$-norm of $\boldsymbol{g}_k^*$ on our trained feature map.

**Lemma 15.** *Given Assumption 1, if we further assume that $1 - \mathbb{P}(\mathcal{W}) \leq \exp\big(-c \log^2 N\big)$ for some constant $c > 0$, then there exists some constant $c' > 0$ such that for any $0 \leq k \leq n$,*

$$\mathbb{P}(\|\boldsymbol{g}_k^*\|_\infty \geq \text{polylog} N) \leq \exp\big(-c' \log^2 N\big).$$

**Proof.** We follow the proof of [HL20, Lemma 23] and first analyze one coordinate of $\boldsymbol{g}_k^*$ defined by (C.5), which WLOG we select to be the last coordinate. For concise notation, we instead augment the weight matrix with an $(N+1)$-th column and study the corresponding $[\boldsymbol{g}_k^*]_{N+1}$. Denote the weight vector $\boldsymbol{w}_{N+1} = \boldsymbol{w}_{N+1}^0 + \boldsymbol{\delta}_{N+1}$, where $\boldsymbol{w}_{N+1}^0$ is the $(N+1)$-th column of the initialized $\boldsymbol{W}_0$, and $\boldsymbol{\delta}$ is the perturbation (i.e., gradient update for $\boldsymbol{W}$).

To further simplify the notation, we define $\boldsymbol{r}_i \in \mathbb{R}^N$, where $\boldsymbol{r}_i = \frac{1}{\sqrt{N}} \Big( \mu_1 \boldsymbol{W}^\top \boldsymbol{x}_i + \mu_2 \boldsymbol{z}_i \Big)$, $\boldsymbol{z}_i \overset{\text{i.i.d.}}{\sim} \mathcal{N}(0, \boldsymbol{I})$ for $i \leq k$, and $\boldsymbol{r}_i = \frac{1}{\sqrt{N}} \sigma(\boldsymbol{W}^\top \boldsymbol{x}_i)$ for $k < i \leq n$. Recall that $\boldsymbol{W} = \boldsymbol{W}_0 + \boldsymbol{\Delta}$, in which the initialization $[\boldsymbol{W}_0]_{i,j} = \mathcal{N}(0, d^{-1})$; we denote the $i$-th feature vector at initialization $\boldsymbol{W}_0$ by $\boldsymbol{r}_i^0$. Let $\boldsymbol{f} \in \mathbb{R}^n$ be the feature vector at the last coordinate, i.e., $f_i = [\boldsymbol{f}]_i = \frac{1}{\sqrt{N}} \big( \mu_1 \boldsymbol{x}_i^\top \boldsymbol{w}_{N+1} + \mu_2 z_i \big)$, $z_i \overset{\text{i.i.d.}}{\sim} \mathcal{N}(0, 1)$ for $i \leq k$, and $f_i = [\boldsymbol{f}]_i = \frac{1}{\sqrt{N}} \sigma(\boldsymbol{x}_i^\top \boldsymbol{w}_{N+1})$ for $k < i \leq n$; similarly, we introduce a superscript in $\boldsymbol{f}^0 \in \mathbb{R}^n$ to denote the features produced by the initial $\boldsymbol{w}_{N+1}^0$.

The $(N+1)$-th coordinate of interest, which we denote as $u^*$, can be written as the solution to the following optimization problem,

$$u^* = \underset{u}{\arg\min} \min_{\boldsymbol{g}} \sum_{i=1}^{n} \ell\big(\boldsymbol{r}_i^\top \boldsymbol{g} + f_i u; y_t\big) + \frac{n}{N} \Big( \lambda \|\boldsymbol{g}\|^2 + Q(\boldsymbol{g}) + \lambda u^2 + q(u) + \big(2\gamma_1 \mu_1^2 \boldsymbol{w}_{N+1}^\top \boldsymbol{W} \boldsymbol{g}\big) u \Big),$$

where we defined $q(u) = \gamma_1\left(\mu_1^2\|\boldsymbol{w}_{N+1}\|^2 + \mu_2^2\right)u^2 + \gamma_2\left(\mu_1\sqrt{N}\boldsymbol{\beta}_*^\top\boldsymbol{w}_{N+1}\right)u$. By [HL20, Equation (249)], we know that for $\boldsymbol{W} \in \mathcal{W}$,

$$|u^*| \lesssim \frac{1}{\lambda}\left|2\gamma_1\mu_1^2\boldsymbol{w}_{N+1}^\top\boldsymbol{W}\boldsymbol{g}_k^* + \gamma_2\mu_1\sqrt{N}\boldsymbol{\beta}_*^\top\boldsymbol{w}_{N+1} + \sum_{i=1}^n \ell'\left(\boldsymbol{r}_i^\top\boldsymbol{g}_k^*; y_i\right)f_i\right|. \tag{C.6}$$

We control each term on the right hand side of (C.6) separately. Note that $\boldsymbol{W} \in \mathcal{W}$ implies that $\|\boldsymbol{\delta}_{N+1}\| = \mathcal{O}\left(\frac{\text{polylog}d}{\sqrt{d}}\right)$ due to the definition (C.3). Since $\left|\boldsymbol{\beta}_*^\top\boldsymbol{w}_{N+1}\right| \leq \left|\boldsymbol{\beta}_*^\top\boldsymbol{w}_{N+1}^0\right| + \|\boldsymbol{\delta}_{N+1}\|\|\boldsymbol{\beta}_*\|$, by combining [HL20, Equation (252)] and our assumption that $\|\boldsymbol{\beta}_*\| = 1$, we know that for some constant $c_1 > 0$ and large $N$,

$$\mathbb{P}\left(\left|\sqrt{N}\boldsymbol{\beta}_*^\top\boldsymbol{w}_{N+1}\right| \geq \text{polylog}N\right) \leq \exp\left(-c_1\log^2 N\right).$$

Similarly, $\left|\boldsymbol{w}_{N+1}^\top\boldsymbol{W}\boldsymbol{g}_k^*\right| \leq \left|\boldsymbol{w}_{N+1}^{0\top}\boldsymbol{W}\boldsymbol{g}_k^*\right| + \|\boldsymbol{\delta}_{N+1}\|\|\boldsymbol{W}\boldsymbol{g}_k^*\|$, and therefore by [HL20, Lemma 17] (note that the lemma only requires $\boldsymbol{W}$ to satisfy (C.4)), we have

$$\mathbb{P}\left(\left|\boldsymbol{w}_{N+1}^\top\boldsymbol{W}\boldsymbol{g}_k^*\right| \geq \text{polylog}N\right) \leq \exp\left(-c_2\log^2 N\right),$$

for constant $c_2 > 0$. To control the sum of $\ell'$ in (C.6), for simplicity we define $\boldsymbol{\theta}^* \in \mathbb{R}^n$, where $\theta_i^* = [\boldsymbol{\theta}^*]_i = \ell'(\boldsymbol{r}_i^\top\boldsymbol{g}_k^*; y_i)$ for $i \in [n]$. Notice that $\left|\boldsymbol{x}_i^\top\boldsymbol{w}_j - \boldsymbol{x}_i^\top\boldsymbol{w}_j^0\right| = \left|\boldsymbol{x}_i^\top\boldsymbol{\delta}_j\right|$. Due to the assumed independence between $\boldsymbol{X}$ and $\boldsymbol{\Delta}$, and the assumption on $\mathbb{P}(\mathcal{W})$, we know that $\left|\boldsymbol{x}_i^\top\boldsymbol{\delta}_j\right| \lesssim \|\boldsymbol{\delta}_j\| \cdot \log N = \mathcal{O}\left(\frac{\text{polylog}N}{\sqrt{N}}\right)$ with high probability. In addition, since the activation function $\sigma$ is Lipschitz, for $k < i \leq n$, we may take a union bound over the weight vectors $\boldsymbol{w}_j$ and obtain

$$\mathbb{P}\left(\sqrt{N}\|\boldsymbol{r}_i - \boldsymbol{r}_i^0\| \geq \text{polylog}N\right) \leq N \cdot \exp\left(-c_3\log^2 N\right), \tag{C.7}$$

for some $c_3 > 0$. The case where $i \leq k$ (i.e., the features are linear) follows from the exact same argument. Also, because of $\left|\boldsymbol{r}_i^\top\boldsymbol{g}_k^*\right| \leq \left|\boldsymbol{r}_i^{0\top}\boldsymbol{g}_k^*\right| + \|\boldsymbol{r}_i - \boldsymbol{r}_i^0\|\|\boldsymbol{g}_k^*\|$, we know that [HL20, Equation (257)], [HL20, Lemma 17], and (C.7) together ensure that

$$\mathbb{P}(|\theta_i^*| \geq \text{polylog}N) \leq \exp\left(-c_4\log^2 N\right), \tag{C.8}$$

for some constant $c_4 > 0$ and large enough $N$. Now we can control $\left|\sum_{i=1}^n \ell'\left(\boldsymbol{r}_i^\top\boldsymbol{g}_k^*; y_i\right)f_i\right|$ in (C.6). Again using the Lipschitz property of activation $\sigma$, we get

$$\left|\sum_{i=1}^n \ell'\left(\boldsymbol{r}_i^\top\boldsymbol{g}_k^*; y_i\right)(f_i - f_i^0 + f_i^0)\right| \leq \left|\sum_{i=1}^n \ell'\left(\boldsymbol{r}_i^\top\boldsymbol{g}_k^*; y_i\right)f_i^0\right| + \left|\sum_{i=1}^n \ell'\left(\boldsymbol{r}_i^\top\boldsymbol{g}_k^*; y_i\right)\left(f_i - f_i^0\right)\right|$$

$$\lesssim \left|\sum_{i=1}^n \theta_i^* f_i^0\right| + \frac{1}{\sqrt{N}}\sum_{i=1}^n |\theta_i^*| \cdot \left|\boldsymbol{x}_i^\top\boldsymbol{\delta}_{N+1}\right|.$$

Given (C.8) (which implies that $\frac{1}{\sqrt{N}}\|\boldsymbol{\theta}^*\| = \mathcal{O}(\text{polylog}N)$ with high probability), it has been shown in [HL20, Proof of Lemma 23] that $\mathbb{P}\left(\left|\sum_{i=1}^n \theta_i^* f_i^0\right| \geq \text{polylog}N\right) \leq \exp\left(-c_5\log^2 N\right)$ for some constant $c_5 > 0$. Hence, by taking union bound over the failure events $|\theta_i^*| \geq \text{polylog}N$ and $\sqrt{N} \cdot \left|\boldsymbol{x}_i^\top\boldsymbol{\delta}_{N+1}\right| \geq \text{polylog}N$, we arrive at the following high probability upper bound on $u^*$ in terms of (C.6):

$$\mathbb{P}(|u^*| \geq \text{polylog}N) \leq n^2 N \cdot \exp\left(-c_6\log^2 N\right),$$

for some constant $c_6 > 0$ and all large $N$. Finally, since the assumption on $\|\boldsymbol{\Delta}\|_{2,\infty}$ implies control of $\|\boldsymbol{\delta}_i\|$ for all $i \in [N]$, we complete the proof by a union bound over the $N$ coordinates. $\square$

**Putting things together.** Denote the optimal value of objective (C.5) by

$$R_k^* \triangleq \min_{\boldsymbol{g}\in\mathbb{R}^N}\left\{\frac{1}{n}\sum_{i=1}^k \ell(y_i, \langle\boldsymbol{g}, \bar{\boldsymbol{\phi}}_i\rangle) + \frac{1}{n}\sum_{j=k+1}^n \ell(y_j, \langle\boldsymbol{g}, \boldsymbol{\phi}_j\rangle) + \frac{1}{N}\left(\lambda\|\boldsymbol{g}\|_2^2 + Q(\boldsymbol{g})\right)\right\}$$

From [HL20, Section 2.3], we know that Proposition 14 and Lemma 15 imply that for any $\boldsymbol{W} \in \mathcal{W}$ and $1 \leq k \leq n$, the discrepancy due to one swap can be bounded as

$$\left| \mathbb{E}[\psi(R_k^*)] - \mathbb{E}[\psi(R_{k-1}^*)] \right| = \mathcal{O}\left( \frac{\text{polylog}N}{N^{3/2}} \right),$$

for bounded test function $\psi$ with bounded first and second derivatives. As there are $n = \Theta(N)$ total swaps to be made, we can obtain the desired Gaussian equivalence ([HL20, Theorem 1]) if the failure probability $(1 - \mathbb{P}(\mathcal{W}))$ is sufficiently small. Hence we conclude Proposition 13.

**Proof of Theorem 3.** Finally, we establish Theorem 3 by verifying that in our setting the event $\mathcal{W}$ occurs with high probability. For one gradient step on the squared loss with learning rate $\eta = \Theta(1)$, Lemma 10 together with $\|\boldsymbol{\beta}_*\| = 1$ entail that for proportional $n, d, N$, there exists some constant $c, C > 0$ such that

$$\mathbb{P}(\|\boldsymbol{W}_1\| \geq C) \leq \exp(-cd),$$

$$\mathbb{P}\left( \max_{i \neq j} |\langle \boldsymbol{w}_i^1, \boldsymbol{w}_j^1 \rangle| \geq \frac{C \log^2 d}{\sqrt{d}} \right) \leq \exp(-c \log^2 d),$$

$$\mathbb{P}\left( \max_i |\langle \boldsymbol{w}_i^1, \boldsymbol{\beta}_* \rangle| \geq \frac{C \log^2 d}{\sqrt{d}} \right) \leq \exp(-c \log^2 d),$$

where $\boldsymbol{w}_i^1$ stands for the $i$-th column of $\boldsymbol{W}_1$ for $i \in [N]$. In addition, under Assumption 1, when $\lambda > 0$, it is straightforward to verify that prediction risk of the Gaussian equivalent model $\mathcal{R}_{\text{GE}}(\lambda) \xrightarrow{\mathbb{P}} C_\lambda$ for some finite constant $C_\lambda > 0$ as $n, N, d \to \infty$ proportionally. Theorem 3 therefore follows from Proposition 13 (or equivalently, Equation (16) in [HL20, Theorem 1]). $\square$

## C.2 Prediction risk under Gaussian equivalence

Now we compute the prediction risk of the CK ridge estimator on the feature map after one gradient step $\boldsymbol{x} \to \sigma(\boldsymbol{W}_1^\top \boldsymbol{x})$. Recall the closed-form solution of the ridge regression estimator:

$$\hat{\boldsymbol{a}} = \arg\min_{\boldsymbol{a}} = \left( \boldsymbol{\Phi}^\top \boldsymbol{\Phi} + \frac{\lambda n}{N} \boldsymbol{I} \right)^{-1} \boldsymbol{\Phi}^\top \tilde{\boldsymbol{y}},$$

where $\boldsymbol{\Phi} = \frac{1}{\sqrt{N}} \sigma(\tilde{\boldsymbol{X}} \boldsymbol{W}_1) \in \mathbb{R}^{n \times N}$, $\tilde{\boldsymbol{X}} \in \mathbb{R}^{n \times d}$ denotes a new batch of training data independent of $\boldsymbol{W}_1$, and $\tilde{\boldsymbol{y}} = f^*(\tilde{\boldsymbol{X}}) + \tilde{\boldsymbol{\varepsilon}} \in \mathbb{R}^n$ is the corresponding training labels (following the same Assumption 1). Also, recall the following Gaussian covariates model:

$$\bar{\boldsymbol{\Phi}} \triangleq \frac{1}{\sqrt{N}} \left( \mu_1 \tilde{\boldsymbol{X}} \boldsymbol{W}_1 + \mu_2 \boldsymbol{Z} \right) \in \mathbb{R}^{n \times N}; \quad \bar{\boldsymbol{a}} \triangleq \left( \bar{\boldsymbol{\Phi}}^\top \bar{\boldsymbol{\Phi}} + \frac{\lambda n}{N} \boldsymbol{I} \right)^{-1} \bar{\boldsymbol{\Phi}}^\top \tilde{\boldsymbol{y}}.$$

where $[\boldsymbol{Z}]_{ij} \sim \mathcal{N}(0,1)$ independent of $\tilde{\boldsymbol{X}}$ and $\boldsymbol{W}_1$. Due to the Gaussian equivalence property (4.2), we can analyze the prediction risk of the Gaussian covariates model, which we denote as $\mathcal{R}_{\text{GE}}(\lambda)$.

**Bias-variance decomposition.** The following lemma simplifies the prediction risk $\mathcal{R}_{\text{GE}}(\lambda)$ and separates the bias (due to learning the teacher $f^*$) and variance (due to the label noise $\tilde{\boldsymbol{\varepsilon}}$).

**Lemma 16.** *Under Assumption 1, we have*

$$\mathcal{R}_{\text{GE}}(\lambda) - (B_1 + B_2 + V) \xrightarrow{\mathbb{P}} 0,$$

*where the bias and variance terms are given as*

$$B_1 = \mu_1^{*2} + \mu_2^{*2} - \frac{2\mu_1 \mu_1^*}{\sqrt{N}} \boldsymbol{\beta}_*^\top \boldsymbol{W}_1 \left( \widehat{\boldsymbol{\Sigma}}_\Phi + \tilde{\lambda} \boldsymbol{I} \right)^{-1} \bar{\boldsymbol{\Phi}}^\top \boldsymbol{f}^*. \tag{C.9}$$

$$B_2 = \boldsymbol{f}^{*\top} \bar{\boldsymbol{\Phi}} \left( \widehat{\boldsymbol{\Sigma}}_\Phi + \tilde{\lambda} \boldsymbol{I} \right)^{-1} \overline{\boldsymbol{\Sigma}}_\Phi \left( \widehat{\boldsymbol{\Sigma}}_\Phi + \tilde{\lambda} \boldsymbol{I} \right)^{-1} \bar{\boldsymbol{\Phi}}^\top \boldsymbol{f}^*. \tag{C.10}$$

$$V = \sigma_{\tilde{\varepsilon}}^2 \text{Tr}\left( \left( \widehat{\boldsymbol{\Sigma}}_\Phi + \tilde{\lambda} \boldsymbol{I} \right)^{-1} \widehat{\boldsymbol{\Sigma}}_\Phi \left( \widehat{\boldsymbol{\Sigma}}_\Phi + \tilde{\lambda} \boldsymbol{I} \right)^{-1} \overline{\boldsymbol{\Sigma}}_\Phi \right). \tag{C.11}$$

*and we defined* $\tilde{\lambda} = \frac{\lambda n}{N}, \widehat{\boldsymbol{\Sigma}}_\Phi = \bar{\boldsymbol{\Phi}}^\top \bar{\boldsymbol{\Phi}}, \overline{\boldsymbol{\Sigma}}_\Phi = \frac{1}{N}\left( \mu_1^2 \boldsymbol{W}_1^\top \boldsymbol{W}_1 + \mu_2^2 \boldsymbol{I} \right)$, *and* $[\boldsymbol{f}^*]_i = f^*(\tilde{\boldsymbol{x}}_i)$.

**Proof.** First note that $\mathcal{R}_{\mathrm{GE}}$ is given by [HL20, Equation (57)]:

$$\mathcal{R}_{\mathrm{GE}} = \mathbb{E}_{\boldsymbol{x}}\left(\sigma^*(\boldsymbol{x}^\top\boldsymbol{\beta}_*) - \bar{\boldsymbol{\phi}}_{\boldsymbol{x}}^\top\bar{\boldsymbol{a}}\right)^2 \tag{C.12}$$

$$= \mathbb{E}_{z_1,z_2}\left(\sigma^*(z_1) - \frac{\mu_1}{\sqrt{N}}\boldsymbol{\beta}_*^\top\boldsymbol{W}_1\bar{\boldsymbol{a}}\cdot z_1 + \sqrt{\frac{1}{N}\bar{\boldsymbol{a}}^\top(\mu_1^2\boldsymbol{W}_1^\top\boldsymbol{W}_1 + \mu_2^2\boldsymbol{I} - \mu_1^2\boldsymbol{W}_1^\top\boldsymbol{\beta}_*\boldsymbol{\beta}_*^\top\boldsymbol{W}_1)\bar{\boldsymbol{a}}}\cdot z_2\right)^2$$

where $z_1, z_2 \stackrel{\text{i.i.d.}}{\sim} \mathcal{N}(0,1)$. Because of the independence between $z_1, z_2$, we only need to show the following as $n, d, N \to \infty$ proportionally:

$$\frac{1}{\sqrt{N}}\boldsymbol{\beta}_*^\top\boldsymbol{W}_1\left(\widehat{\boldsymbol{\Sigma}}_\Phi + \tilde{\lambda}\boldsymbol{I}\right)^{-1}\bar{\boldsymbol{\Phi}}^\top\tilde{\boldsymbol{\varepsilon}} \stackrel{\mathbb{P}}{\to} 0,$$

$$\boldsymbol{f}^{*\top}\bar{\boldsymbol{\Phi}}\left(\widehat{\boldsymbol{\Sigma}}_\Phi + \tilde{\lambda}\boldsymbol{I}\right)^{-1}\overline{\boldsymbol{\Sigma}}_\Phi\left(\widehat{\boldsymbol{\Sigma}}_\Phi + \tilde{\lambda}\boldsymbol{I}\right)^{-1}\bar{\boldsymbol{\Phi}}^\top\tilde{\boldsymbol{\varepsilon}} \stackrel{\mathbb{P}}{\to} 0.$$

Both equations follow from the general Hoeffding inequality for $\tilde{\varepsilon}$ (e.g., see Theorem 2.6.3 [Ver18]) since both $\left\|\boldsymbol{\beta}_*^\top\boldsymbol{W}_1\left(\widehat{\boldsymbol{\Sigma}}_\Phi + \tilde{\lambda}\boldsymbol{I}\right)^{-1}\bar{\boldsymbol{\Phi}}^\top\right\|$ and $\left\|\sqrt{N}\cdot\boldsymbol{f}^{*\top}\bar{\boldsymbol{\Phi}}\left(\widehat{\boldsymbol{\Sigma}}_\Phi + \tilde{\lambda}\boldsymbol{I}\right)^{-1}\overline{\boldsymbol{\Sigma}}_\Phi\left(\widehat{\boldsymbol{\Sigma}}_\Phi + \tilde{\lambda}\boldsymbol{I}\right)^{-1}\bar{\boldsymbol{\Phi}}^\top\right\|$ are bounded by some constant with high probability when $\lambda > 0$.

□

Also, the risk lower bound for the Gaussian equivalent model is a direct consequence of (C.12).

**Proof of Fact 4.** Under Assumption 1, we may write $\sigma^*(z) = \mu_1^* z + \sigma_\perp^*(z)$, where $\mathbb{E}_z[z\sigma_\perp^*(z)] = 0, \mathbb{E}_z[\sigma_\perp^*(z)^2] = \mu_2^{*2}$ for $z \sim \mathcal{N}(0,1)$. Hence from (C.12) we know that

$$\mathcal{R}_{\mathrm{GE}} \geq \mathbb{E}_{z_1}\left(\sigma^*(z_1) - \frac{\mu_1}{\sqrt{N}}\boldsymbol{\beta}_*^\top\boldsymbol{W}_1\bar{\boldsymbol{a}}\cdot z_1\right)^2 = \left(\mu_1^* - \frac{\mu_1}{\sqrt{N}}\boldsymbol{\beta}_*^\top\boldsymbol{W}_1\bar{\boldsymbol{a}}\right)^2 + \mu_2^{*2}.$$

This implies that $\mathcal{R}_{\mathrm{GE}} \geq \|\mathsf{P}_{>1}f^*\|_{L^2}^2 = \mu_2^{*2}$ with probability one as $d \to \infty$. □

In the following sections, we compare the bias and variance terms given in (C.9), (C.10) and (C.11) before and after one feature learning step. We first simplify the calculation by showing that the values of these equations remain asymptotically unchanged if we remove certain low-order terms.

**Stability of the bias and variance.** We now control the errors in the bias and variance terms after ignoring the lower-order terms in the weight matrix.

Recall that $\boldsymbol{W}_1 = \boldsymbol{W}_0 + \eta\sqrt{N}\boldsymbol{G}_0$; we introduce $\tilde{\boldsymbol{W}} := \boldsymbol{W}_0 + \eta\sqrt{N}\boldsymbol{A}$, in which we ignored the terms $\boldsymbol{B}$ and $\boldsymbol{C}$ in the gradient matrix (B.1). We also denote the corresponding CK features and kernel matrix as $\tilde{\boldsymbol{\Phi}} := \frac{1}{\sqrt{N}}\left(\mu_1\tilde{\boldsymbol{X}}\tilde{\boldsymbol{W}} + \mu_2\boldsymbol{Z}\right)$, $\tilde{\boldsymbol{\Sigma}}_\Phi := \tilde{\boldsymbol{\Phi}}^\top\tilde{\boldsymbol{\Phi}}$, and the bias terms as $\tilde{B}_1, \tilde{B}_2$ (parallel to (C.9) and (C.10)). Finally, we write the initial random feature matrix as $\bar{\boldsymbol{\Phi}}_0 := \frac{1}{\sqrt{N}}\left(\mu_1\tilde{\boldsymbol{X}}\boldsymbol{W}_0 + \mu_2\boldsymbol{Z}\right)$, $\widehat{\boldsymbol{\Sigma}}_{\Phi_0} := \bar{\boldsymbol{\Phi}}_0^\top\bar{\boldsymbol{\Phi}}_0$, and refer to the variance of the initialized RF ridge estimator as $V_0$.

**Lemma 17.** *Given Assumption 1 and $\lambda > 0$. Then for $\eta = \Theta(1)$, we have*

$$|B_1 - \tilde{B}_1| = o_{d,\mathbb{P}}(1), \ |B_2 - \tilde{B}_2| = o_{d,\mathbb{P}}(1), \ |V - V_0| = o_{d,\mathbb{P}}(1).$$

**Proof.** To start with, recall that the operator norms of all matrices $\boldsymbol{W}_1, \boldsymbol{W}_0, \tilde{\boldsymbol{W}}, \bar{\boldsymbol{\Phi}}, \bar{\boldsymbol{\Phi}}_0$ and $\tilde{\boldsymbol{\Phi}}$ are uniformly bounded by some constants with high probability. We first consider the change in Frobenius norm of first-layer $\boldsymbol{W}$ to analyze the difference between $V$ and $V_0$. By Lemma 10, standard calculation yields:

$$\left\|\boldsymbol{W}_1^\top\boldsymbol{W}_1 - \boldsymbol{W}_0^\top\boldsymbol{W}_0\right\|_F = \mathcal{O}_{d,\mathbb{P}}(1); \quad \left\|\bar{\boldsymbol{\Phi}} - \bar{\boldsymbol{\Phi}}_0\right\|_F = \mathcal{O}_{d,\mathbb{P}}(1); \quad \left\|\widehat{\boldsymbol{\Sigma}}_\Phi - \widehat{\boldsymbol{\Sigma}}_{\Phi_0}\right\|_F = \mathcal{O}_{d,\mathbb{P}}(1).$$

Utilizing the above estimates, we obtain

$$\left\|\left(\widehat{\boldsymbol{\Sigma}}_\Phi + \tilde{\lambda}\boldsymbol{I}\right)^{-1}\bar{\boldsymbol{\Phi}}^\top - \left(\widehat{\boldsymbol{\Sigma}}_{\Phi_0} + \tilde{\lambda}\boldsymbol{I}\right)^{-1}\bar{\boldsymbol{\Phi}}_0^\top\right\|_F$$

$$\leq \left\|\bar{\boldsymbol{\Phi}} - \bar{\boldsymbol{\Phi}}_0\right\|_F \left\|\left(\widehat{\boldsymbol{\Sigma}}_\Phi + \tilde{\lambda}\boldsymbol{I}\right)^{-1}\right\| + \|\bar{\boldsymbol{\Phi}}_0\| \left\|\left(\widehat{\boldsymbol{\Sigma}}_\Phi + \tilde{\lambda}\boldsymbol{I}\right)^{-1} - \left(\widehat{\boldsymbol{\Sigma}}_{\Phi_0} + \tilde{\lambda}\boldsymbol{I}\right)^{-1}\right\|_F \stackrel{(i)}{=} \mathcal{O}_{d,\mathbb{P}}(1).$$

where $(i)$ is due to our assumption that $\lambda > 0$. Denote $\boldsymbol{M} := \left(\widehat{\boldsymbol{\Sigma}}_\Phi + \tilde{\lambda}\boldsymbol{I}\right)^{-1}\bar{\boldsymbol{\Phi}}^\top$ and likewise $\boldsymbol{M}_0 := \left(\widehat{\boldsymbol{\Sigma}}_{\Phi_0} + \tilde{\lambda}\boldsymbol{I}\right)^{-1}\bar{\boldsymbol{\Phi}}_0^\top$. Then we have

$$|V - V_0| \stackrel{(ii)}{\lesssim} \frac{1}{N}\left|\mathrm{Tr}\left(\boldsymbol{M}\boldsymbol{M}^\top\left(\mu_1^2\boldsymbol{W}_1^\top\boldsymbol{W}_1 + \mu_2^2\boldsymbol{I}\right) - \boldsymbol{M}_0\boldsymbol{M}_0^\top\left(\mu_1^2\boldsymbol{W}_0^\top\boldsymbol{W}_0 + \mu_2^2\boldsymbol{I}\right)\right)\right|$$

$$\lesssim \frac{1}{N}\left\|\boldsymbol{M}_0\boldsymbol{M}_0^\top\right\|_F \cdot \left\|\boldsymbol{W}_1^\top\boldsymbol{W}_1 - \boldsymbol{W}_0^\top\boldsymbol{W}_0\right\|_F + \frac{1}{N}\left\|\boldsymbol{M}\boldsymbol{M}^\top - \boldsymbol{M}_0\boldsymbol{M}_0^\top\right\|_F \cdot \left\|\mu_1^2\boldsymbol{W}_1^\top\boldsymbol{W}_1 + \mu_2^2\boldsymbol{I}\right\|_F$$

$$= o_{d,\mathbb{P}}(1),$$

as $n, d, N \to \infty$ at comparable rate, where we dropped the constant $\sigma_\varepsilon^2$ in $(ii)$.

For the bias terms, we consider perturbation on $\boldsymbol{W}_1$ in the operator norm. Again, Lemma 10 entails

$$\left\|\boldsymbol{W}_1^\top\boldsymbol{W}_1 - \tilde{\boldsymbol{W}}^\top\tilde{\boldsymbol{W}}\right\| = o_{d,\mathbb{P}}(1); \quad \left\|\bar{\boldsymbol{\Phi}} - \tilde{\boldsymbol{\Phi}}\right\| = o_{d,\mathbb{P}}(1); \quad \left\|\widehat{\boldsymbol{\Sigma}}_\Phi - \tilde{\boldsymbol{\Sigma}}_\Phi\right\| = o_{d,\mathbb{P}}(1).$$

Define $\tilde{\boldsymbol{M}} := \left(\tilde{\boldsymbol{\Sigma}}_\Phi + \tilde{\lambda}\boldsymbol{I}\right)^{-1}\tilde{\boldsymbol{\Phi}}^\top$. Following the same procedure, we obtain

$$\left\|\boldsymbol{M} - \tilde{\boldsymbol{M}}\right\| = \left\|\left(\widehat{\boldsymbol{\Sigma}}_\Phi + \tilde{\lambda}\boldsymbol{I}\right)^{-1}\bar{\boldsymbol{\Phi}}^\top - \left(\tilde{\boldsymbol{\Sigma}}_\Phi + \tilde{\lambda}\boldsymbol{I}\right)^{-1}\tilde{\boldsymbol{\Phi}}^\top\right\|$$

$$\leq \left\|\bar{\boldsymbol{\Phi}} - \tilde{\boldsymbol{\Phi}}\right\|\left\|\left(\widehat{\boldsymbol{\Sigma}}_\Phi + \tilde{\lambda}\boldsymbol{I}\right)^{-1}\right\| + \|\tilde{\boldsymbol{\Phi}}\|\left\|\left(\widehat{\boldsymbol{\Sigma}}_\Phi + \tilde{\lambda}\boldsymbol{I}\right)^{-1} - \left(\tilde{\boldsymbol{\Sigma}}_\Phi + \tilde{\lambda}\boldsymbol{I}\right)^{-1}\right\| = o_{d,\mathbb{P}}(1).$$

Based on this result, it is straightforward to show that

$$|B_1 - \tilde{B}_1| \lesssim \left\|\boldsymbol{W}_1 - \tilde{\boldsymbol{W}}\right\|\|\tilde{\boldsymbol{M}}\| + \|\boldsymbol{W}_1\|\left\|\boldsymbol{M} - \tilde{\boldsymbol{M}}\right\| = o_{d,\mathbb{P}}(1).$$

Similarly, for $B_2$, we have

$$|B_2 - \tilde{B}_2| \lesssim \frac{1}{N}\|\boldsymbol{f}^*\|^2 \cdot \left\|\boldsymbol{M}^\top\left(\mu_1^2\boldsymbol{W}_1^\top\boldsymbol{W}_1 + \mu_2^2\boldsymbol{I}\right)\boldsymbol{M} - \tilde{\boldsymbol{M}}^\top\left(\mu_1^2\tilde{\boldsymbol{W}}^\top\tilde{\boldsymbol{W}} + \mu_2^2\boldsymbol{I}\right)\tilde{\boldsymbol{M}}\right\|$$

$$\stackrel{(iii)}{\lesssim} \mathcal{O}_{d,\mathbb{P}}(1) \cdot \left(\left(\|\boldsymbol{M}\| + \|\tilde{\boldsymbol{M}}\|\right)\left\|\boldsymbol{W}_1^\top\boldsymbol{W}_1\right\|\left\|\boldsymbol{M} - \tilde{\boldsymbol{M}}\right\| + \|\tilde{\boldsymbol{M}}\|^2\left\|\boldsymbol{W}_1^\top\boldsymbol{W}_1 - \tilde{\boldsymbol{W}}^\top\tilde{\boldsymbol{W}}\right\|\right)$$

$$= o_{d,\mathbb{P}}(1),$$

where in $(iii)$ we used the fact that $\sigma^*$ is Lipschitz and $\|\boldsymbol{\beta}_*\| = 1$ (for example see [BMR21, Lemma A.12]). The statement is proved by combining all the above calculations. $\square$

Lemma 17 entails that the variance term in the risk does not change after one gradient step with $\eta = \Theta(1)$, and for the bias terms, we may consider the rank-1 approximation of the gradient matrix given in Proposition 2 instead. In the following, we use this property to simplify the risk expressions.

### C.3 Precise characterization of prediction risk

Now we compute the asymptotic expressions of the bias and variance terms defined in Lemma 16. As previously remarked, due to the dependence between the feature matrix $\boldsymbol{\Phi}$ and the teacher $\boldsymbol{\beta}_*$, we cannot naively employ a rotation invariance argument to simplify the calculation (as in [MM22]). Instead, based on the Gaussian equivalence property, we first make use of the Woodbury formula to separate the low-rank terms in the risk expressions. In particular, because of Lemma 10 and Lemma 17, we may simply consider the rank-one approximation of the first-step gradient: $\boldsymbol{W}_1 = \boldsymbol{W}_0 + \boldsymbol{u}\boldsymbol{a}^\top$, where $\boldsymbol{u} = \frac{\mu_1\eta}{n}\boldsymbol{X}^\top\boldsymbol{y}$ and $\boldsymbol{y} = f^*(\boldsymbol{X}) + \boldsymbol{\varepsilon}$ satisfying Assumption 1. Notice here $\boldsymbol{u}, \tilde{\boldsymbol{X}}, \boldsymbol{W}_0$ and $\boldsymbol{a}$ are mutually independent. To distinguish the terms in the CK ridge regression estimator

using the initial weights $\boldsymbol{W}_0$ and the trained weights $\boldsymbol{W}_1$, in this section we denote

$$
\begin{aligned}
\bar{\boldsymbol{\Phi}} &:= \frac{1}{\sqrt{N}}\Big(\mu_1 \tilde{\boldsymbol{X}} \boldsymbol{W}_1 + \mu_2 \boldsymbol{Z}\Big), &\quad \boldsymbol{\Phi}_0 &:= \frac{1}{\sqrt{N}}\Big(\mu_1 \tilde{\boldsymbol{X}} \boldsymbol{W}_0 + \mu_2 \boldsymbol{Z}\Big), \\
\widehat{\boldsymbol{\Sigma}}_\Phi &:= \bar{\boldsymbol{\Phi}}^\top \bar{\boldsymbol{\Phi}}, &\quad \widehat{\boldsymbol{\Sigma}}_{\Phi_0} &:= \boldsymbol{\Phi}_0^\top \boldsymbol{\Phi}_0 \in \mathbb{R}^{N \times N}, \\
\boldsymbol{R} &:= \Big(\widehat{\boldsymbol{\Sigma}}_\Phi + \tilde{\lambda}\boldsymbol{I}\Big)^{-1}, &\quad \boldsymbol{R}_0 &:= \Big(\widehat{\boldsymbol{\Sigma}}_{\Phi_0} + \tilde{\lambda}\boldsymbol{I}\Big)^{-1}, \\
\overline{\boldsymbol{\Sigma}}_\Phi &:= \frac{1}{N}\Big(\mu_1^2 \boldsymbol{W}_1^\top \boldsymbol{W}_1 + \mu_2^2 \boldsymbol{I}\Big), &\quad \overline{\boldsymbol{\Sigma}}_{\Phi_0} &:= \frac{1}{N}\Big(\mu_1^2 \boldsymbol{W}_0^\top \boldsymbol{W}_0 + \mu_2^2 \boldsymbol{I}\Big).
\end{aligned}
\tag{C.13}
$$

Also, we write $\boldsymbol{f}^* := f^*(\tilde{\boldsymbol{X}}) \in \mathbb{R}^n$, which can be decomposed into $\boldsymbol{f}^* = \mu_1^* \tilde{\boldsymbol{X}} \boldsymbol{\beta}_* + \boldsymbol{f}_{\mathrm{NL}}^*$, where $[\boldsymbol{f}_{\mathrm{NL}}^*]_i = \mathsf{P}_{>1} f^*(\tilde{\boldsymbol{x}}_i)$ (recall that $\mu_0^* = 0$ by Assumption 1). Furthermore, we introduce the following terms which will be important in the decomposition of the prediction risk:

$$
\begin{aligned}
T_1 &:= \boldsymbol{a}^\top \boldsymbol{R}_0 \boldsymbol{a}, &\quad T_2 &:= \frac{\mu_1^2}{N} \boldsymbol{u}^\top \tilde{\boldsymbol{X}}^\top \boldsymbol{\Phi}_0 \boldsymbol{R}_0 \boldsymbol{\Phi}_0^\top \tilde{\boldsymbol{X}} \boldsymbol{u}, \\
T_3 &:= \frac{\mu_1^2}{N} \boldsymbol{u}^\top \tilde{\boldsymbol{X}}^\top \tilde{\boldsymbol{X}} \boldsymbol{u}, &\quad T_4 &:= \mu_1^* \boldsymbol{\beta}_*^\top \boldsymbol{u}, \\
T_5 &:= \frac{\mu_1^2 \mu_1^*}{N} \boldsymbol{\beta}_*^\top \tilde{\boldsymbol{X}}^\top \tilde{\boldsymbol{X}} \boldsymbol{u}, &\quad \tilde{T}_5 &:= \frac{\mu_1^2}{N} \boldsymbol{f}_{\mathrm{NL}}^{*\top} \tilde{\boldsymbol{X}} \boldsymbol{u}, \\
T_6 &:= \frac{\mu_1 \mu_1^*}{2\sqrt{N}} \boldsymbol{\beta}_*^\top \Big(\boldsymbol{W}_0 \boldsymbol{R}_0 \boldsymbol{\Phi}_0^\top \tilde{\boldsymbol{X}} + \tilde{\boldsymbol{X}}^\top \boldsymbol{\Phi}_0 \boldsymbol{R}_0 \boldsymbol{W}_0^\top\Big) \boldsymbol{u}, &\quad \tilde{T}_6 &:= \frac{\mu_1}{2\sqrt{N}} \boldsymbol{f}_{\mathrm{NL}}^{*\top} \boldsymbol{\Phi}_0 \boldsymbol{R}_0 \boldsymbol{W}_0^\top \boldsymbol{u}, &\quad \text{(C.14)} \\
T_7 &:= \frac{\mu_1^2 \mu_1^*}{N} \boldsymbol{u}^\top \tilde{\boldsymbol{X}}^\top \boldsymbol{\Phi}_0 \boldsymbol{R}_0 \boldsymbol{\Phi}_0^\top \tilde{\boldsymbol{X}} \boldsymbol{\beta}_*, &\quad \tilde{T}_7 &:= \frac{\mu_1^2}{N} \boldsymbol{u}^\top \tilde{\boldsymbol{X}}^\top \boldsymbol{\Phi}_0 \boldsymbol{R}_0 \boldsymbol{\Phi}_0^\top \boldsymbol{f}_{\mathrm{NL}}^*, \\
T_8 &:= \frac{N}{\mu_1^2} \boldsymbol{a}^\top \boldsymbol{R}_0 \overline{\boldsymbol{\Sigma}}_{\Phi_0} \boldsymbol{R}_0 \boldsymbol{a}, &\quad \tilde{T}_9 &:= \frac{\mu_1}{2\sqrt{N}} \boldsymbol{u}^\top \Big(\boldsymbol{W}_0 \boldsymbol{R}_0 \boldsymbol{\Phi}_0^\top \tilde{\boldsymbol{X}} + \tilde{\boldsymbol{X}}^\top \boldsymbol{\Phi}_0 \boldsymbol{R}_0 \boldsymbol{W}_0^\top\Big) \boldsymbol{u}, \\
T_{11} &:= \boldsymbol{u}^\top \tilde{\boldsymbol{X}}^\top \boldsymbol{\Phi}_0 \boldsymbol{R}_0 \overline{\boldsymbol{\Sigma}}_{\Phi_0} \boldsymbol{R}_0 \boldsymbol{\Phi}_0^\top \tilde{\boldsymbol{X}} \boldsymbol{u}, &\quad T_{10} &:= \|\boldsymbol{u}\|^2, \\
T_{12} &:= \mu_1^* \boldsymbol{u}^\top \tilde{\boldsymbol{X}}^\top \boldsymbol{\Phi}_0 \boldsymbol{R}_0 \overline{\boldsymbol{\Sigma}}_{\Phi_0} \boldsymbol{R}_0 \boldsymbol{\Phi}_0^\top \tilde{\boldsymbol{X}} \boldsymbol{\beta}_*, &\quad \tilde{T}_{12} &:= \boldsymbol{u}^\top \tilde{\boldsymbol{X}}^\top \boldsymbol{\Phi}_0 \boldsymbol{R}_0 \overline{\boldsymbol{\Sigma}}_{\Phi_0} \boldsymbol{R}_0 \boldsymbol{\Phi}_0^\top \boldsymbol{f}_{\mathrm{NL}}^*.
\end{aligned}
$$

In the following subsections we will characterize the limiting value of each $T_i$ as $n, d, N \to \infty$.

### C.3.1 Concentration and simplification

In the following lemma, we show that each $T_i$ will concentrate around some $T_i^0$ given by

$$
\begin{aligned}
T_1^0 &:= \operatorname{tr} \boldsymbol{R}_0, &\quad T_2^0 &:= \frac{\mu_1^2}{N} \theta_1^2 \operatorname{tr}\Big(\tilde{\boldsymbol{X}}^\top \boldsymbol{\Phi}_0 \boldsymbol{R}_0 \boldsymbol{\Phi}_0^\top \tilde{\boldsymbol{X}}\Big), \\
T_3^0 &:= \frac{\mu_1^2 \theta_1^2}{N} \operatorname{tr}\Big(\tilde{\boldsymbol{X}}^\top \tilde{\boldsymbol{X}}\Big), &\quad T_4^0 &:= \mu_1^* \theta_2, \\
T_5^0 &:= \frac{\mu_1^2 \mu_1^* \theta_2}{N} \operatorname{tr}\Big(\tilde{\boldsymbol{X}}^\top \tilde{\boldsymbol{X}}\Big), &\quad T_6^0 &:= \frac{\mu_1 \mu_1^* \theta_2}{\sqrt{N}} \operatorname{tr}\Big(\boldsymbol{W}_0 \boldsymbol{R}_0 \boldsymbol{\Phi}_0^\top \tilde{\boldsymbol{X}}\Big), \\
T_7^0 &:= \frac{\mu_1^2 \mu_1^* \theta_2}{N} \operatorname{tr}\Big(\tilde{\boldsymbol{X}}^\top \boldsymbol{\Phi}_0 \boldsymbol{R}_0 \boldsymbol{\Phi}_0^\top \tilde{\boldsymbol{X}}\Big), &\quad T_8^0 &:= \frac{N}{\mu_1^2} \operatorname{tr}\big(\boldsymbol{R}_0 \overline{\boldsymbol{\Sigma}}_{\Phi_0} \boldsymbol{R}_0\big), &\quad \text{(C.15)} \\
T_9^0 &:= \frac{\mu_1 \theta_1^2}{\sqrt{N}} \operatorname{tr}\Big(\boldsymbol{W}_0 \boldsymbol{R}_0 \boldsymbol{\Phi}_0^\top \tilde{\boldsymbol{X}}\Big), &\quad T_{10}^0 &:= \theta_1^2, \\
T_{11}^0 &:= \theta_1^2 \operatorname{tr}\Big(\tilde{\boldsymbol{X}}^\top \boldsymbol{\Phi}_0 \boldsymbol{R}_0 \overline{\boldsymbol{\Sigma}}_{\Phi_0} \boldsymbol{R}_0 \boldsymbol{\Phi}_0^\top \tilde{\boldsymbol{X}}\Big), &\quad T_{12}^0 &:= \mu_1^* \theta_2 \operatorname{tr}\Big(\tilde{\boldsymbol{X}}^\top \boldsymbol{\Phi}_0 \boldsymbol{R}_0 \overline{\boldsymbol{\Sigma}}_{\Phi_0} \boldsymbol{R}_0 \boldsymbol{\Phi}_0^\top \tilde{\boldsymbol{X}}\Big),
\end{aligned}
$$

where scalars $\theta_1$ and $\theta_2$ are defined in (B.17). In what follows, we first use the following simplification of quadratic forms to obtain the desired $T_i^0$. Here, we extend Lemma 12 to cover the case where matrix $\boldsymbol{D}$ is also random to establish the concentrations for all $T_i$'s in (C.14).

**Lemma 18.** *Consider a random matrix $\boldsymbol{D} \in \mathbb{R}^{d \times d}$ that does not rely on $\boldsymbol{\beta}_*$ and is rotational invariant in distribution, namely $\boldsymbol{D} \stackrel{d}{=} \boldsymbol{O}^\top \boldsymbol{D} \boldsymbol{O}$ for any random rotational matrix $\boldsymbol{O} \in \mathbb{R}^{d \times d}$.*

*Assume that $\|\boldsymbol{D}\| \leq C$ with high probability for some universal constant $C > 0$. Then as $d \to \infty$,*

$$\left|\boldsymbol{\beta}_*^\top \boldsymbol{D} \boldsymbol{\beta}_* - \operatorname{tr} \boldsymbol{D}\right| \xrightarrow{\mathbb{P}} 0,$$

*Also as a corollary, we have $\left|T_i - T_i^0\right| \xrightarrow{\mathbb{P}} 0$ as $n, d, N \to \infty$ proportionally for all $1 \leq i \leq 12$, where $T_i, T_i^0$ are defined in (C.14) and (C.15).*

**Proof.** Given any rotational matrix $\boldsymbol{O} \in \mathbb{R}^{d \times d}$ following the Haar distribution, notice that $\boldsymbol{\beta}_*^\top \boldsymbol{D} \boldsymbol{\beta}_* = \boldsymbol{\beta}_*'^\top \boldsymbol{O}^\top \boldsymbol{D} \boldsymbol{O} \boldsymbol{\beta}_*' \overset{d}{=} \boldsymbol{\beta}_*'^\top \boldsymbol{D} \boldsymbol{\beta}_*'$, where $\boldsymbol{\beta}_*' := \boldsymbol{O}^\top \boldsymbol{\beta}_*$. Consequently, we can equivalently take $\boldsymbol{\beta}_*$ to be a random vector uniformly distributed on the unit sphere $\mathbb{S}^{d-1}$. Notice that $\boldsymbol{\beta}_* \sim \mathrm{Unif}(\mathbb{S}^{d-1})$ satisfies the convex concentration property

$$\mathbb{P}(|f(\boldsymbol{\beta}_*) - \mathbb{E}[f(\boldsymbol{\beta}_*)]| > t) \leq e^{-cdt^2}$$

for any 1-Lipschitz function $f$. Therefore, conditioned on the event $\|\boldsymbol{D}\| \leq C$, by Theorem 2.5 in [Ada15], one can conclude that $\left|\boldsymbol{\beta}_*^\top \boldsymbol{D} \boldsymbol{\beta}_* - \mathbb{E}_{\boldsymbol{\beta}_*}[\boldsymbol{\beta}_*^\top \boldsymbol{D} \boldsymbol{\beta}_*]\right| \xrightarrow{\mathbb{P}} 0$. Finally, note that $\mathbb{E}_{\boldsymbol{\beta}_*}[\boldsymbol{\beta}_*^\top \boldsymbol{D} \boldsymbol{\beta}_*] = \operatorname{tr} \boldsymbol{D}$ because the covariance of the uniform random vector on $\mathbb{S}^{d-1}$ is $\frac{1}{d}\boldsymbol{I}$; this concludes the proof. Convergence of each $T_i$ to the corresponding $T_i^0$ follows from a direct application of Lemma 12 and this lemma. $\qquad\square$

Finally, following the above rotation invariance argument and applying Lemma 4.9 in [MZ20], we can verify that each $\tilde{T}_i$ in (C.14) asymptotically vanishes in probability for $i = 5, 6, 7, 12$.

**Lemma 19.** *Under Assumption 1, as $n, d, N \to \infty$ proportionally, we have*

$$|\tilde{T}_5|, |\tilde{T}_6|, |\tilde{T}_7|, |\tilde{T}_{12}| \xrightarrow{\mathbb{P}} 0.$$

The proof of Lemma 19 is analogous to the proof of Lemma A.5 in [MZ20]. We omit the proof of this lemma here for the sake of clarity and compactness of current paper. For a detailed proof of this lemma, see Appendix C.4 in [BES+22].

### C.3.2  Risk calculation via linear pencils

In this section, we derive analytic expressions of the terms $T_i$ defined in (C.14) as $n, d, N \to \infty$ proportionally. In particular, the exact values are described by self-consistent equations defined in the following proposition.

**Proposition 20.** *Given Assumption 1 and $\lambda > 0$. For $T_i$ defined in (C.14) and $1 \leq i \leq 12$, we have*

$$T_i \to \tau_i,$$

*in probability, as $n/d \to \psi_1$ and $N/d \to \psi_2$, where $\tau_i$'s are defined as follows*

$$\tau_1 := \frac{\psi_1}{\psi_2} m_1 + \left(\frac{\psi_2}{\psi_1} - 1\right)\frac{1}{\lambda}, \qquad \tau_2 := \mu_1^2 \theta_1^2 \frac{\psi_1}{\psi_2}\left(1 - \lambda \frac{\psi_1}{\psi_2} m_2\right), \qquad \tau_3 := \mu_1^2 \theta_1^2 \frac{\psi_1}{\psi_2},$$

$$\tau_4 := \mu_1^* \theta_2, \qquad \tau_5 := \mu_1^2 \mu_1^* \theta_2 \frac{\psi_1}{\psi_2}, \qquad \tau_6 := \mu_1^* \theta_2 \left(1 - \frac{m_2}{m_1}\right),$$

$$\tau_7 := \mu_1^2 \mu_1^* \theta_2 \frac{\psi_1}{\psi_2}\left(1 - \lambda \frac{\psi_1}{\psi_2} m_2\right), \qquad \tau_8 := \frac{m_1 + \frac{\psi_1}{\psi_2}\lambda m_1'}{\left(\mu_1 \frac{\psi_1}{\psi_2}\lambda m_1\right)^2}, \qquad \tau_9 := \theta_1^2 \left(1 - \frac{m_2}{m_1}\right),$$

$$\tau_{10} := \theta_1^2, \qquad \tau_{11} := \theta_1^2 \left(1 - \frac{2m_2}{m_1} - \frac{m_2'}{m_1^2}\right), \qquad \tau_{12} := \mu_1^* \theta_2 \left(1 - \frac{2m_2}{m_1} - \frac{m_2'}{m_1^2}\right),$$

*where $\theta_1$ and $\theta_2$ are defined by (B.17). All scalars $\tau_i$'s are only determined by parameters $\psi_1, \psi_2, \eta, \mu_1, \mu_2, \lambda$, and $m_1, m_2, m_1', m_2'$. Here, $m_1 := m_1\left(\lambda \frac{\psi_1}{\psi_2}\right)$, $m_1' := m_1'\left(\lambda \frac{\psi_1}{\psi_2}\right)$, $m_2 :=$*

$m_2\left(\lambda\frac{\psi_1}{\psi_2}\right)$ and $m_2' := m_2'\left(\lambda\frac{\psi_1}{\psi_2}\right)$, where $m_1(z)$ and $m_2(z) \in \mathbb{C}^+ \cup \mathbb{R}_+$ are the solutions to the following self-consistent equations for $z \in \mathbb{C}^+ \cup \mathbb{R}_+$,

$$\frac{1}{\psi_1}(m_1(z) - m_2(z))(\mu_2^2 m_1(z) + \mu_1^2 m_2(z)) + \mu_1^2 m_1(z)m_2(z)(zm_1(z) - 1) = 0,$$

$$\frac{\psi_2}{\psi_1}\left(\mu_1^2 m_1(z)m_2(z) + \frac{1}{\psi_1}(m_2(z) - m_1(z))\right) + \mu_1^2 m_1(z)m_2(z)(zm_1(z) - 1) = 0. \quad \text{(C.16)}$$

**Proof.** First note that due to Lemma 18, it suffices to consider the limits of $T_i^0$ instead. Convergence of $T_3, T_4, T_5$, and $T_{10}$ directly follows from Lemma 12, Lemma 18 and the Marchenko-Pastur law for $\frac{1}{n}\tilde{\boldsymbol{X}}^\top\tilde{\boldsymbol{X}}$. In addition, $T_7, T_9$ and $T_{12}$ are analogous to $T_2, T_6$ and $T_{11}$, respectively. To characterize the remaining $T_1, T_2, T_6, T_8$ and $T_{11}$, we adopt the *linear pencil* method in basis of operator-valued free probability theory [FOBS06, HFS07, MS17, HMS18]. Specifically, the linear pencil allows us to relate the quantities of interest to the trace of certain large block matrices; in our case, variants of $T_1, T_2, T_6, T_8, T_{11}$ have already appeared in prior constructions from [AP20, BM21, TAP21], which we build upon in the following calculation.

For $z \in \mathbb{C}^+ \cup \mathbb{R}_+$, let us define $\boldsymbol{R}_0(z) := \left(\widehat{\boldsymbol{\Sigma}}_{\Phi_0} + z\boldsymbol{I}\right)^{-1}$ and $\bar{\boldsymbol{R}}_0(z) := \left(\boldsymbol{\Phi}_0\boldsymbol{\Phi}_0^\top + z\boldsymbol{I}\right)^{-1} \in \mathbb{R}^{n\times n}$. Note that due to the Gaussian equivalent property, as $n, N, d \to \infty$ at comparable rate, the limit of $\operatorname{tr}\bar{\boldsymbol{R}}_0(z)$ is exactly the Stieltjes transform of the limiting spectrum of the (nonlinear) CK, namely $\boldsymbol{\Phi}\boldsymbol{\Phi}^\top \in \mathbb{R}^{n\times n}$, evaluated at $-z$. We denote $m_1(z) := \lim_{n\to\infty}\operatorname{tr}\bar{\boldsymbol{R}}_0(z)$. Similarly, the limit of $\operatorname{tr}\boldsymbol{R}_0(z)$ is the *companion* Stieltjes transform of $m_1(z)$, as $\boldsymbol{\Phi}_0\boldsymbol{\Phi}_0^\top$ and $\boldsymbol{\Phi}_0^\top\boldsymbol{\Phi}_0$ have the same non-zero eigenvalues. We denote $\tau_1(z) := \lim_{n\to\infty}\operatorname{tr}\boldsymbol{R}_0(z)$. The defined Stieltjes transforms will be evaluated at $z = \frac{\psi_1}{\psi_2}\lambda$. Also recall the the following relationship between $\boldsymbol{R}_0(z)$ and $\bar{\boldsymbol{R}}_0(z)$,

$$\tau_1(z) = \frac{\psi_1}{\psi_2}m_1(z) + \left(1 - \frac{\psi_1}{\psi_2}\right)\frac{1}{z}. \quad \text{(C.17)}$$

Analogously, we introduce the following quantities: for any $z \in \mathbb{C}^+ \cup \mathbb{R}_+$, as $n, N, d \to \infty$ proportionally,

$$m_2(z) := \lim_{n\to\infty}\operatorname{tr}\left(\frac{1}{d}\tilde{\boldsymbol{X}}\tilde{\boldsymbol{X}}^\top\bar{\boldsymbol{R}}_0(z)\right), \qquad \tau_2(z) := \lim_{n\to\infty}\operatorname{tr}\left(\frac{1}{n}\tilde{\boldsymbol{X}}^\top\boldsymbol{\Phi}_0\boldsymbol{R}_0(z)\boldsymbol{\Phi}_0^\top\tilde{\boldsymbol{X}}\right),$$

$$\tau_6(z) := \lim_{n\to\infty}\frac{1}{\sqrt{N}}\operatorname{tr}\left(\boldsymbol{W}_0\boldsymbol{R}_0(z)\boldsymbol{\Phi}_0^\top\tilde{\boldsymbol{X}}\right), \qquad \tau_8(z) := \lim_{n\to\infty}\operatorname{tr}\left(\boldsymbol{R}_0(z)\left(\mu_1^2\boldsymbol{W}_0^\top\boldsymbol{W}_0 + \mu_2^2\boldsymbol{I}\right)\right).$$

It is straightforward to verify all the above limits exist and are finite. Finally, in the following analysis we will repeatedly make use of the following identities:

$$\boldsymbol{R}_0(z)\boldsymbol{\Phi}_0^\top = \boldsymbol{\Phi}_0^\top\bar{\boldsymbol{R}}_0(z),$$

$$\tilde{\boldsymbol{X}}^\top\boldsymbol{\Phi}_0\boldsymbol{R}_0(z)\boldsymbol{\Phi}_0^\top\tilde{\boldsymbol{X}} = \tilde{\boldsymbol{X}}^\top\tilde{\boldsymbol{X}} - z\tilde{\boldsymbol{X}}^\top\bar{\boldsymbol{R}}_0(z)\tilde{\boldsymbol{X}}.$$

The analyses of individual $T_i$ is based on the risk computation in [AP20, TAP21]. For instance, to match our situation, we can simply set $\sigma_{W_2} = 0$ in [AP20] (which considered the sum of the CK and the first-layer NTK). In the following, we give a concrete example where we derive the limit of $T_6$. We refer to the proof of Proposition 35 in [BES+22] for other omitted parts of this computation.

**Analysis of $T_6$.** For $T_6$, we utilize the computations in Appendix I.6.1 of [TAP21] by setting the covariance $\Sigma = \boldsymbol{I}$. More precisely, based on Equations (S370) and (S418) in [TAP21],

$$\mu_1\tau_6(z) = 1 - G_{6,6}^{K^{-1}} = \frac{z\mu_1^2\psi_1 m_1(z)\tau_1(z)}{1 + z\mu_1^2\psi_1 m_1(z)\tau_1(z)} \overset{(i)}{=} z\mu_1^2\psi_1 m_1(z)\tau_1(z) \overset{(ii)}{=} 1 - \frac{m_2(z)}{m_1(z)}, \quad \text{(C.18)}$$

where $(i)$ and $(ii)$ are both due to (C.16) and (C.17). Hence we obtain the formulae of $\tau_6$ and $\tau_9$.

$\square$

Having obtained the asymptotic expressions of each term in the decomposition of the prediction risk, we can now compute the difference in the prediction risk of CK ridge regression before and after one gradient descent step, i.e., $\mathcal{R}_0(\lambda) - \mathcal{R}_1(\lambda)$ in Theorem 5. The following statement is the complete version of Theorem 5.

**Theorem 21.** *Given Assumption 1, consider $\psi_1, \psi_2 \in (0, +\infty)$. Fix $\eta = \Theta(1)$ and $\lambda > 0$. Denote $\mathcal{R}_0(\lambda)$ and $\mathcal{R}_1(\lambda)$ as the prediction risk of CK ridge regression in (4.1) using initial weight $\boldsymbol{W}_0$ and first-step updated $\boldsymbol{W}_1$, respectively. The difference between these two risk values satisfies*

$$\mathcal{R}_0(\lambda) - \mathcal{R}_1(\lambda) \xrightarrow{\mathbb{P}} \delta(\eta, \lambda, \psi_1, \psi_2),$$

*where $\delta$ is a non-negative function of $\eta, \lambda, \psi_1, \psi_2 \in (0, +\infty)$ with parameters $\mu_1^*, \mu_1, \mu_2$ given as*

$$
\delta(\eta, \lambda, \psi_1, \psi_2) = \frac{\tau_1(\tau_7 - \tau_5)(\tau_4 + \tau_{12} - 2\tau_6)}{\tau_1(\tau_2 - \tau_3) - 1}
$$
$$
- \frac{\tau_1(\tau_7 - \tau_5)(\tau_4 + \tau_{12} - 2\tau_6) + (\tau_7 - \tau_5)^2 \tau_8}{(\tau_1(\tau_2 - \tau_3) - 1)^2}. \tag{C.19}
$$

*Here the scalars $\tau_i$'s are defined in Proposition 20. Furthermore, $\delta(\eta, \lambda, \psi_1, \psi_2) = 0$ if and only if at least one of $\mu_1^*, \mu_1$ and $\eta$ is zero.*

**Proof.** Due to Lemma 17, we can see that variance $V$ is unchanged after one gradient descent step with $\eta = \Theta(1)$. Hence we only need to analyze the changes in (C.9) and (C.10). Also, due to Lemma 17 and Lemma 10, we can ignore $\boldsymbol{B}$ and $\boldsymbol{C}$ in $\boldsymbol{W}_1$ and take $\boldsymbol{W}_1 := \boldsymbol{W}_0 + \boldsymbol{u}\boldsymbol{a}^\top$, where $\boldsymbol{u} = \frac{\mu_1 \eta}{n} \boldsymbol{X}^\top \boldsymbol{y}$ and $\boldsymbol{y} = f^*(\boldsymbol{X}) + \boldsymbol{\varepsilon}$, without changing the bias terms.

**Separation of low-rank terms.** First note that if $\mu_1 = 0$, then $\boldsymbol{u} = \boldsymbol{0}$ and therefore $\mathcal{R}_0(\lambda) = \mathcal{R}_1(\lambda)$ as $n \to \infty$. In the following, we take $\mu_1 \neq 0$ which implies that $\theta_1$ defined in (B.17) will not vanish. Now we aim to extract the low-rank perturbation $\boldsymbol{u}\boldsymbol{a}^\top$ from bias terms (C.9) and (C.10). We adhere to the notions in (C.13), (C.14) and (C.15) and define $D := T_1(T_2 - T_3) - 1$. Similar to [MM22, Lemma C.1], we use the following linearization trick to separate the gradient step $\boldsymbol{u}\boldsymbol{a}^\top$ from the matrices $\boldsymbol{R}, \bar{\boldsymbol{\Phi}}, \overline{\boldsymbol{\Sigma}}_\Phi$ and $\boldsymbol{W}_1$.

Define $\boldsymbol{b} := \frac{\mu_1}{\sqrt{N}} \tilde{\boldsymbol{X}} \boldsymbol{u}$ and $\boldsymbol{c} := \boldsymbol{\Phi}_0^\top \boldsymbol{b}$; observe that $T_2 = \boldsymbol{c}^\top \boldsymbol{R}_0 \boldsymbol{c}, T_3 = \boldsymbol{b}^\top \boldsymbol{b}$, and

$$
\widehat{\boldsymbol{\Sigma}}_\Phi = \widehat{\boldsymbol{\Sigma}}_{\Phi_0} + \begin{bmatrix} \boldsymbol{a} & \boldsymbol{c} \end{bmatrix} \begin{bmatrix} T_3 & 1 \\ 1 & 0 \end{bmatrix} \begin{bmatrix} \boldsymbol{a}^\top \\ \boldsymbol{c}^\top \end{bmatrix}.
$$

Therefore, by the Sherman-Morrison-Woodbury formula and Hanson-Wright inequality, we have

$$
\boldsymbol{R} = \boldsymbol{R}_0 - \boldsymbol{\Delta}_{aa} - \boldsymbol{\Delta}_{cc} + \boldsymbol{\Delta}_{ac} + \boldsymbol{\Delta}_{ca} + o_{d,\mathbb{P}}(1), \tag{C.20}
$$

where we further defined

$$
\boldsymbol{\Delta}_{aa} := \frac{T_2 - T_3}{D} \boldsymbol{R}_0 \boldsymbol{a}\boldsymbol{a}^\top \boldsymbol{R}_0, \quad \boldsymbol{\Delta}_{cc} := \frac{T_1}{D} \boldsymbol{R}_0 \boldsymbol{c}\boldsymbol{c}^\top \boldsymbol{R}_0,
$$
$$
\boldsymbol{\Delta}_{ca} := \frac{1}{D} \boldsymbol{R}_0 \boldsymbol{c}\boldsymbol{a}^\top \boldsymbol{R}_0, \quad \boldsymbol{\Delta}_{ac} := \frac{1}{D} \boldsymbol{R}_0 \boldsymbol{a}\boldsymbol{c}^\top \boldsymbol{R}_0.
$$

Consider the linear part of the subtracted term in (C.9):

$$
B_{1,1} := -\frac{2\mu_1 \mu_1^{*2}}{\sqrt{N}} \boldsymbol{\beta}_*^\top \boldsymbol{W}_1 \boldsymbol{R} \bar{\boldsymbol{\Phi}}^\top \tilde{\boldsymbol{X}} \boldsymbol{\beta}_*, \qquad B_{1,1}^0 := -\frac{2\mu_1 \mu_1^{*2}}{\sqrt{N}} \boldsymbol{\beta}_*^\top \boldsymbol{W}_0 \boldsymbol{R}_0 \boldsymbol{\Phi}_0^\top \tilde{\boldsymbol{X}} \boldsymbol{\beta}_*.
$$

By repeatedly applying the Hanson-Wright inequality (since $\boldsymbol{a}$ is centered and independent of all other terms) and Lemma 19, we can employ decomposition (C.20) to obtain

$$
B_{1,1} = B_{1,1}^0 - 2T_1(T_7 - T_5)(T_4 - T_6)/D + o_{d,\mathbb{P}}(1).
$$

Now we denote $\boldsymbol{\Delta}_{ua} := \frac{\mu_1^2}{N} \boldsymbol{W}_0^\top \boldsymbol{u}\boldsymbol{a}^\top$, $\boldsymbol{\Delta}_{au} := \boldsymbol{\Delta}_{ua}^\top$ and $\boldsymbol{\Delta}_{aua} := \frac{\mu_1^2 T_{10}}{N} \boldsymbol{a}\boldsymbol{a}^\top$. Hence,

$$
\overline{\boldsymbol{\Sigma}}_\Phi = \overline{\boldsymbol{\Sigma}}_{\Phi_0} + \boldsymbol{\Delta}_{ua} + \boldsymbol{\Delta}_{au} + \boldsymbol{\Delta}_{aua}. \tag{C.21}
$$

Let $B_2^0 := \boldsymbol{f}^{*\top} \tilde{\boldsymbol{X}}^\top \boldsymbol{\Phi}_0 \boldsymbol{R}_0 \overline{\boldsymbol{\Sigma}}_{\Phi_0} \boldsymbol{R}_0 \boldsymbol{\Phi}_0^\top \tilde{\boldsymbol{X}} \boldsymbol{f}^*$. With (C.20) and (C.21), we can decompose $B_2$ defined in (C.10) as follows

$$
B_2 = B_2^0 + \frac{2T_1(T_7 - T_5)(T_6 - T_{12})}{D} + \frac{(T_7 - T_5)^2(T_1^2 T_{11} + T_1^2 T_{10} + T_8 - 2T_1^2 T_9)}{D^2} + o_{d,\mathbb{P}}(1),
$$

where we repeatedly make use of Lemma 19 and the concentration for $a$ to simplify the computations. Therefore, one can obtain

$$\mathcal{R}_0(\lambda) - \mathcal{R}_1(\lambda) = B_{1,1}^0 - B_{1,1} + B_2^0 - B_2 + o_{d,\mathbb{P}}(1)$$

$$= \frac{2T_1(T_7 - T_5)(T_4 + T_{12} - 2T_6)}{D} - \frac{(T_7 - T_5)^2(T_1^2 T_{11} + T_1^2 T_{10} + T_8 - 2T_1^2 T_9)}{D^2} + o_{d,\mathbb{P}}(1).$$

For the details of the above computation, we refer to the Theorem 36 in [BES$^+$22]. Meanwhile, from Proposition 20 we know that

$$\mathcal{R}_0(\lambda) - \mathcal{R}_1(\lambda) \xrightarrow{\mathbb{P}} \underbrace{\frac{2\tau_1(\tau_7 - \tau_5)(\tau_4 + \tau_{12} - 2\tau_6)}{\tau_1(\tau_2 - \tau_3) - 1} - \frac{(\tau_7 - \tau_5)^2(\tau_1^2 \tau_{11} + \tau_1^2 \tau_{10} + \tau_8 - 2\tau_1^2 \tau_9)}{(\tau_1(\tau_2 - \tau_3) - 1)^2}}_{\triangleq \delta(\eta, \lambda, \psi_1, \psi_2)},$$

where the right hand side is the quantity of interest $\delta(\eta, \lambda, \psi_1, \psi_2)$ defined in Theorem 5. Also observe the following equivalences from Proposition 20,

$$\mu_1^* \theta_2(\tau_2 - \tau_3) = \theta_1^2(\tau_7 - \tau_5), \quad \mu_1^* \theta_2(\tau_{11} + \tau_{10} - 2\tau_9) = \theta_1^2(\tau_4 + \tau_{12} - 2\tau_6).$$

Hence, we can simplify $\delta(\eta, \lambda, \psi_1, \psi_2)$ to conclude (C.19).

**Non-negativity of** $\delta(\eta, \lambda, \psi_1, \psi_2)$. Finally, we validate that the function $\delta(\eta, \lambda, \psi_1, \psi_2)$ is non-negative on variables $\eta, \lambda, \psi_1$ and $\psi_2 \in (0, +\infty)$. Observe that the formula of $\delta(\eta, \lambda, \psi_1, \psi_2)$ in (C.19) is decomposed into two parts. From Proposition 20 we know that $\tau_1$ and $m_1$ are the limits of $\operatorname{tr} \boldsymbol{R}_0(z)$ and $\operatorname{tr} \bar{\boldsymbol{R}}_0(z)$ evaluated at $z = \psi_1 \lambda / \psi_2$; this indicates that $\tau_1 \in (0, \psi_2/\lambda\psi_1]$ is non-negative. For the same reason, $m_2 \in (0, \psi_2/\lambda\psi_1]$ and $-m_1', -m_2' \in (0, \psi_2^2/\lambda^2\psi_1^2]$. Also due to Proposition 20, we have

$$\tau_2 - \tau_3 = -\mu_1^2 \theta_1^2 \left(\frac{\psi_1}{\psi_2}\right)^2 \lambda m_2 \le 0, \qquad \tau_7 - \tau_5 = -\mu_1^2 \mu_1^* \theta_2 \left(\frac{\psi_1}{\psi_2}\right)^2 \lambda m_2 \le 0,$$

$$\tau_4 + \tau_{12} - 2\tau_6 = -\mu_1^* \theta_2 \frac{m_2'}{m_1^2} \ge 0, \qquad \tau_{11} + \tau_{10} - 2\tau_9 = -\theta_1^2 \frac{m_2'}{m_1^2} \ge 0, \qquad \text{(C.22)}$$

$$\tau_8 = \frac{1}{m_1} \frac{1}{\mu_1^2 \lambda^2} \left(\frac{\psi_2}{\psi_1}\right)^2 + \frac{m_1'}{m_1^2} \left(\frac{\psi_2}{\psi_1}\right) \frac{1}{\mu_1^2 \lambda}.$$

Therefore, $\tau_1(\tau_7 - \tau_5)(\tau_4 + \tau_{12} - 2\tau_6) \le 0$ and $\tau_1(\tau_2 - \tau_3) - 1 \le -1$. This entails that the first part of $\delta(\eta, \lambda, \psi_1, \psi_2)$ is non-negative:

$$\frac{\tau_1(\tau_7 - \tau_5)(\tau_4 + \tau_{12} - 2\tau_6)}{\tau_1(\tau_2 - \tau_3) - 1} \ge 0.$$

As for the second part, it suffices to evaluate $\Delta := \tau_1(\tau_4 + \tau_{12} - 2\tau_6) + (\tau_7 - \tau_5)\tau_8$ since

$$-\frac{\tau_1(\tau_7 - \tau_5)(\tau_4 + \tau_{12} - 2\tau_6) + (\tau_7 - \tau_5)^2 \tau_8}{(\tau_1(\tau_2 - \tau_3) - 1)^2} = \frac{(\tau_5 - \tau_7)\Delta}{(\tau_1(\tau_2 - \tau_3) - 1)^2}.$$

Plugging in quantities in (C.22) with $z = \lambda\psi_1/\psi_2$, we have

$$\Delta = -\mu_1^* \theta_2 \left(\frac{\psi_1}{\psi_2} \frac{m_2}{zm_1^2}(m_1 + zm_1') + \frac{\tau_1 m_2'}{m_1^2}\right)$$

$$= -\frac{\mu_1^* \theta_2}{zm_1^2} \left(\frac{\psi_1}{\psi_2}(m_1 m_2 + zm_2 m_1' + zm_1 m_2') + \left(1 - \frac{\psi_1}{\psi_2}\right)m_2'\right)$$

$$= -\frac{\mu_1^* \theta_2}{zm_1^2} \frac{d}{dz}\Big|_{z=\lambda\psi_1/\psi_2} \left(\frac{\psi_1}{\psi_2}zm_1(z)m_2(z) + \left(1 - \frac{\psi_1}{\psi_2}\right)m_2(z)\right)$$

$$\overset{(i)}{=} -\frac{\mu_1^* \theta_2}{zm_1^2 \psi_1 \mu_1^2} \frac{d}{dz}\Big|_{z=\lambda\psi_1/\psi_2} \left(1 - \frac{m_2(z)}{m_1(z)}\right)$$

$$\overset{(ii)}{=} -\frac{\mu_1^* \theta_2}{zm_1^2 \mu_1^2} \frac{d}{dz}\Big|_{z=\lambda\psi_1/\psi_2} z\mu_1^2 m_1(z)\tau_1(z),$$

where $(i)$ and $(ii)$ are due to (C.16) and (C.18), respectively. By Lemma A.1 in [TAP21], we know function $z\mu_1^2 m_1(z)\tau_1(z)$ has non-positive derivative when $z > 0$. This implies that $\Delta \geq 0$ and hence the second part of $\delta(\eta, \lambda, \psi_1, \psi_2)$ is also non-negative.

Finally, we note that when $\mu_1^* = 0$, the function $\delta(\eta, \lambda, \psi_1, \psi_2) = 0$. This is because

$$\tau_7 - \tau_5 = -\mu_1^2 \mu_1^* \theta_2 \psi_1^2 \lambda m_2 / \psi_2^2 = 0,$$

when $\mu_1^* = 0$. Whereas when $\eta = 0$, we know that $\theta_1 = \theta_2 = 0$, which entails $\delta(\eta, \lambda, \psi_1, \psi_2)$ is also vanishing. Also observe that in (C.22), $m_1, m_2, m_1', m_2', \tau_1$ are all positive. Hence we conclude that if $\delta(\eta, \lambda, \psi_1, \psi_2) = 0$, then at least one of $\eta, \mu_1 \mu_1^*$ must be zero. $\qquad\square$

# D    Proof for large learning rate ($\eta = \Theta(\sqrt{N})$)

In this section we restrict ourselves to a single-index target function (generalized linear model): $f^*(\boldsymbol{x}) = \sigma^*(\langle \boldsymbol{x}, \boldsymbol{\beta}_* \rangle)$, and study the impact of one gradient step with large learning rate $\eta = \Theta(\sqrt{N})$. For simplicity, we denote $\eta = \bar{\eta}\sqrt{N}$ where $\bar{\eta} > 0$ is a fixed constant not depending on $N$.

As the Gaussian equivalence property is no longer applicable, we instead establish an upper bound on the prediction risk of the CK ridge estimator. Our proof is divided into two parts: $(i)$ we show that there exists an "oracle" second-layer $\tilde{\boldsymbol{a}}$ that achieves small prediction risk $\tau^*$ when $n/d$ is large; $(ii)$ based on $\tau^*$, we provide an upper bound on the prediction risk when the second layer is estimated via ridge regression.

Here we provide a short summary on the construction of $\tilde{\boldsymbol{a}}$ and upper bound on the prediction risk.

- We first introduce $f_r(\boldsymbol{x}) := \frac{1}{|\mathcal{A}_r|}\sum_{i\in\mathcal{A}_r}\sigma(\langle \boldsymbol{x}, \boldsymbol{w}_i^1 \rangle)$, which is the average of a subset of neurons in $\mathcal{A}_r \subset [N]$ defined in (D.4). Intuitively, this subset of neurons approximately matches the target direction $\boldsymbol{\beta}_*$. This averaging corresponds to setting the second-layer $\tilde{\boldsymbol{a}}_i = \frac{\sqrt{N}}{|\mathcal{A}_r|}$ for all $i \in \mathcal{A}_r$.

- We show that $f_r$ can be approximated up to $\Theta(d/n)$-error by an "expected" single-index model $\bar{f}(\boldsymbol{x}) := \mathbb{E}_{\boldsymbol{w}\sim\mathcal{N}(0, \boldsymbol{I}/d)}[\sigma(\langle \boldsymbol{w} + c\boldsymbol{\beta}_*, \boldsymbol{x} \rangle)]$, for some $c \in \mathbb{R}$ that depends on the learning rate and nonlinearities. To bound this substitution error, we establish a more refined control of gradient norm in Section D.1.

- By choosing an "optimal" subset $\mathcal{A}_r$, we simplify the prediction risk of $\bar{f}$ into the one-dimensional expectation $\tau^*$ defined in (5.1). This provides a high-probability upper bound of the prediction risk of the constructed $\tilde{\boldsymbol{a}}$ up to $\Theta(d/n)$-error.

After constructing some $\tilde{\boldsymbol{a}}$ that achieves reasonable test performance, we can then show that the prediction risk of CK ridge regression estimator with trained weight $\boldsymbol{W}_1$ is also upper-bounded by $\tau^*$ when $n \gg d$. This result is established in Section D.3 and follows from classical analysis of kernel ridge regression.

## D.1    Refined properties of the first-step gradient

Recall that $\boldsymbol{W}_1 = \boldsymbol{W}_0 + \eta\sqrt{N}\boldsymbol{G}_0$, where $\boldsymbol{G}_0 = \boldsymbol{A}_1 + \boldsymbol{A}_2 + \boldsymbol{B} + \boldsymbol{C}$ is defined in Lemma 10 and 11, and the full-rank term $\boldsymbol{B}$ is given as

$$\boldsymbol{B} = \frac{1}{n} \cdot \frac{1}{\sqrt{N}}\boldsymbol{X}^\top \big(\boldsymbol{y}\boldsymbol{a}^\top \odot \sigma'_\perp(\boldsymbol{X}\boldsymbol{W}_0)\big).$$

We first refine the estimate on the Frobenius norm of certain submatrix of $\boldsymbol{B}$; the choice of such submatrices will be explained in Section D.2.

**Lemma 22.** *Given Assumption 1, take $\boldsymbol{B}_r \in \mathbb{R}^{d \times N_r}$ which is a submatrix of $\boldsymbol{B}$ via selecting any $N_r \in [N]$ columns in $\boldsymbol{B}$, and let $\boldsymbol{a}_r \in \mathbb{R}^{N_r}$ be the corresponding 2nd layer coefficients. If entries of $\boldsymbol{a}_r$ are uniformly bounded by $\alpha/\sqrt{N}$, then for any $\varepsilon \in (0, 1/4)$, we have*

$$\mathbb{E}\|\boldsymbol{B}_r\|_F^2 \leq \frac{C_0\alpha^2 N_r}{N}\left(\frac{1}{Nd^{\frac{1}{2}-\varepsilon}} + \frac{d}{nN}\right), \tag{D.1}$$

*and*

$$\mathbb{P}\left(\frac{N}{N_r}\|\boldsymbol{B}_r\|_F^2 \leq \frac{C_1}{Nd^{\frac{1}{4}-\varepsilon}} + \frac{C_2 d}{Nn}\right) \geq 1 - \frac{\alpha^2}{d^{\frac{1}{4}}} - \frac{\alpha^4}{n}, \tag{D.2}$$

*where constants $C_0, C_1, C_2 > 0$ only depend on $\lambda_\sigma$ and $\|f^*\|_{L^2(\mathbb{R}^d, \Gamma)}$.*

**Proof.** Let $\boldsymbol{X}^\top = (\boldsymbol{x}_1, \boldsymbol{x}_2, \ldots, \boldsymbol{x}_n)$ and $\boldsymbol{y}^\top = (y_1, \ldots, y_n)$. Then, matrix $\boldsymbol{B}_r$ can be written as

$$\boldsymbol{B}_r = \frac{1}{n\sqrt{N}} \sum_{i=1}^n y_i \boldsymbol{x}_i \sigma'_\perp(\boldsymbol{x}_i^\top \boldsymbol{W}_0^r) \operatorname{diag}(\boldsymbol{a}_r),$$

where $\boldsymbol{W}_0^r \in \mathbb{R}^{d \times N_r}$ is a submatrix of $\boldsymbol{W}_0$ by choosing any $N_r$ columns of $\boldsymbol{W}_0$, and $\boldsymbol{a}_r \in \mathbb{R}^{N_r}$ is the corresponding second layer (note that by assumption $\|\boldsymbol{a}_r\|_\infty \leq \alpha/\sqrt{N}$). Hence,

$$\|\boldsymbol{B}_r\|_F^2 = \operatorname{Tr}(\boldsymbol{B}_r \boldsymbol{B}_r^\top) = \frac{1}{n^2 N} \sum_{i,j=1}^n y_i y_j \operatorname{Tr}\left[\boldsymbol{x}_i \sigma'_\perp(\boldsymbol{x}_i^\top \boldsymbol{W}_0^r) \operatorname{diag}(\boldsymbol{a}_r)^2 \sigma'_\perp(\boldsymbol{x}_j^\top \boldsymbol{W}_0^r)^\top \boldsymbol{x}_j^\top\right]$$

$$= \frac{1}{n^2 N} \sum_{i,j=1}^n y_i y_j \left(\sigma'_\perp(\boldsymbol{x}_i^\top \boldsymbol{W}_0^r) \operatorname{diag}(\boldsymbol{a}_r)^2 \sigma'_\perp(\boldsymbol{x}_j^\top \boldsymbol{W}_0^r)^\top \boldsymbol{x}_j^\top \boldsymbol{x}_i\right)$$

$$= \frac{1}{n^2 N} \sum_{i \neq j}^n y_i y_j \left(\sigma'_\perp(\boldsymbol{x}_i^\top \boldsymbol{W}_0^r) \operatorname{diag}(\boldsymbol{a}_r)^2 \sigma'_\perp(\boldsymbol{x}_j^\top \boldsymbol{W}_0^r)^\top \boldsymbol{x}_j^\top \boldsymbol{x}_i\right)$$

$$+ \frac{1}{n^2 N} \sum_{i=1}^n y_i^2 \left(\sigma'_\perp(\boldsymbol{x}_i^\top \boldsymbol{W}_0^r) \operatorname{diag}(\boldsymbol{a}_r)^2 \sigma'_\perp(\boldsymbol{W}_0^{r\top} \boldsymbol{x}_i)\|\boldsymbol{x}_i\|^2\right) =: J_1 + J_2.$$

Here, $J_1$ represents the sum for distinct $i \neq j \in [n]$ and $J_2$ is the sum when $i = j \in [n]$. Therefore,

$$\mathbb{E}[\|\boldsymbol{B}_r\|_F^2] \leq \frac{\alpha^2 N_r}{N} \frac{n(n-1)}{n^2 N} \mathbb{E}\left[f^*(\boldsymbol{x}_1) f^*(\boldsymbol{x}_2) \sigma'_\perp(\boldsymbol{x}_1^\top \boldsymbol{w}) \sigma'_\perp(\boldsymbol{x}_2^\top \boldsymbol{w}) \boldsymbol{x}_2^\top \boldsymbol{x}_1\right]$$

$$+ \frac{\alpha^2 N_r}{nN^2} \mathbb{E}\left[(f^*(\boldsymbol{x}_1)^2 + \sigma_\varepsilon^2) \sigma'_\perp(\boldsymbol{x}_1^\top \boldsymbol{w})^2 \|\boldsymbol{x}_1\|^2\right],$$

where $\boldsymbol{w} \sim \mathcal{N}(0, \boldsymbol{I})$ independent of $\boldsymbol{a}, \boldsymbol{X}$. We compute the aforementioned expectations as follows

$$\frac{N}{\alpha^2 N_r} \mathbb{E}[\|\boldsymbol{B}\|_F^2] \leq \frac{1}{N} \left|\mathbb{E}\left[f^*(\boldsymbol{x}_1) f^*(\boldsymbol{x}_2) \sigma'_\perp(\boldsymbol{x}_1^\top \boldsymbol{w}) \sigma'_\perp(\boldsymbol{x}_2^\top \boldsymbol{w}) \boldsymbol{x}_2^\top \boldsymbol{x}_1\right]\right|$$

$$+ \frac{1}{nN} \mathbb{E}\left[f^*(\boldsymbol{x}_1)^2 \sigma'_\perp(\boldsymbol{x}_1^\top \boldsymbol{w})^2 \|\boldsymbol{x}_1\|^2\right] + \frac{\sigma_\varepsilon^2}{nN} \mathbb{E}\left[\sigma'_\perp(\boldsymbol{x}_1^\top \boldsymbol{w})^2 \|\boldsymbol{x}_1\|^2\right] =: I_1 + I_2 + I_3.$$

To verify (D.1), we in turn control $I_1, I_2$ and $I_3$. For $I_1$, the moment calculation in [BES$^+$22, Appendix D.1] entails that $I_1 \leq C \frac{d}{N} d^{\varepsilon - 3/2}$, for all large $d$, sufficiently large constant $C > 0$ and sufficient small $\varepsilon$. As for $I_2$ and $I_3$, notice that

$$I_2 \leq \frac{\lambda_\sigma^2}{nN} \mathbb{E}[f^*(\boldsymbol{x}_1)^4]^{\frac{1}{2}} \mathbb{E}[\|\boldsymbol{x}_1\|^4]^{\frac{1}{2}} \leq \frac{3C\lambda_\sigma^2 d}{nN},$$

because $\sigma^*$ is Lipschitz and $f^* \in L^4(\mathbb{R}^d, \Gamma)$. Following the same computation, we also have $I_3 \leq \frac{\lambda_\sigma^2 d}{nN}$. This establishes a bound for $\mathbb{E}[[\|\boldsymbol{B}\|_F^2]$ in (D.1).

For the tail control (D.2), recall that $\|\boldsymbol{B}_r\|_F^2 = J_1 + J_2$ where $\mathbb{E}[|J_1|] \leq \frac{\alpha^2 N_r}{N} I_1$ and $\mathbb{E}[J_2] \leq \frac{\alpha^2 N_r}{N}(I_2 + I_3)$. Hence Markov's inequality and the upper bound for $I_1$ implies that

$$\mathbb{P}\left(\frac{N}{N_r}|J_1| \geq t\right) \leq \frac{C\alpha^2}{t d^{\frac{1}{2}-\varepsilon} N}.$$

By choosing $t = C/Nd^{\frac{1}{4}-\varepsilon}$, we conclude that $\frac{N}{N_r}|J_1|$ cannot exceed $C/Nd^{\frac{1}{4}-\varepsilon}$ with probability at least $1 - \alpha^2/d^{\frac{1}{4}}$, for any $\varepsilon \in (0, 1/4)$. As for $J_2$, since $\sigma'_\perp$ is uniformly bounded by $\lambda_\sigma$ and all entries of $\boldsymbol{a}_r$ are bounded by $\alpha/\sqrt{N}$, we have

$$\frac{N}{N_r}|J_2| \leq \frac{\lambda_\sigma^2 \alpha^2}{Nn^2} \sum_{i=1}^n y_i^2 \|\boldsymbol{x}_i\|^2 =: J_2'.$$

Similarly for $I_2$ and $I_3$, it is easy to check $|\mathbb{E}[J_2']| \leq \frac{3\alpha^2 Cd}{Nn}$. Besides,

$$\mathrm{Var}(J_2') = \frac{\lambda_\sigma^4 \alpha^4}{N^2 n^3} \mathrm{Var}(y_1^2 \|\boldsymbol{x}_1\|^2) \leq \frac{\lambda_\sigma^4 \alpha^4}{N^2 n^3} \mathbb{E}[y_1^4 \|\boldsymbol{x}_1\|^4] \leq \frac{c\alpha^4 d^2}{N^2 n^3},$$

where constant $c > 0$ only depends on $\lambda_\sigma$ and $\|f^*\|_{L^8(\mathbb{R},\Gamma)}$. By Chebyshev's inequality,

$$\mathbb{P}(|J_2' - \mathbb{E}[J_2']| > t) \leq \frac{c\alpha^4 d^2}{t^2 N^2 n^3}.$$

Letting $t = \sqrt{cd}/Nn$, we arrive at

$$\mathbb{P}\left(\frac{N}{N_r} J_2 \leq \frac{\sqrt{cd}}{Nn} + \frac{3Cd}{Nn}\right) \geq \mathbb{P}\left(J_2' \leq \frac{\sqrt{cd}}{Nn} + \frac{3Cd}{Nn}\right) \geq 1 - \frac{\alpha^4}{n}.$$

We conclude (D.2) by combining the above estimates of $J_1$ and $J_2$.

$\qquad\square$

## D.2 Construction of "oracle" estimator

In this subsection we prove the following lemma related to Lemma 6.

**Lemma 23** (Reformulation of Lemma 6). *Suppose Assumption 1 holds, $\eta = \Theta(\sqrt{N})$ and the activation $\sigma$ is bounded. Then given any $\varepsilon > 0$, for $N$ sufficiently large, there exists some constant $C$ and second-layer $\tilde{\boldsymbol{a}}$ such that the model $\tilde{f}(\boldsymbol{x}) = \frac{1}{\sqrt{N}} \tilde{\boldsymbol{a}}^\top \sigma(\boldsymbol{W}_1^\top \boldsymbol{x})$ has prediction risk*

$$\mathcal{R}(\tilde{f}) \leq \tau^* + C\left(\sqrt{\tau^*} \cdot \sqrt{\frac{d}{n}} + \frac{d}{n}\right) + \varepsilon + o_{d,\mathbb{P}}(1), \tag{D.3}$$

*where the scalar $\tau^*$ is defined in (5.1).*

We first introduce a constant $\alpha$ (independent to $N$). Recall that $[\boldsymbol{a}]_i = a_i \overset{\text{i.i.d.}}{\sim} \mathcal{N}(0, N^{-1})$ for $i \in [N]$. For any $\alpha \in \mathbb{R}$, define the subset of initialized weights:

$$\mathcal{A}_r^\alpha = \left\{i \in [N] : \left|\sqrt{N} \cdot a_i - \alpha\right| \leq N^{-r}\right\}, \quad \text{for any given } r > 0. \tag{D.4}$$

The size of the subset is given by $|\mathcal{A}_r^\alpha| = \sum_{i=1}^N \mathbf{1}_{|\sqrt{N}a_i - \alpha| \leq N^{-r}}$, and hence its expectation is $\mathbb{E}|\mathcal{A}_r^\alpha| = N \cdot \mathbb{E}_{z \sim \mathcal{N}(0,1)}[\mathbf{1}_{|z-\alpha| \leq N^{-r}}] = C(\alpha) N^{1-r}$ for some constant $C(\alpha) \propto \exp(-\alpha^2)$. By Hoeffding's inequality,

$$\mathbb{P}(||\mathcal{A}_r^\alpha| - \mathbb{E}|\mathcal{A}_r^\alpha|| \geq t) \leq 2\exp\left(-\frac{2t^2}{N}\right). \tag{D.5}$$

Hence we may conclude that for any $r \in (0, 1/2)$ and large enough $N$, $|\mathcal{A}_r^\alpha| = \Theta_{d,\mathbb{P}}(N^{1-r})$ with probability at least $1 - 2\exp(-c\log^2 N)$. This is to say, for any constant $\alpha$, we know that with high probability, there exist a large number of initialized second-layer coefficients $a_i$'s that are close to $\alpha$. We specify our choice of $\alpha \in \mathbb{R}$ via (5.1) in the subsequent analysis.

**Rank-1 approximation of gradient.** Denote $N_r := |\mathcal{A}_r^\alpha|$ for some constant $\alpha$, and $i_r \in [N]$ as the index such that $i_r \in \mathcal{A}_r^\alpha$. We define $f_r$ as an average over neurons with indices $i_r \in \mathcal{A}_r^\alpha$, and $f_A$ as an approximation of $f_r$ in which the first-step gradient matrix $\boldsymbol{G}_0$ in (B.1) is replaced by the rank-1 matrix $\boldsymbol{A}_1$ defined in (B.14):

$$f_r(\boldsymbol{x}) := \frac{1}{N_r} \sum_{i_r \in \mathcal{A}_r^\alpha} \sigma(\langle \boldsymbol{x}, \boldsymbol{w}_{i_r}^1 \rangle); \quad f_A(\boldsymbol{x}) := \frac{1}{N_r} \sum_{i_r \in \mathcal{A}_r^\alpha} \sigma(\langle \boldsymbol{x}, \boldsymbol{w}_{i_r}^A \rangle), \tag{D.6}$$

where $\boldsymbol{w}_{i_r}^1$ is the $i_r$-th neuron in $\boldsymbol{W}_1$, $\boldsymbol{w}_i^A = \boldsymbol{w}_i^0 + \eta\sqrt{N}[\boldsymbol{A}_1]_i$, $[\boldsymbol{A}_1]_i$ is the $i$-th column of $\boldsymbol{A}_1$ and $\boldsymbol{w}_i^0 \in \mathbb{R}^d$ is the corresponding initial neuron in $\boldsymbol{W}_0$. Applying the Lipschitz property of the activation function, one can control $\mathbb{E}_{\boldsymbol{x}}[(f_A(\boldsymbol{x}) - f_r(\boldsymbol{x}))^2]$ as follows

$$|f_A(\boldsymbol{x}) - f_r(\boldsymbol{x})| \lesssim \frac{1}{N_r} \sum_{i_r \in \mathcal{A}_r^\alpha} |\langle \boldsymbol{w}_{i_r}^1 - \boldsymbol{w}_{i_r}^A, \boldsymbol{x} \rangle| = \frac{\eta\sqrt{N}}{N_r} \sum_{i_r \in \mathcal{A}_r^\alpha} |\langle \boldsymbol{\delta}_{i_r}, \boldsymbol{x} \rangle|.$$

where the "residual" is entry-wisely defined as $[\boldsymbol{\delta}_i]_j := [\boldsymbol{A}_2 + \boldsymbol{B} + \boldsymbol{C}]_{ji}$ for $i \in [N]$ and $j \in [d]$. Recall that $\boldsymbol{A}_2, \boldsymbol{B}$ and $\boldsymbol{C}$ have been analyzed in Lemmas 10 and 11. Let us further denote $\boldsymbol{A}_r \in \mathbb{R}^{d \times N_r}$ as a submatrix of $\boldsymbol{A}_2$ by selecting all $i_r \in \mathcal{A}_r^\alpha$ columns of $\boldsymbol{A}_2$. Similarly, we choose $\boldsymbol{B}_r, \boldsymbol{C}_r \in \mathbb{R}^{d \times N_r}$ as submatrices of $\boldsymbol{B}, \boldsymbol{C}$ related to $\mathcal{A}_r^\alpha$, respectively. Using Lemma 11 applied to the submatrix, we have

$$\mathbb{P}\left( \frac{N}{N_r} \|\boldsymbol{A}_r\|_F^2 \geq \frac{Cd}{nN} \right) \leq C'\left( ne^{-c\sqrt{n}} + \frac{1}{d} \right). \tag{D.7}$$

Moreover, by definition of $\mathcal{A}_r^\alpha$, all $a_{i_r}$'s are close to $\frac{\alpha}{\sqrt{N}}$ for $i_r \in \mathcal{A}_r^\alpha$; thus Lemma 22 (in particular (D.2)) can be directly applied to $\boldsymbol{B}_r$. As for $\boldsymbol{C}_r$, since $\|\boldsymbol{C}_r\|_F \leq \|\boldsymbol{C}\|_F$, we use part $(iii)$ in Lemma 10 to obtain

$$\mathbb{P}\left( \|\boldsymbol{C}_r\|_F \geq \frac{C \log n \log N}{N} \right) \leq C'\left( ne^{-c\log^2 n} + Ne^{-c\log^2 N} \right). \tag{D.8}$$

With these concentration estimates, we know that when $n > d$,

$$\mathbb{E}_{\boldsymbol{x}}[(f_{\boldsymbol{A}}(\boldsymbol{x}) - f_r(\boldsymbol{x}))^2] \lesssim \mathbb{E}_{\boldsymbol{x}}\left[ \left( \frac{\eta\sqrt{N}}{N_r} \sum_{i_r \in \mathcal{A}_r^\alpha} |\langle \boldsymbol{\delta}_{i_r}, \boldsymbol{x} \rangle| \right)^2 \right] = \frac{\eta^2 N}{N_r^2} \mathbb{E}_{\boldsymbol{x}}\left[ \sum_{i_r, j_r \in \mathcal{A}_r^\alpha} \left| \boldsymbol{\delta}_{i_r}^\top \boldsymbol{x} \right| \left| \boldsymbol{\delta}_{j_r}^\top \boldsymbol{x} \right| \right]$$

$$\leq \frac{\eta^2 N}{N_r^2} \sum_{i_r, j_r \in \mathcal{A}_r^\alpha} \mathbb{E}_{\boldsymbol{x}}\left[ \left( \boldsymbol{\delta}_{i_r}^\top \boldsymbol{x} \right)^2 \right]^{\frac{1}{2}} \mathbb{E}_{\boldsymbol{x}}\left[ \left( \boldsymbol{\delta}_{j_r}^\top \boldsymbol{x} \right)^2 \right]^{\frac{1}{2}} = \frac{\eta^2 N}{N_r^2} \sum_{i_r, j_r \in \mathcal{A}_r^\alpha} \|\boldsymbol{\delta}_{i_r}\| \|\boldsymbol{\delta}_{j_r}\|$$

$$= \frac{\eta^2 N}{N_r^2} \left( \sum_{i_r \in \mathcal{A}_r^\alpha} \|\boldsymbol{\delta}_{i_r}\| \right)^2 \leq \frac{\eta^2 N}{N_r} \left( \|\boldsymbol{A}_r\|_F^2 + \|\boldsymbol{B}_r\|_F^2 + \|\boldsymbol{C}_r\|_F^2 \right) \lesssim \frac{d}{n} + \frac{1}{d^{\frac{1}{4}-\varepsilon}} + \frac{\log^2 n \log^2 N}{N_r}, \tag{D.9}$$

with probability at least $1 - c\left( \frac{\alpha^2}{d^{\frac{1}{4}}} + \frac{\alpha^4}{n} + \frac{1}{\sqrt{N}} + ne^{-c\log^2 n} + Ne^{-\log^2 N} \right)$ for some constant $c > 0$; this is due to the defined step size $\eta = \Theta(\sqrt{N})$, (D.7), (D.8) in (D.2) of Lemma 22 outlined above. In (D.9), we ignore the constants in the upper bound since we are only interested in the rate with respect to $n, d, N$.

**Simplification under "population" gradient.** Recall the definition of the single-index teacher: $f^*(\boldsymbol{x}) = \sigma^*(\langle \boldsymbol{x}, \boldsymbol{\beta}_* \rangle)$, and the definition of rank-1 matrix $\boldsymbol{A}_1 = \frac{1}{n} \cdot \frac{\mu_1 \mu_1^*}{\sqrt{N}} \boldsymbol{X}^\top \boldsymbol{X} \boldsymbol{\beta}_* \boldsymbol{a}^\top$. Define $\boldsymbol{v} = \frac{\eta \mu_1 \mu_1^*}{n\sqrt{N}} \boldsymbol{X}^\top \boldsymbol{X} \boldsymbol{\beta}_* \in \mathbb{R}^d$, we can write

$$f_{\boldsymbol{A}}(\boldsymbol{x}) = \frac{1}{N_r} \sum_{i_r \in \mathcal{A}_r^\alpha} \sigma\left( \langle \boldsymbol{w}_{i_r} + \sqrt{N} a_{i_r} \boldsymbol{v}, \boldsymbol{x} \rangle \right), \quad \tilde{f}_{\boldsymbol{A}}(\boldsymbol{x}) := \frac{1}{N_r} \sum_{i_r \in \mathcal{A}_r^\alpha} \sigma(\langle \boldsymbol{w}_{i_r} + \alpha \boldsymbol{v}, \boldsymbol{x} \rangle),$$

where we dropped the superscript in the initialized weights $\boldsymbol{w}_{i_r}^0$ to simplify the notation. Note that the difference between $f_{\boldsymbol{A}}$ and $\tilde{f}_{\boldsymbol{A}}$ is that the each second-layer coefficient $a_i$ is replaced by the same scalar $\alpha$. By the definition of $\mathcal{A}_r^\alpha$ and the Lipschitz property of $\sigma$, one can obtain

$$\left| f_{\boldsymbol{A}}(\boldsymbol{x}) - \tilde{f}_{\boldsymbol{A}}(\boldsymbol{x}) \right| \lesssim \frac{1}{N_r} \sum_{i_r \in \mathcal{A}_r^\alpha} \frac{\eta N^{-r}}{\sqrt{N}} \cdot \left| \left\langle \frac{1}{n} \boldsymbol{X}^\top \boldsymbol{X} \boldsymbol{\beta}_*, \boldsymbol{x} \right\rangle \right|$$

$$\lesssim N^{-r} \cdot \left| \left\langle \frac{1}{n} \boldsymbol{X}^\top \boldsymbol{X} \boldsymbol{\beta}_*, \boldsymbol{x} \right\rangle \right|. \tag{D.10}$$

Define $\bar{\boldsymbol{v}} := \frac{\eta \mu_1 \mu_1^*}{\sqrt{N}} \boldsymbol{\beta}_* = \bar{\eta} \mu_1 \mu_1^* \boldsymbol{\beta}_*$, which corresponds to the "population" version of $\boldsymbol{v}$, and denote

$$\bar{f}_{\boldsymbol{A}}(\boldsymbol{x}) := \frac{1}{N_r} \sum_{i_r \in \mathcal{A}_r^\alpha} \sigma(\langle \boldsymbol{w}_{i_r} + \alpha \bar{\boldsymbol{v}}, \boldsymbol{x} \rangle). \tag{D.11}$$

Similar to (D.10), we have

$$\left| \bar{f}_{\boldsymbol{A}}(\boldsymbol{x}) - \tilde{f}_{\boldsymbol{A}}(\boldsymbol{x}) \right| \lesssim \frac{1}{N_r} \sum_{i_r \in \mathcal{A}_r^\alpha} \frac{\eta}{\sqrt{N}} \left| \left\langle \left( \frac{1}{n} \boldsymbol{X}^\top \boldsymbol{X} - \boldsymbol{I} \right) \boldsymbol{\beta}_*, \boldsymbol{x} \right\rangle \right|$$

$$\lesssim \left| \left\langle \left( \frac{1}{n} \boldsymbol{X}^\top \boldsymbol{X} - \boldsymbol{I} \right) \boldsymbol{\beta}_*, \boldsymbol{x} \right\rangle \right|. \tag{D.12}$$

Combining the inequalities (D.10) and (D.12), we know that for some constant $C$,

$$\mathbb{E}_{\boldsymbol{x}}[(f_{\boldsymbol{A}}(\boldsymbol{x}) - \bar{f}_{\boldsymbol{A}}(\boldsymbol{x}))^2] \lesssim N^{-2r} \cdot \mathbb{E}_{\boldsymbol{x}}\left( \left\langle \frac{1}{n} \boldsymbol{X}^\top \boldsymbol{X} \boldsymbol{\beta}_*, \boldsymbol{x} \right\rangle \right)^2 + \mathbb{E}_{\boldsymbol{x}}\left( \left\langle \left( \frac{1}{n} \boldsymbol{X}^\top \boldsymbol{X} - \boldsymbol{I} \right) \boldsymbol{\beta}_*, \boldsymbol{x} \right\rangle \right)^2$$

$$\leq \left( N^{-2r} \left\| \frac{1}{n} \boldsymbol{X}^\top \boldsymbol{X} \right\|^2 + \left\| \frac{1}{n} \boldsymbol{X}^\top \boldsymbol{X} - \boldsymbol{I} \right\|^2 \right) \cdot \| \boldsymbol{\beta}_* \|^2$$

$$\lesssim \left( \left( 1 + \frac{d}{n} \right) N^{-2r} + \frac{d}{n} \right),$$

where the last inequality holds with probability at least $1 - \exp(-cd)$ for some universal constant $c > 0$, due to the operator norm bound and concentration of the sample covariance matrix $\frac{1}{n} \boldsymbol{X}^\top \boldsymbol{X}$ (for instance see [Ver18, Theorem 4.6.1]).

Now we take the expectation of $\bar{f}_{\boldsymbol{A}}$ over initial weight $\boldsymbol{w}_{i_r}$ in (D.11) to define

$$\bar{f}(\boldsymbol{x}) := \mathbb{E}_{\boldsymbol{w} \sim \mathcal{N}(0, d^{-1} \boldsymbol{I})}[\sigma(\langle \boldsymbol{w} + \alpha \bar{\boldsymbol{v}}, \boldsymbol{x} \rangle)].$$

Note that for fixed $\boldsymbol{x}$, $\langle \boldsymbol{w}, \boldsymbol{x} \rangle \sim \mathcal{N}(0, \| \boldsymbol{x} \|^2 / d)$. Since $\sigma$ is $\lambda_\sigma$-Lipschitz, by the Hoeffding bound on sub-Gaussian random variables, conditionally on $\boldsymbol{x}$, we have

$$\mathbb{P}\left( \left| \bar{f}_{\boldsymbol{A}}(\boldsymbol{x}) - \bar{f}(\boldsymbol{x}) \right| > t \,\middle|\, \boldsymbol{x} \right) \leq 2 \exp\left( -\frac{t^2 N_r}{2 \lambda_\sigma^2 \cdot \| \boldsymbol{x} \|_2^2 / d} \right), \tag{D.13}$$

Also notice that

$$\mathbb{E}_{\boldsymbol{w}}(\bar{f}(\boldsymbol{x}) - \bar{f}_{\boldsymbol{A}}(\boldsymbol{x}))^2 = \int_0^\infty \mathbb{P}\left( \left| \bar{f}_{\boldsymbol{A}}(\boldsymbol{x}) - \bar{f}(\boldsymbol{x}) \right|^2 > t \,\middle|\, \boldsymbol{x} \right) \mathrm{d}t$$

$$\leq \int_0^\infty 2 \exp\left( -\frac{t N_r}{2 \lambda_\sigma^2 \cdot \| \boldsymbol{x} \|_2^2 / d} \right) \mathrm{d}t = \frac{4 \lambda_\sigma^2 \| \boldsymbol{x} \|^2}{N_r d}.$$

Thus, by taking expectation over $\boldsymbol{x}$ in the above bound, we know that $\mathbb{E}(\bar{f}(\boldsymbol{x}) - \bar{f}_{\boldsymbol{A}}(\boldsymbol{x}))^2 \leq \frac{4 \lambda_\sigma^2 \mathbb{E}[\| \boldsymbol{x} \|^2]}{N_r d} = \frac{4 \lambda_\sigma^2}{N_r}$. By Markov's inequality, we have

$$\mathbb{P}\left( \mathbb{E}_{\boldsymbol{x}}(\bar{f}(\boldsymbol{x}) - \bar{f}_{\boldsymbol{A}}(\boldsymbol{x}))^2 \geq t \right) \leq \frac{\mathbb{E}(\bar{f}(\boldsymbol{x}) - \bar{f}_{\boldsymbol{A}}(\boldsymbol{x}))^2}{t} \leq \frac{4 \lambda_\sigma^2}{N_r t}. \tag{D.14}$$

Hence we deduce that $\mathbb{E}_{\boldsymbol{x}}(\bar{f}(\boldsymbol{x}) - \bar{f}_{\boldsymbol{A}}(\boldsymbol{x}))^2 \leq \frac{4 \lambda_\sigma^2}{\sqrt{N_r}}$ with probability $1 - \frac{1}{\sqrt{N_r}}$.

Observe that $\bar{f}$ is given by an expectation over $\boldsymbol{w}$ in a single-index model. To calculate its difference from the true model: $\mathbb{E}_{\boldsymbol{x}}(\bar{f}(\boldsymbol{x}) - f^*(\boldsymbol{x}))^2$, first recall the assumption that $\| \boldsymbol{\beta}_* \| = 1$, and $\boldsymbol{w} \sim \mathcal{N}(0, \boldsymbol{I}/d)$, $\boldsymbol{x} \sim \mathcal{N}(0, \boldsymbol{I})$. Denote $\xi_1 := \langle \boldsymbol{x}, \boldsymbol{\beta}_* \rangle \sim \mathcal{N}(0, 1)$ and, condition on $\boldsymbol{x}$, $\langle \boldsymbol{x}, \boldsymbol{w} \rangle \overset{d}{=} \xi_2 \| \boldsymbol{x} \| / \sqrt{d}$, where $\xi_2 \sim \mathcal{N}(0, 1)$ independent of $\xi_1$. Since $\eta / \sqrt{N} = \bar{\eta}$, we can write $\kappa := \frac{\alpha \eta \mu_1 \mu_1^*}{\sqrt{N}} = \alpha \bar{\eta} \mu_1 \mu_1^* \in \mathbb{R}$. Following these definitions, we have $\bar{f}(\boldsymbol{x}) = \mathbb{E}_{\xi_2}[\sigma(\xi_2 \| \boldsymbol{x} \| / \sqrt{d} + \kappa \xi_1)]$, and

$$\mathbb{E}_{\boldsymbol{x}}(\bar{f}(\boldsymbol{x}) - f^*(\boldsymbol{x}))^2 = \mathbb{E}_{\xi_1}\left( \sigma^*(\xi_1) - \mathbb{E}_{\xi_2}[\sigma(\kappa \xi_1 + \xi_2 \| \boldsymbol{x} \| / \sqrt{d})] \right)^2. \tag{D.15}$$

In addition, given $\kappa \in \mathbb{R}$, we introduce a scalar quantity

$$\tau := \mathbb{E}_{\xi_1}\left( \sigma^*(\xi_1) - \mathbb{E}_{\xi_2}[\sigma(\kappa \xi_1 + \xi_2)] \right)^2. \tag{D.16}$$

Note that $\sigma^* \in L^2(\mathbb{R}, \Gamma)$ and $\sigma$ is uniformly bounded by assumption; one can easily check that $\tau$ is uniformly bounded for all $\kappa \in \mathbb{R}$. Hence $\tau$ defined above is always finite. We now show that the difference between $\tau$ and $\mathbb{E}_{\boldsymbol{x}}(\bar{f}(\boldsymbol{x}) - f^*(\boldsymbol{x}))^2$ is asymptotically negligible, again using the Lipschitz property of $\sigma$,

$$\left| \sigma(\kappa \xi_1 + \xi_2) - \sigma(\kappa \xi_1 + \xi_2 \|\boldsymbol{x}\|/\sqrt{d}) \right| \lesssim \left| 1 - \frac{\|\boldsymbol{x}\|}{\sqrt{d}} \right| \cdot |\xi_2|.$$

Since $\sigma^* \in L^2(\mathbb{R}, \Gamma)$, and $\sigma$ is uniformly bounded and Lipschitz, based on (D.15) and (D.16), we can apply the Cauchy-Schwarz inequality to get

$$\left| \tau - \mathbb{E}_{\boldsymbol{x}}(\bar{f}(\boldsymbol{x}) - f^*(\boldsymbol{x}))^2 \right|$$
$$\lesssim \mathbb{E}\left[ \left( \sigma(\kappa \xi_1 + \xi_2) - \sigma(\kappa \xi_1 + \xi_2 \|\boldsymbol{x}\|/\sqrt{d}) \right)^2 \right]^{\frac{1}{2}}$$
$$\lesssim \mathbb{E}[\xi_2^2]^{\frac{1}{2}} \mathbb{E}\left[ \left| 1 - \frac{\|\boldsymbol{x}\|}{\sqrt{d}} \right|^2 \right]^{\frac{1}{2}} \leq \frac{C}{\sqrt{2d}}, \tag{D.17}$$

where the last inequality is due to property of the sub-Gaussian norm $\|\|\boldsymbol{x}\|/\sqrt{d} - 1\|_{\psi_2} \leq C/\sqrt{d}$ (see e.g. [Ver18, Theorem 3.1.1]) for some universal constant $C > 0$.

**Proof of Lemma 23.** Based on above calculations, we now control the prediction risk of $\tilde{f}$ by combining the substitution errors, where $\tilde{f} = f_r$ is constructed as the average over subset $\mathcal{A}_r^\alpha$ defined in (D.6).

Given any $\alpha \in \mathbb{R}$ and $r \in (0, 1/2)$, we define the subset $\mathcal{A}_r^\alpha$ and the corresponding $\tilde{f}(\boldsymbol{x}) = f_r(\boldsymbol{x}) = \frac{1}{\sqrt{N}} \tilde{\boldsymbol{a}}^\top \sigma(\boldsymbol{W}_1^\top \boldsymbol{x})$, where the second-layer $\tilde{\boldsymbol{a}}$ is given as $[\tilde{\boldsymbol{a}}]_i = \sqrt{N}/N_r$ if $i \in \mathcal{A}_r^\alpha$, otherwise $[\tilde{\boldsymbol{a}}]_i = 0$. Moreover, (D.5) implies that $N_r = \Theta_{d,\mathbb{P}}(N^{1-r})$ with probability at least $1 - \exp(-\log^2 N)$. Therefore, together with (D.9), (D.13), (D.14), and (D.17), we know that

$$\mathbb{E}_{\boldsymbol{x}}(f_r(\boldsymbol{x}) - \bar{f}(\boldsymbol{x}))^2 \leq \frac{Cd}{n} + o_{d,\mathbb{P}}(1); \quad \mathbb{E}_{\boldsymbol{x}}(f^*(\boldsymbol{x}) - \bar{f}(\boldsymbol{x}))^2 = \tau + o_d(1),$$

as $n, d, N \to \infty$, for some constant $C > 0$. By the Cauchy-Schwarz inequality,

$$\mathbb{E}_{\boldsymbol{x}}(f^*(\boldsymbol{x}) - f_r(\boldsymbol{x}))^2 \leq \tau + C\left( \sqrt{\tau} \cdot \sqrt{\frac{d}{n}} + \frac{d}{n} \right) + o_{d,\mathbb{P}}(1),$$

where the failure probability only relates to $r, \alpha, N, d, n$ and vanishes as $N, d, n \to \infty$. For simplicity, we only keep the leading orders and ignore the subordinate terms in the probability bounds.

Note that the above characterization holds for any finite $\alpha$; since our goal is to construct an estimator $f_r$ that achieves as small prediction risk as possible, we optimize over $\alpha \in \mathbb{R}$ by defining

$$\tau^* := \inf_{\alpha \in \mathbb{R}} \mathbb{E}_{\xi_1}\left( \sigma^*(\xi_1) - \mathbb{E}_{\xi_2}\left( \sigma\left( \alpha \bar{\eta} \mu_1 \mu_1^* \cdot \xi_1 + \xi_2 \right) \right) \right)^2, \quad \tau_\varepsilon^* := \tau^* + \varepsilon,$$

where $\varepsilon \geq 0$ is a small constant. This definition of $\tau^*$ is identical to (5.1) and is always finite because $\tau$ defined in (D.16) is uniformly bounded and non-negative (observe that optimizing over $\kappa$ or $\alpha \in \mathbb{R}$ are equivalent, since we can reparameterize $\kappa = \alpha \bar{\eta} \mu_1 \mu_1^*$ where $\mu_1, \mu_1^* \neq 0$). When $\tau^*$ is attained at some finite $\alpha$, then we may simply set $\varepsilon = 0$ and define

$$\alpha^* := \underset{\alpha \in \mathbb{R}}{\operatorname{argmin}} \, \mathbb{E}_{\xi_1}\left( \sigma^*(\xi_1) - \mathbb{E}_{\xi_2}\left( \sigma\left( \alpha \bar{\eta} \mu_1 \mu_1^* \cdot \xi_1 + \xi_2 \right) \right) \right)^2.$$

Otherwise, observe that as a bounded and continuous function of $\alpha$ on the real line, $\tau(\alpha) := \mathbb{E}_{\xi_1}\left( \sigma^*(\xi_1) - \mathbb{E}_{\xi_2}\left( \sigma\left( \alpha \bar{\eta} \mu_1 \mu_1^* \cdot \xi_1 + \xi_2 \right) \right) \right)^2$ will approach its minimum at infinity. Therefore, in this case, for any $\varepsilon > 0$, we can find some finite $\alpha_\varepsilon^*$ such that $\tau(\alpha_\varepsilon^*) \leq \tau_\varepsilon^* = \tau^* + \varepsilon$; hence, we may set $\alpha = \alpha_\varepsilon^*$ and conclude the proof. Finally, note that given nonlinearities $\sigma$ and $\sigma^*$ (which determine the relation between $\varepsilon$ and $\alpha_\varepsilon$), we can take $\varepsilon \to 0$ at a slow enough rate as $n, d, N \to \infty$, as long as $C(\alpha) \cdot N^{1-r} \to \infty$. Thus we also obtain an asymptotic version of Lemma 23: with

probability one, there exists some second-layer $\tilde{\boldsymbol{a}}$ such that the prediction risk of the corresponding student model $\tilde{f}(\boldsymbol{x}) = \frac{1}{\sqrt{N}}\tilde{\boldsymbol{a}}^\top \sigma(\boldsymbol{W}_1^\top \boldsymbol{x})$ satisfies

$$\mathcal{R}(\tilde{f}) \leq \tau^* + C\left(\sqrt{\tau^*} \cdot \sqrt{\frac{d}{n}} + \frac{d}{n}\right), \tag{D.18}$$

for some constant $C > 0$, as $n, d, N \to \infty$ proportionally.

$\square$

The above analysis illustrates that because of the Gaussian initialization of $a_i$, for any $\eta = \Theta(\sqrt{N})$, we can find a subset of neurons $\mathcal{A}_r^\alpha$ that receive a "good" learning rate, in the sense that the corresponding (sub-) network defined by $f_r$ can achieve the prediction risk close to $\tau_\varepsilon^*$ when $n \gg d$.

**Some examples.** Equation (D.3) reduces the prediction risk of our constructed $f_r$ to a one-dimensional Gaussian integral, which can be numerically evaluated for pairs of $(\sigma, \sigma^*)$. Denote $\kappa^* = \alpha^* \bar{\eta} \mu_1 \mu_1^*$, we give a few examples in which we set $\varepsilon = 0$ and the corresponding $\tau^*$ is small. Note that due to Assumption 1, choices of $\sigma$ and $\sigma^*$ considered below are centered with respect to standard Gaussian measure $\Gamma$.

- $\underline{\sigma = \sigma^* = \mathrm{erf}}$. Note that for $c_1, c_2 \in \mathbb{R}$, $\mathbb{E}_{z \sim \mathcal{N}(0,1)}[\mathrm{erf}(c_1 z + c_2)] = \mathrm{erf}\left(\frac{c_2}{\sqrt{1+2c_1^2}}\right)$. Hence we can choose $\kappa^* = \sqrt{3}$, and the corresponding minimum value $\tau^* = 0$.
- $\underline{\sigma = \sigma^* = \tanh}$. Numerical integration yields $\tau^* \approx 3 \times 10^{-4}$, $\kappa^* \approx 1.6$.
- $\underline{\sigma = \sigma^* = \mathrm{SoftPlus}}$. Numerical integration yields $\tau^* \approx 0.03$, $\kappa^* \approx 0.96$.
- $\underline{\sigma = \mathrm{ReLU}, \sigma^* = \mathrm{SoftPlus}}$. Numerical integration yields $\tau^* \approx 0.09$, $\kappa^* \approx 0.94$.

Observe that in all the above examples, $\tau^*$ can be obtained by some finite $\alpha^*$ (or equivalently $\kappa^*$). In the following analysis of kernel ridge regression, we drop the small constant $\varepsilon$ in Lemma 23 and directly apply the asymptotic statement given in (D.18).

**Remark.** *We make the following remarks on the calculation of $\tau^*$ in (5.1).*

- *When $\sigma = \sigma^*$, we intuitively expect $\tau^*$ to be small when the nonlinearity is smooth such that it is to some extent unchanged under Gaussian convolution (when $\kappa$ is chosen appropriately).*

- *Adding weight decay with strength $\lambda < 1$ to the first-layer parameters $\boldsymbol{W}_0$ simply corresponds to multiplying $\xi_2$ in the definition of $\tau$ (D.16) by a factor of $(1 - \lambda)$.*

### D.3 Prediction risk of ridge regression

In this section we prove Theorem 7. Recall that we aim to upper-bound the prediction risk of the CK ridge regression estimator defined as

$$\hat{f}(\boldsymbol{x}) = \left\langle \frac{1}{\sqrt{N}}\sigma(\boldsymbol{W}_1^\top \boldsymbol{x}), \hat{\boldsymbol{a}} \right\rangle, \quad \text{where } \hat{\boldsymbol{a}} := \left(\boldsymbol{\Phi}^\top \boldsymbol{\Phi} + \lambda n \boldsymbol{I}\right)^{-1} \boldsymbol{\Phi}^\top \tilde{\boldsymbol{y}}, \quad \boldsymbol{\Phi} := \frac{1}{\sqrt{N}}\sigma(\tilde{\boldsymbol{X}}\boldsymbol{W}_1), \tag{D.19}$$

where $\{\tilde{\boldsymbol{X}}, \tilde{\boldsymbol{y}}\}$ is a new set of training data independent of $\boldsymbol{W}$. For concise notation, in this section we rescale the ridge parameter in (4.1) by replacing $\frac{\lambda}{N}$ with $\lambda$.

Given feature map $\boldsymbol{x} \to \frac{1}{\sqrt{N}}\sigma(\boldsymbol{W}_1^\top \boldsymbol{x})$ conditioned on first layer weights $\boldsymbol{W}_1$, we denote the associated Hilbert space as $\mathcal{H}$. Note that $\mathcal{H}$ is a finite-dimensional reproducing kernel Hilbert space and is hence closed; we define the optimal predictor in the RKHS as $\check{f} := \mathrm{argmin}_{f \in \mathcal{H}} \mathbb{E}_{\boldsymbol{x}}(f(\boldsymbol{x}) - f^*(\boldsymbol{x}))^2$, which takes the form of $\check{f}(\boldsymbol{x}) = \langle \frac{1}{\sqrt{N}}\sigma(\boldsymbol{W}_1^\top \boldsymbol{x}), \check{\boldsymbol{a}}\rangle$ for some $\check{\boldsymbol{a}} \in \mathbb{R}^N$. In addition, we may write the orthogonal decomposition in $L^2(\mathbb{R}^d, \Gamma)$: $f^*(\boldsymbol{x}) = \check{f}(\boldsymbol{x}) + f_\perp(\boldsymbol{x})$. By definition of $f_\perp$, we have $\|f_\perp\|_{L^2}^2 = \mathbb{E}_{\boldsymbol{x}}[f_\perp(\boldsymbol{x})^2] \leq \mathcal{R}(h) = \|f^* - h\|_{L^2}^2$, for any $h \in \mathcal{H}$ and $\boldsymbol{x} \sim \mathcal{N}(0, 1)$. Finally, from Assumption 1 we know that $\|f^*\|_{L^2}$ is bounded by some constant, and thus $\|f_\perp\|_{L^2}$ is also bounded.

We are interested in the prediction risk of the CK ridge regression estimator denoted as $\mathcal{R}_1(\lambda)$. We first define the following quantities which $\mathcal{R}_1(\lambda)$ can be decomposed into (see Lemma 25):

$$
\begin{aligned}
B_1 &:= \mathbb{E}_{\boldsymbol{x}}\big(f^*(\boldsymbol{x}) - \check{f}(\boldsymbol{x})\big)^2, \\
B_2 &:= \mathbb{E}_{\boldsymbol{x}}\left(\check{f}(\boldsymbol{x}) - \frac{1}{n}\phi_x\big(\widehat{\boldsymbol{\Sigma}}_\Phi + \lambda \boldsymbol{I}\big)^{-1}\boldsymbol{\Phi}^\top \check{\boldsymbol{f}}\right)^2, \\
V_1 &:= \frac{1}{n^2}\tilde{\varepsilon}^\top \boldsymbol{\Phi}\big(\widehat{\boldsymbol{\Sigma}}_\Phi + \lambda \boldsymbol{I}\big)^{-1}\boldsymbol{\Sigma}_\Phi\big(\widehat{\boldsymbol{\Sigma}}_\Phi + \lambda \boldsymbol{I}\big)^{-1}\boldsymbol{\Phi}^\top \tilde{\varepsilon}, \\
V_2 &:= \frac{1}{n^2}\boldsymbol{f}_\perp^\top \boldsymbol{\Phi}\big(\widehat{\boldsymbol{\Sigma}}_\Phi + \lambda \boldsymbol{I}\big)^{-1}\boldsymbol{\Sigma}_\Phi\big(\widehat{\boldsymbol{\Sigma}}_\Phi + \lambda \boldsymbol{I}\big)^{-1}\boldsymbol{\Phi}^\top \boldsymbol{f}_\perp,
\end{aligned}
\tag{D.20}
$$

where the $i$-th entry of vector $\check{\boldsymbol{f}}$ and $\boldsymbol{f}_\perp$ are given by $[\check{\boldsymbol{f}}]_i = \check{f}(\tilde{\boldsymbol{x}}_i)$, $[\boldsymbol{f}_\perp]_i = f_\perp(\tilde{\boldsymbol{x}}_i)$, respectively, and $\widehat{\boldsymbol{\Sigma}}_\Phi := \frac{1}{n}\boldsymbol{\Phi}^\top\boldsymbol{\Phi}$, $\boldsymbol{\Sigma}_\Phi := \frac{1}{N}\mathbb{E}_{\boldsymbol{x}}\big[\sigma(\boldsymbol{W}_1^\top\boldsymbol{x})\sigma(\boldsymbol{W}_1^\top\boldsymbol{x})^\top\big]$. Also, $\phi_{\boldsymbol{x}} := \frac{1}{\sqrt{N}}\sigma(\boldsymbol{x}^\top\boldsymbol{W}_1)$ for $\boldsymbol{x} \in \mathbb{R}^d$, which gives $\boldsymbol{\Phi}^\top = [\phi_{\tilde{\boldsymbol{x}}_1}^\top, \dots, \phi_{\tilde{\boldsymbol{x}}_i}^\top, \dots, \phi_{\tilde{\boldsymbol{x}}_n}^\top]$, where $\tilde{\boldsymbol{x}}_i^\top$ is the $i$-th row of $\tilde{\boldsymbol{X}}$. To simplify the notation, we omit the accent in $\tilde{\boldsymbol{x}}, \tilde{\varepsilon}$ when the context is clear. In the following subsections, to control $\mathcal{R}_1(\lambda)$, we provide high-probability upper-bounds for $B_1, B_2, V_1$ and $V_2$ separately.

**Concentration of feature covariance.** We begin by defining a concentration event $\mathcal{A}$ on the empirical feature matrix $\widehat{\boldsymbol{\Sigma}}_\Phi$, under which the prediction risk can be controlled. We modify the proof of [Ver18, Theorem 4.7.1] to obtain a normalized version of the concentration for CK matrix, the detailed proof can be found in [BES$^+$22, Appendix D.3].

**Lemma 24.** *Under Assumption 1, and using the above notations, there exists some constant $c > 0$ such that the following holds[4]*

$$
\mathbb{P}\left(\left\|(\boldsymbol{\Sigma}_\Phi + \lambda\boldsymbol{I})^{-1/2}(\boldsymbol{\Sigma}_\Phi - \widehat{\boldsymbol{\Sigma}}_\Phi)(\boldsymbol{\Sigma}_\Phi + \lambda\boldsymbol{I})^{-1/2}\right\| \geq 2K^2 \cdot \sqrt{\frac{N}{n}}\right) \leq 2\exp\left(-c\sqrt{N}\right),
$$

*for all large $n > N$, where $K := \frac{\lambda_\sigma}{\sqrt{N}}\|\boldsymbol{W}_1\|_F$.*

From Lemma 10 we know that when $n, d, N$ are proportional and $\eta = \Theta(\sqrt{N})$, there exist some constants $c, C$ such that $\mathbb{P}\big(\|\boldsymbol{W}_1\|_F \geq C\sqrt{N}\big) \leq \exp(-cN)$. We denote $t = 2C^2 N/n$ and consider sufficiently large $n$ (but still proportional to $d$) such that $t < 1$. Now given fixed $\lambda > 0$, we define the concentration event

$$
\mathcal{A}_\lambda = \left\{-t\boldsymbol{I} \preccurlyeq (\boldsymbol{\Sigma}_\Phi + \lambda\boldsymbol{I})^{-1/2}(\boldsymbol{\Sigma}_\Phi - \widehat{\boldsymbol{\Sigma}}_\Phi)(\boldsymbol{\Sigma}_\Phi + \lambda\boldsymbol{I})^{-1/2} \preccurlyeq t\boldsymbol{I}\right\}.
$$

Similarly, for the "ridgeless" case $\lambda = 0$, we define

$$
\mathcal{A}_0 = \left\{-t\boldsymbol{I} \preccurlyeq \boldsymbol{\Sigma}_\Phi^{-1/2}(\boldsymbol{\Sigma}_\Phi - \widehat{\boldsymbol{\Sigma}}_\Phi)\boldsymbol{\Sigma}_\Phi^{-1/2} \preccurlyeq t\boldsymbol{I}\right\}.
$$

Lemma 24 entails that both $\mathcal{A}_\lambda$ and $\mathcal{A}_0$ hold with probability at least $1 - 2e^{-c\sqrt{N}}$. Following the remark on [Bac23, Lemma 7.1], under events $\mathcal{A}_\lambda$ and $\mathcal{A}_0$, we can obtain that

$$
\left\|\boldsymbol{\Sigma}_\Phi^{-1/2}\big(\widehat{\boldsymbol{\Sigma}}_\Phi - \boldsymbol{\Sigma}_\Phi\big)\boldsymbol{\Sigma}_\Phi^{-1/2}\right\| \leq t,
\tag{D.21}
$$

and $(1 - t)(\boldsymbol{\Sigma}_\Phi - \widehat{\boldsymbol{\Sigma}}_\Phi) \preccurlyeq t(\widehat{\boldsymbol{\Sigma}}_\Phi + \lambda\boldsymbol{I})$, which implies that

$$
\boldsymbol{\Sigma}_\Phi\big(\widehat{\boldsymbol{\Sigma}}_\Phi + \lambda\boldsymbol{I}\big)^{-1} \preccurlyeq \frac{t}{1-t}\boldsymbol{I} + \widehat{\boldsymbol{\Sigma}}_\Phi\big(\widehat{\boldsymbol{\Sigma}}_\Phi + \lambda\boldsymbol{I}\big)^{-1} \preccurlyeq \frac{1}{1-t}\boldsymbol{I},
$$

since $\left\|\widehat{\boldsymbol{\Sigma}}_\Phi\big(\widehat{\boldsymbol{\Sigma}}_\Phi + \lambda\boldsymbol{I}\big)^{-1}\right\| \leq 1$. Analogously, we claim that

$$
\big(\widehat{\boldsymbol{\Sigma}}_\Phi + \lambda\boldsymbol{I}\big)^{-1/2}\boldsymbol{\Sigma}_\Phi\big(\widehat{\boldsymbol{\Sigma}}_\Phi + \lambda\boldsymbol{I}\big)^{-1/2} \preccurlyeq \frac{1}{1-t}\boldsymbol{I}.
$$

Thus, under events $\mathcal{A}_\lambda$ and $\mathcal{A}_0$, we know that

$$
\left\|\boldsymbol{\Sigma}_\Phi^{1/2}\big(\widehat{\boldsymbol{\Sigma}}_\Phi + \lambda\boldsymbol{I}\big)^{-1}\boldsymbol{\Sigma}_\Phi^{1/2}\right\|, \ \left\|\boldsymbol{\Sigma}_\Phi\big(\widehat{\boldsymbol{\Sigma}}_\Phi + \lambda\boldsymbol{I}\big)^{-1}\right\| \leq \frac{1}{1-t}.
\tag{D.22}
$$

We now control $B_1, B_2, V_1, V_2$ under the high probability events $\mathcal{A}_\lambda$ and $\mathcal{A}_0$.

---

[4]Note that for $\lambda = 0$, the LHS of the inequality may be interpreted as a pseudo-inverse.

**Controlling $B_1, B_2$.** By the definition of $\check{f}$, we have

$$B_1 = \inf_{f \in \mathcal{H}} \mathbb{E}_{\boldsymbol{x}}(f^*(\boldsymbol{x}) - f(\boldsymbol{x}))^2 \leq \mathbb{E}_{\boldsymbol{x}}(f^*(\boldsymbol{x}) - f_r(\boldsymbol{x}))^2 = \mathcal{R}(\tilde{f}), \tag{D.23}$$

where $f_r = \tilde{f} \in \mathcal{H}$ is the estimator we constructed in Lemma 23. Note that the upper bound $\mathcal{R}(\tilde{f})$ has already been characterized in (D.18) in the previous subsection.

As for $B_2$, since $\check{\boldsymbol{f}} = \frac{1}{\sqrt{N}}\sigma(\tilde{\boldsymbol{X}}\boldsymbol{W}_1)\check{\boldsymbol{a}}$, simple calculation yields,

$$B_2 = \text{Tr}\left(\left(\boldsymbol{I} - \left(\widehat{\boldsymbol{\Sigma}}_\Phi + \lambda\boldsymbol{I}\right)^{-1}\widehat{\boldsymbol{\Sigma}}_\Phi\right)^\top \boldsymbol{\Sigma}_\Phi\left(\boldsymbol{I} - \left(\widehat{\boldsymbol{\Sigma}}_\Phi + \lambda\boldsymbol{I}\right)^{-1}\widehat{\boldsymbol{\Sigma}}_\Phi\right)\check{\boldsymbol{a}}\check{\boldsymbol{a}}^\top\right)$$

$$= \lambda^2 \left\langle \check{\boldsymbol{a}}, \left(\widehat{\boldsymbol{\Sigma}}_\Phi + \lambda\boldsymbol{I}\right)^{-1}\boldsymbol{\Sigma}_\Phi\left(\widehat{\boldsymbol{\Sigma}}_\Phi + \lambda\boldsymbol{I}\right)^{-1}\check{\boldsymbol{a}}\right\rangle.$$

Following [Bac23, Proposition 7.2], we define $\boldsymbol{a}_\lambda = \boldsymbol{\Sigma}_\Phi(\boldsymbol{\Sigma}_\Phi + \lambda\boldsymbol{I})^{-1}\check{\boldsymbol{a}}$ and obtain

$$B_2 \leq 2\lambda^2 \left\|\boldsymbol{\Sigma}_\Phi^{1/2}(\boldsymbol{\Sigma}_\Phi + \lambda\boldsymbol{I})^{-1}\check{\boldsymbol{a}}\right\|^2 + 2\left\|\boldsymbol{\Sigma}_\Phi^{1/2}\left((\boldsymbol{\Sigma}_\Phi + \lambda\boldsymbol{I})^{-1}\boldsymbol{\Sigma}_\Phi - \left(\widehat{\boldsymbol{\Sigma}}_\Phi + \lambda\boldsymbol{I}\right)^{-1}\widehat{\boldsymbol{\Sigma}}_\Phi\right)\check{\boldsymbol{a}}\right\|^2.$$

In addition, note that

$$\left(\widehat{\boldsymbol{\Sigma}}_\Phi + \lambda\boldsymbol{I}\right)^{-1}\widehat{\boldsymbol{\Sigma}}_\Phi - (\boldsymbol{\Sigma}_\Phi + \lambda\boldsymbol{I})^{-1}\boldsymbol{\Sigma}_\Phi$$

$$= \left(\widehat{\boldsymbol{\Sigma}}_\Phi + \lambda\boldsymbol{I}\right)^{-1}\left(\widehat{\boldsymbol{\Sigma}}_\Phi - \boldsymbol{\Sigma}_\Phi\right) + \left[\left(\widehat{\boldsymbol{\Sigma}}_\Phi + \lambda\boldsymbol{I}\right)^{-1} - (\boldsymbol{\Sigma}_\Phi + \lambda\boldsymbol{I})^{-1}\right]\boldsymbol{\Sigma}_\Phi$$

$$= \lambda\left(\widehat{\boldsymbol{\Sigma}}_\Phi + \lambda\boldsymbol{I}\right)^{-1}\left(\widehat{\boldsymbol{\Sigma}}_\Phi - \boldsymbol{\Sigma}_\Phi\right)(\boldsymbol{\Sigma}_\Phi + \lambda\boldsymbol{I})^{-1}.$$

Therefore, we know that under events $\mathcal{A}_\lambda$ and $\mathcal{A}_0$,

$$\left\|\boldsymbol{\Sigma}_\Phi^{1/2}\left((\boldsymbol{\Sigma}_\Phi + \lambda\boldsymbol{I})^{-1}\boldsymbol{\Sigma}_\Phi - \left(\widehat{\boldsymbol{\Sigma}}_\Phi + \lambda\boldsymbol{I}\right)^{-1}\widehat{\boldsymbol{\Sigma}}_\Phi\right)\check{\boldsymbol{a}}\right\|^2$$

$$= \lambda^2\left\|\boldsymbol{\Sigma}_\Phi^{1/2}\left(\widehat{\boldsymbol{\Sigma}}_\Phi + \lambda\boldsymbol{I}\right)^{-1}\left(\widehat{\boldsymbol{\Sigma}}_\Phi - \boldsymbol{\Sigma}_\Phi\right)(\boldsymbol{\Sigma}_\Phi + \lambda\boldsymbol{I})^{-1}\check{\boldsymbol{a}}\right\|^2$$

$$\leq \left\|\boldsymbol{\Sigma}_\Phi^{1/2}\left(\widehat{\boldsymbol{\Sigma}}_\Phi + \lambda\boldsymbol{I}\right)^{-1}\boldsymbol{\Sigma}_\Phi^{1/2}\right\|^2 \cdot \left\|\boldsymbol{\Sigma}_\Phi^{-1/2}\left(\widehat{\boldsymbol{\Sigma}}_\Phi - \boldsymbol{\Sigma}_\Phi\right)\boldsymbol{\Sigma}_\Phi^{-1/2}\right\|^2 \cdot \lambda^2\left\|\boldsymbol{\Sigma}_\Phi^{1/2}(\boldsymbol{\Sigma}_\Phi + \lambda\boldsymbol{I})^{-1}\check{\boldsymbol{a}}\right\|^2$$

$$\overset{(i)}{\leq} \frac{t^2}{(1-t)^2} \cdot \lambda^2\left\|\boldsymbol{\Sigma}_\Phi^{1/2}(\boldsymbol{\Sigma}_\Phi + \lambda\boldsymbol{I})^{-1}\check{\boldsymbol{a}}\right\|^2,$$

where $(i)$ follows from the definition of the concentration events $\mathcal{A}$, (D.21) and (D.22).

Finally, from [Bac23, Lemma 7.2], we have

$$\lambda^2\left\|\boldsymbol{\Sigma}_\Phi^{1/2}(\boldsymbol{\Sigma}_\Phi + \lambda\boldsymbol{I})^{-1}\check{\boldsymbol{a}}\right\|^2 \leq \lambda\langle\check{\boldsymbol{a}}, (\boldsymbol{\Sigma}_\Phi + \lambda\boldsymbol{I})^{-1}\boldsymbol{\Sigma}_\Phi\check{\boldsymbol{a}}\rangle$$

$$= \inf_{f \in \mathcal{H}}\left\{\|f - \check{f}\|_{L^2}^2 + \lambda\|f\|_{\mathcal{H}}^2\right\} \leq 2\|f^* - f_r\|_{L^2}^2 + \lambda\|f_r\|_{\mathcal{H}}^2, \tag{D.24}$$

where the last step is a triangle inequality due to $\|f^* - \check{f}\|_{L^2}^2 \leq \|f^* - f_r\|_{L^2}^2$.

**Controlling $V_1, V_2$.** For $V_1$, note that under event $\mathcal{A}_\lambda$,

$$V_1 = \frac{1}{n^2}\tilde{\boldsymbol{\varepsilon}}^\top\boldsymbol{\Phi}\left(\widehat{\boldsymbol{\Sigma}}_\Phi + \lambda\boldsymbol{I}\right)^{-1}\boldsymbol{\Sigma}_\Phi\left(\widehat{\boldsymbol{\Sigma}}_\Phi + \lambda\boldsymbol{I}\right)^{-1}\boldsymbol{\Phi}^\top\tilde{\boldsymbol{\varepsilon}}$$

$$\leq \left\|\frac{1}{n}\boldsymbol{\varepsilon}^\top\boldsymbol{\Phi}\right\|^2 \cdot \left\|\left(\widehat{\boldsymbol{\Sigma}}_\Phi + \lambda\boldsymbol{I}\right)^{-1}\boldsymbol{\Sigma}_\Phi\right\| \cdot \left\|\left(\widehat{\boldsymbol{\Sigma}}_\Phi + \lambda\boldsymbol{I}\right)^{-1}\right\| \overset{(ii)}{\lesssim} \frac{1}{\lambda(1-t)} \cdot \left\|\frac{1}{n}\boldsymbol{\Phi}^\top\tilde{\boldsymbol{\varepsilon}}\right\|^2,$$

where $\boldsymbol{\Phi}$ is defined in (D.19), and $(ii)$ is based on the concentration property for $\mathcal{A}_\lambda$ given in (D.22). Denote $\chi_k^i := [\phi_k]_i \cdot \tilde{\varepsilon}_k$ whence $\left[\boldsymbol{\Phi}^\top\tilde{\boldsymbol{\varepsilon}}\right]_i = \sum_{k=1}^n \chi_k^i$. Note that $\mathbb{E}[\chi_k^i\chi_k^j] = 0$, $\mathbb{E}[(\chi_k^i)^2] \lesssim \frac{\sigma_\varepsilon^2}{N}$

for any $k \in [n]$ and $i \neq j \in [N]$, due to the assumptions on label noise and bounded activation $\sigma$. Therefore, by Markov's inequality, for any $x > 0$, we have

$$\mathbb{P}\left(\frac{1}{n^2}\|\boldsymbol{\Phi}^\top\tilde{\varepsilon}\|^2 \geq x\right) \leq \frac{\mathbb{E}\|\boldsymbol{\Phi}^\top\tilde{\varepsilon}\|^2}{n^2 x} \lesssim \frac{\sigma_\varepsilon^2}{nx}. \tag{D.25}$$

Similarly for $V_2$, under event $\mathcal{A}_\lambda$, we have

$$V_2 = \frac{1}{n^2}\boldsymbol{f}_\perp^\top\boldsymbol{\Phi}\left(\widehat{\boldsymbol{\Sigma}}_\Phi + \lambda\boldsymbol{I}\right)^{-1}\boldsymbol{\Sigma}_\Phi\left(\widehat{\boldsymbol{\Sigma}}_\Phi + \lambda\boldsymbol{I}\right)^{-1}\boldsymbol{\Phi}^\top\boldsymbol{f}_\perp$$

$$\leq \left\|\frac{1}{n}\boldsymbol{f}_\perp^\top\boldsymbol{\Phi}\right\|^2 \cdot \left\|\left(\widehat{\boldsymbol{\Sigma}}_\Phi + \lambda\boldsymbol{I}\right)^{-1}\boldsymbol{\Sigma}_\Phi\right\| \cdot \left\|\left(\widehat{\boldsymbol{\Sigma}}_\Phi + \lambda\boldsymbol{I}\right)^{-1}\right\| \lesssim \frac{1}{\lambda(1-t)} \cdot \left\|\frac{1}{n}\boldsymbol{\Phi}^\top\boldsymbol{f}_\perp\right\|^2.$$

Recall that $\mathbb{E}[\boldsymbol{\phi}_{\boldsymbol{x}}f_\perp(\boldsymbol{x})] = \boldsymbol{0}$ due to the orthogonality condition. Hence we may apply the exact same argument as $V_1$ to obtain an upper bound similar to (D.25); by Markov's inequality,

$$\mathbb{P}\left(\frac{1}{n^2}\|\boldsymbol{\Phi}^\top\boldsymbol{f}_\perp\|^2 \geq x\right) \leq \frac{\mathbb{E}\|\boldsymbol{\Phi}^\top\boldsymbol{f}_\perp\|^2}{n^2 x} \overset{(iii)}{\lesssim} \frac{\|f_\perp\|_{L^2}^2}{nx}, \tag{D.26}$$

where $(iii)$ is due to the boundedness of $\sigma$ and $\|f_\perp\|_{L^2}$. Combining $V_1$ and $V_2$, and taking $x = Cn^{\varepsilon-1}$ in (D.25) and (D.26), for some $C > 0$ and any small $\varepsilon > 0$, we arrive at

$$V_1 + V_2 \lesssim \frac{\sigma_\varepsilon^2 + \|f_\perp\|_{L^2}^2}{n^{1-\varepsilon}\lambda(1-t)}, \tag{D.27}$$

with probability at least $1 - n^{-\varepsilon}$.

**Putting things together.** The following lemma provides a decomposition of the prediction risk $\mathcal{R}_1(\lambda)$ in terms of $B_1, B_2, V_1, V_2$ analyzed above.

**Lemma 25.** *Under the same assumptions as Lemma 6, if we choose $\lambda = \Omega(n^{\varepsilon-1})$ for small $\varepsilon > 0$, then the prediction risk of the CK ridge estimator admits the following upper bound*

$$\mathcal{R}_1(\lambda) \leq B_1 + B_2 + 2\sqrt{B_1 B_2} + o_{d,\mathbb{P}}(1),$$

*where $B_1, B_2$ are defined in* (D.20).

**Proof.** Based on the definition of prediction risk, we have

$$\mathcal{R}_1(\lambda) = \mathbb{E}_{\boldsymbol{x}}\left((f^*(\boldsymbol{x}) - \check{f}(\boldsymbol{x})) + \left(\check{f}(\boldsymbol{x}) - \frac{1}{n}\boldsymbol{\phi}_{\boldsymbol{x}}\left(\widehat{\boldsymbol{\Sigma}}_\Phi + \lambda\boldsymbol{I}\right)^{-1}\boldsymbol{\Phi}^\top\tilde{\boldsymbol{y}}\right)\right)^2$$

$$\leq \underbrace{\mathbb{E}_{\boldsymbol{x}}\left(f^*(\boldsymbol{x}) - \check{f}(\boldsymbol{x})\right)^2}_{B_1} + 2\sqrt{B_1 S_1} + S_1,$$

where we defined

$$S_1 := \mathbb{E}_{\boldsymbol{x}}\left(\check{f}(\boldsymbol{x}) - \frac{1}{n}\boldsymbol{\phi}_{\boldsymbol{x}}\left(\widehat{\boldsymbol{\Sigma}}_\Phi + \lambda\boldsymbol{I}\right)^{-1}\boldsymbol{\Phi}^\top(\check{\boldsymbol{f}} + \boldsymbol{f}_\perp + \tilde{\varepsilon})\right)^2$$

$$\leq \underbrace{\mathbb{E}_{\boldsymbol{x}}\left(\check{f}(\boldsymbol{x}) - \frac{1}{n}\boldsymbol{\phi}_{\boldsymbol{x}}\left(\widehat{\boldsymbol{\Sigma}}_\Phi + \lambda\boldsymbol{I}\right)^{-1}\boldsymbol{\Phi}^\top\check{\boldsymbol{f}}\right)^2}_{B_2} + 2\sqrt{B_2 S_2} + S_2,$$

in which

$$S_2 := \underbrace{\frac{1}{n^2}\tilde{\varepsilon}^\top\boldsymbol{\Phi}\left(\widehat{\boldsymbol{\Sigma}}_\Phi + \lambda\boldsymbol{I}\right)^{-1}\boldsymbol{\Sigma}_\Phi\left(\widehat{\boldsymbol{\Sigma}}_\Phi + \lambda\boldsymbol{I}\right)^{-1}\boldsymbol{\Phi}^\top\tilde{\varepsilon} + \frac{1}{n^2}\boldsymbol{f}_\perp^\top\boldsymbol{\Phi}\left(\widehat{\boldsymbol{\Sigma}}_\Phi + \lambda\boldsymbol{I}\right)^{-1}\boldsymbol{\Sigma}_\Phi\left(\widehat{\boldsymbol{\Sigma}}_\Phi + \lambda\boldsymbol{I}\right)^{-1}\boldsymbol{\Phi}^\top\boldsymbol{f}_\perp}_{V_1 \qquad\qquad\qquad\qquad\qquad\qquad\qquad\qquad\qquad\qquad V_2}$$

$$+ \frac{2}{n^2}\boldsymbol{f}_\perp^\top\boldsymbol{\Phi}\left(\widehat{\boldsymbol{\Sigma}}_\Phi + \lambda\boldsymbol{I}\right)^{-1}\boldsymbol{\Sigma}_\Phi\left(\widehat{\boldsymbol{\Sigma}}_\Phi + \lambda\boldsymbol{I}\right)^{-1}\boldsymbol{\Phi}^\top\tilde{\varepsilon}$$

$$\leq 2(V_1 + V_2).$$

Recall that Lemma 24 entails that events $\mathcal{A}_\lambda$ and $\mathcal{A}_0$ occur with high probability, for constant $t \in (0,1)$. Hence from (D.27) we know that for $\lambda = \Omega(n^{\varepsilon-1})$ with small $\varepsilon > 0$, $V_1 + V_2 = o_{d,\mathbb{P}}(1)$, and thus $S_2$ is vanishing when $n, d, N \to \infty$ proportionally. On the other hand, (D.23) and (D.24) entail that $B_1$ and $B_2$ are both finite. The claim is established by combining the calculations.

$\square$

**Proof of Theorem 7.** Since Lemma 24 ensures that events $\mathcal{A}_\lambda$ and $\mathcal{A}_0$ happens with high probability for fixed $t \in (0,1)$, if we set $\lambda = \Omega(n^{\varepsilon-1})$ for some small $\varepsilon > 0$, then Lemma 25 entails

$$\mathcal{R}_1(\lambda) \leq B_1 + B_2 + 2\sqrt{B_1 B_2} + o_{d,\mathbb{P}}(1),$$

$$\text{where } B_1 \leq \|f^* - f_r\|_{L^2}^2, \quad B_2 \leq 2\left(1 + \frac{t^2}{(1-t)^2}\right) \cdot \left(2\|f^* - f_r\|_{L^2}^2 + \lambda\|f_r\|_{\mathcal{H}}^2\right) + o_{d,\mathbb{P}}(1),$$

in which $f_r$ is defined by (D.6) in the proof of Lemma 23. Here, we applied the upper bounds on $B_1$ in (D.23) and $B_2$ in (D.24). Since $\|f^* - f_r\|_{L^2}^2 = \mathcal{R}(\tilde{f})$, by (D.18) we know that as $n, d, N \to \infty$, with probability one,

$$\|f^* - f_r\|_{L^2}^2 \leq \tau^* + C\left(\sqrt{\tau^*} \cdot \sqrt{\tfrac{d}{n}} + \tfrac{d}{n}\right), \tag{D.28}$$

for some constant $C > 0$. Finally, recall that in the proof of Lemma 23, we constructed an estimator $f_r \in \mathcal{H}$ with $\|f_r\|_{\mathcal{H}}^2 = \|\tilde{a}\|^2 = N/|\mathcal{A}_r^\alpha| = \Theta_{d,\mathbb{P}}(N^r)$, for $0 < r < 1/2$. In other words, $\lambda\|f_r\|_{\mathcal{H}}^2 = o_{d,\mathbb{P}}(1)$ as long as $N^r\lambda \to 0$ as $n, d, N \to \infty$; this provides a way to choose $r \in (0, 1/2)$ given $\lambda$. Now from Lemma 24 we know that there exists some constant $\psi_1^*$ such that both $\mathcal{A}_\lambda$ and $\mathcal{A}_0$ hold with high probability for $t < 0.1$ when $n/d > \psi_1^*$. In this case, given any $\lambda = n^{-\rho}$ for some $\rho \in (0,1)$, we know that

$$\mathcal{R}_1(\lambda) \leq B_1 + B_2 + 2\sqrt{B_1 B_2} + o_{d,\mathbb{P}}(1)$$

$$\leq \|f^* - f_r\|_{L^2}^2 + 4\left(1 + \frac{t^2}{(1-t)^2}\right) \cdot \|f^* - f_r\|_{L^2}^2 + 4\sqrt{1 + \frac{t^2}{(1-t)^2}} \cdot \|f^* - f_r\|_{L^2}^2 + o_{d,\mathbb{P}}(1).$$

Finally, due to the upper-bound (D.28), we conclude that

$$\mathcal{R}_1(\lambda) \leq 10\tau^* + C'\left(\sqrt{\tau^*} \cdot \sqrt{\tfrac{d}{n}} + \tfrac{d}{n}\right),$$

with probability one as $n, d, N \to \infty$ proportionally and $n/d > \psi_1^*$, where $\tau^*$ is defined in (5.1).

$\square$