# OpenReview forum: "High-dimensional Asymptotics of Feature Learning: How One Gradient Step Improves the Representation"
_NeurIPS.cc/2022/Conference — NeurIPS 2022 Accept_

### Official Review · Reviewer_SfUC · 2022-07-11

**Rating:** 7
**Confidence:** 2
**Soundness:** 3 good
**Presentation:** 3 good
**Contribution:** 3 good

**Summary:**

This paper characterizes the asymptotic risk of a single hidden layer neural network after one gradient descent step. The asymptotic theoretical analysis reveals two regimes of operation depending on the scaling of the learning rate: a regime where the scaling is of order 1 where the learning is better than random weights (random features) but worse than the best linear classifier. In the second regime where the rate is $\mathcal{0}(\sqrt{N})$ with $N$ the number of neurons, the learning not only outperforms a random feature but also any linear classifier. Theoretical analysis is performed using tools from random matrix theory and opens the perspective of a better understanding of the feature learning process in a neural network at least after one learning stage. Some insights from the theoretical formulas are also derived to explain the interplay between the number of samples and the number of neurons in the hidden layer.

**Questions:**

1- In Equation (2.1) line 119, the gradient should be taken in the initial weights $W_0$ and not $W_1$. In other words, $G_0$ should be function of $W_0$ and not $W_1$?

2- In the line 123 about the discussion of the ridge regression for the 2nd layer, the authors might have discussed based on one experiment how the assumption of using fresh dataset for learning $a$ affects the insights drawn in the paper. The explanation of the theoretical motivation of such assumption is great but it needs a better study of the impact of such assumption.

3- Line 226 whence --> when ?

4 - As mentioned earlier, some experiments on real data would have been welcome (at least not for comparing the close fit between theory and practice which will likely be not good but at least the intuitions drawn).

**Limitations:**

1- The authors have discussed quite thoroughly the limitations of their works. But I think applications to real data might give more strengths to the theoretical analysis. Also some experiments about assessment of the strong assumption of using different dataset for learning $W$ and $a$ would have been a great addition to the experimental part.

2- The setup of the experiments should also be thoroughly discussed in the main paper for the sake of reproducibility and assessment of the pertinence of the model used.

**Strengths And Weaknesses:**

*Strengths*

1- Theoretical understanding of the learning features process of neural network is a very important and challenging issue in machine learning and is one of the motivations of this work.

2- The paper is very well written and the authors have done a substantial job in making it pedagogical despite the technical difficulties inherent to the tackled problem and the heaviness of the theoretical analysis.

3 - Although the model studied is simple, the insights derived are non-trivial and the technical approach is original.

*Weaknesses*

1- In an empirical or intuitive approach to explaining neural networks, one would be tempted to think that the strong interdependence of the joint optimization of the hidden layer weights $W$ and the regression vector $a$ is the key factor that explains the high performance of neural networks. It is tempting to say that this work, by breaking this dependency (by taking another independent set of data for the second learning phase) misses a fundamental understanding of the learning process. It would have been very interesting to compare, at least empirically, the strong intuitions that are lost when this kind of hypothesis is made, for example by adding a global learning (using the same dataset) after a gradient descent step in an experiment that would not break this dependence, in order to at least quantify empirically the impact of this strong hypothesis.

2- Although the experiments conducted on synthetic data are very well done, it would have been interesting to run simulations on some real data sets to test the robustness of the conclusions in a real application context.

---

> ### Author Response · Authors · 2022-08-02
> **Reply to reviewer SfUC**
>
> We thank the reviewer for the positive evaluation and thoughtful feedback. We address the technical comments below.
>
> 1. **$\mathbf{G}_0$ should be function of  $\mathbf{W}_0$:**
> Thank you for pointing this out. We have now fixed this typo.
>
> 2. **Impact of data-splitting (using fresh data):**
> We agree that the data-splitting procedure is the main limitation of our current analysis, and, as commented on line 366 in the Conclusion section, handling the dependence between $\mathbf{W}_1$ and $\tilde{\mathbf{X}}$ (used in the computation of kernel ridge estimator) is an important future direction.
> While techniques based on empirical processes may handle such dependency and provide an upper bound on the prediction risk, deriving the *precise asymptotics* is more challenging and we are not aware of existing works that can deal with this setting.
> Nevertheless, we make the following remarks.
>
>     - Following the reviewer's suggestion, we have now included a new subsection in Appendix A.1 that discusses the impact of data-splitting. To summarize, from Lemma 11 and Lemma 23 in the Appendix, one can deduce that when $n/d=\psi_1\to\infty$ (this regime is commented on line 295 in Section 4.2), the difference between using the same $\mathbf{X}$ or independent $\tilde{\mathbf{X}}$ is negligible, i.e., $\tilde{\mathcal{R}}(\lambda) = \mathcal{R}(\lambda) + o_{\psi_1,\mathbb{P}}(1)$, where $\tilde{\mathcal{R}}(\lambda)$ denotes the prediction risk without data-splitting. Intuitively, this is because as the empirical gradient concentrates at the population counterpart, the dependence incurred by using the same $\mathbf{X}$ becomes insignificant.
>
>       Consequently, as we gradually increase the sample size $n$ (or ratio $\psi_1$), we expect the behavior of training $\mathbf{W}$ and $\mathbf{a}$ on the same data to become more aligned with our theoretical results (Theorems 5 and 7). Following the reviewer's suggestion, we also addeed empirical validation in Figure 5 of Appendix A, where we observe that our asymptotic formula (Theorem 5) accurately tracks the prediction risk *without data-splitting* in the regime where $\psi_1=n/d$ is not too small. Moreover, the risk curves are qualitatively similar for both the small and large learning rate cases, which suggests that the data-splitting procedure does not fundamentally alter the model behavior in our problem setting.
>
>     - Other than the connection to pre-training / transfer learning, our two-stage training procedure with independent data can also be found in other theoretical works on learning with neural networks. For example, the concurrent work [Abbe et al. 2022] considered optimizing the two layers sequentially in an online setting, which corresponds to learning $\mathbf{W}$ and $\mathbf{a}$ on separate data. [Daniely and Malach 2020] also considered learning a two-layer network by first taking one large gradient step on the first-layer, but the analysis assumes population gradient (analogous to the $n\gg d$ setting).
>
> 3. **Real-world simulations:**
> Thank you for the valuable suggestion. Due to the theoretical nature of our work, we did not consider experiments on real-world data.
> Moreover, as remarked on line 240 of Section 4.1, empirical study on the Gaussian equivalence property has been already performed in [Loureiro et al. 2021], where the authors assumed a Gaussian equivalence theorem and numerically computed the risk curves. It is empirically observed that the Gaussian equivalence predictions can be fairly accurate on some pre-trained feature maps on real datasets (e.g., MNIST, GAN-generated CIFAR10), but also fail in other settings.
> Our rigorous results provide an explanation to this discrepancy in a simple model, in which the validity of Gaussian equivalence depends on the learning rate in the feature learning procedure. In addition, we also speculate that our theoretical analysis can be extended to more generic data distribution satisfying certain near-orthogonality conditions (see the comment on line 153 in Section 2.2).
>
> We would be happy to clarify any concerns or answer any questions that may come up during the discussion period.
> ___
>
> [Abbe et al. 2022] The merged-staircase property: a necessary and nearly sufficient condition for SGD learning of sparse functions on two-layer neural networks.
>
> [Daniely and Malach 2020] Learning parities with neural networks.
>
> [Loureiro et al. 2021] Learning curves of generic features maps for realistic datasets with a teacher-student model.

---

### Official Review · Reviewer_268n · 2022-07-11

**Rating:** 7
**Confidence:** 3
**Soundness:** 3 good
**Presentation:** 4 excellent
**Contribution:** 3 good

**Summary:**

The authors analyze the behavior of a two-layer network in the proportional asymptotic regime. The authors consider a training procedure whereby a single gradient step is taken for the weights of the initial layer, which then fixes the features for a kernel ridge regressor. The authors analyze two distinct regimes: 1) a small step size regime (step size $O(1)$), and 2) a “large” step size regime (step size $O(\sqrt{N})$). In the small step size regime, the authors leverage a perturbative argument to derive a gaussian equivalence theorem, through which they are able to give a detailed analysis of the prediction risk. In the large step size regime, the authors establish an upper bound on the prediction risk, which for some problems can be contrasted with known lower bounds for random features model to demonstrate a separation for the 1-step training procedure.

**Questions:**

1) Given that the assumption of independent training sets for $W$ and $a$ is one of the main limitations of the theoretic results, would it be possible to provide empirical or heuristic evidence as to its impact (or lack thereof)? In other words - is this limitation due to the analysis or is the behavior fundamentally different?

2) Is it possible to carry out the analysis of theorem 5 by taking different sample sizes for the estimation of $W$ and $a$? This could provide interesting insights into scenarios inspired by e.g. pre-training where the number of samples for $W$ may be much larger than that for $a$. This could also enable one, in the context of applying the current result to a bona-fide sample using data splitting, to tune the ratio to optimize the prediction risk.


**Limitations:**

The limitations of the work are well addressed by the authors. I do not believe this work has any particular negative social impact.

**Strengths And Weaknesses:**

Overall, I found the paper well-written and easy to follow. The paper presents interesting insights in bridging between recent advances in characterizing random features models and more accurate models of neural network training. The results obtained by the authors are as sharp as can be reasonably expected given the problem under consideration.

In my view, the main weakness of the theory presented in this paper is that the analysis performed by the authors requires independence between the data used for the gradient step in $W$ and the data used to estimate $a$. In particular, the obvious to obtain this empirically is to split the data, causing an effective change in the ratio $N / d$ by a factor of 2. In the proportional asymptotic, this may qualitatively change the interpretation of the obtained result.

---

> ### Author Response · Authors · 2022-08-02
> **Reply to reviewer 268n**
>
> We thank the reviewer for the positive evaluation and thoughtful feedback. We address the technical comments below.
>
> 1. **Impact of data-splitting:**
> We agree that the data-splitting procedure is the main limitation of our current analysis, and, as commented on line 366 in the Conclusion section, handling the dependence between $\mathbf{W}_1$ and $\tilde{\mathbf{X}}$ (used in the computation of kernel ridge estimator) is an important future direction.
> While techniques based on empirical processes may handle such dependency and provide an upper bound on the prediction risk, deriving the *precise asymptotics* is more challenging and we are not aware of existing works that can deal with this setting.
> Nevertheless, we make the following remarks.
>
>     - Following the reviewer's suggestion, we have now included a new subsection in Appendix A.1 that discusses the impact of data-splitting. To summarize, from Lemma 11 and Lemma 23 in the Appendix, one can deduce that when $n/d=\psi_1\to\infty$ (this regime is commented on line 295 in Section 4.2), the difference between using the same $\mathbf{X}$ or independent $\tilde{\mathbf{X}}$ is negligible, i.e., $\tilde{\mathcal{R}}(\lambda) = \mathcal{R}(\lambda) + o_{\psi_1,\mathbb{P}}(1)$, where $\tilde{\mathcal{R}}(\lambda)$ denotes the prediction risk without data-splitting. Intuitively, this is because as the empirical gradient concentrates at the population counterpart, the dependence incurred by using the same $\mathbf{X}$ becomes insignificant.
>
>       Consequently, as we gradually increase the sample size $n$ (or ratio $\psi_1$), we expect the behavior of training $\mathbf{W}$ and $\mathbf{a}$ on the same data to become more aligned with our theoretical results (Theorems 5 and 7). Following the reviewer's suggestion, we also added empirical validation in Figure 5 of Appendix A, where we observe that our asymptotic formula (Theorem 5) accurately tracks the prediction risk *without data-splitting* in the regime where $\psi_1=n/d$ is not too small. Moreover, the risk curves are qualitatively similar for both the small and large learning rate cases, which suggests that the data-splitting procedure does not fundamentally alter the model behavior in our problem setting.
>
>     - Other than the connection to pre-training / transfer learning, our two-stage training procedure with independent data can also be found in other theoretical works on learning with neural networks. For example, the concurrent work [Abbe et al. 2022] considered optimizing the two layers sequentially in an online setting, which corresponds to learning $\mathbf{W}$ and $\mathbf{a}$ on separate data. [Daniely and Malach 2020] also considered learning a two-layer network by first taking one large gradient step on the first-layer, but the analysis directly assumes population gradient (analogous to the $n\gg d$ setting).
>
>
>     - Since our goal is to precisely quantify the presence of feature learning, we evaluate the prediction risk of the CK ridge regression estimator learned from the *same training set size $n$*; in other words, we use the exact same procedure to train the 2nd layer coefficients in order to quantify how much the representation improves. Moreover, if we instead use a different training set size (e.g., $n/2$) for the second layer, then the comparison of prediction risk would be less straightforward because of the presence of confounding factors (such as the shift in the "double descent" peak due to different $\psi_1$).
>
> 2. **Different sample sizes when estimating $\mathbf{W}$ and $\mathbf{a}$:**
> We thank the reviewer for the intriguing question. Indeed, our precise characterization (Theorem 5) can be easily adapted to cover the setting where the learning of $\mathbf{W}$ and $\mathbf{a}$ are performed on training sets with different size, as long as the sample sizes are proportional to $d$; we chose the same sample size only to further simplify the asymptotic expressions.
> The optimal data-splitting ratio likely depends on the Hermite coefficient $\mu$ of the nonlinearities as well as the regularization parameter $\lambda$; we intend to investigate this problem in future work.
>
> We would be happy to clarify any concerns or answer any questions that may come up during the discussion period.
> ___
> [Abbe et al. 2022] The merged-staircase property: a necessary and nearly sufficient condition for SGD learning of sparse functions on two-layer neural networks.
>
> [Daniely and Malach 2020] Learning parities with neural networks.

---

### Official Review · Reviewer_nHmM · 2022-07-12

**Rating:** 8
**Confidence:** 4
**Soundness:** 4 excellent
**Presentation:** 4 excellent
**Contribution:** 4 excellent

**Summary:**

This paper studies the exact asymptotics of  feature learning in a two-layer neural network. The training procedure includes two steps: 1) fix the second layer and train the frist layer using gradient descent; 2) fix the first layer and optimize the second layer. The data samples used in the two steps are independent. Under the Gaussian assumption on the initialized weights and data input, a Gaussian equivalence result is established, based on which the exact formula of the generalization error is obtained. Specifically, it is rigorously shown that 1) after one small gradient step, the generalization error is smaller than the random feature model; 2) after one large step, it is possible to obtain a better estimate than the linear model. These results demonstrate the benefits of feature learning in a precise way.

**Questions:**

1. Should $\boldsymbol{W}_1$ in Eq. (2.1) be $\boldsymbol{W}_0$?

2. Does the upper bound on $\delta$ (line 292) directly follow from the form of $\delta(\cdot)$ in Theorem 5? If so, it is better to make an explicit statement.

3. Can the authors give some intuition on how the specific model in Lemma 6 is constructed and where Eq. (5.1) come from?

4. Can the also comment on the impact of the choice of $\sigma(x)$? For example, which $\sigma(x)$ can maximize $\delta(\cdot)$ in Theorem 5 and what is a good $\sigma(x)$ for large gradient step?

5. Can the authors comment on the exact asymptotics after multiple gradient steps?

**Strengths And Weaknesses:**

Strengths: The exact asymptotics of learning in the linear (or NTK) regime has been studied in several works before. However, what is much less unknown is the exact chacracterization beyond the linear regime, where feature learning is possible. This is also one of the key reasons why neural network can outperform traditional methods such as kernel method. This paper take an initial step towards this direction and the results are quite new and interesting. For example, a quantitative result (Theorem 5) is given on the improvement after one gradient step. Besides, the presentation is clear and the proof parts are well-written.

Weaknesses: The model (single-index) and training algorithm (data-spliting) considered in this paper is relatively special and it will be interesting to further extend to more general settings.

---

> ### Author Response · Authors · 2022-08-02
> **Reply to reviewer nHmM**
>
> We thank the reviewer for the positive evaluation and thoughtful feedback. We address the technical comments below.
>
>
> 1. **Should $\mathbf{W}_1$ in Eq. 2.1 be $\mathbf{W}_0$?:**
> Thank you for pointing this out. We have now fixed this typo.
>
> 2. **Upper bound on $\delta$:**
> The upper bound on $\delta$ is due to Fact 4, which lower bounds the prediction risk of the Gaussian equivalence model, which then translates to an upper bound on the possible improvement $\delta$ in this regime.
>
> 3. **Intuition on the model in Lemma 6 and Eq. (5.1):**
> A short summary on the construction of our oracle estimator can be found in the beginning of the Appendix D starting at line 1374 (line 1389 in the revised version). To recap, the student model in Lemma 6 is constructed using an average over a *subset* of neurons defined in Equations (D.5) and (D.7). When $n/d$ is large, we show that this construction essentially gives a "Gaussian-smoothed" single-index model (where the smoothing is due to the random initialization) which achieves the prediction risk $\tau^*$.
>
> 4. **Impact of the choice of $\sigma(x)$:**
> For the small learning rate regime, the choice of $\sigma(x)$ only enters the definition of $\delta(\eta,\lambda,\psi_1,\psi_2)$ through the first and second Hermite expansion coefficients $\mu_1$ and $\mu_2$; in this case, since learning is due to the *linear* component, we know that a larger $\mu_1 = \mathbb{E}[x\sigma(x)]$ implies a more significant improvement in the prediction risk.
> As for the large learning rate regime, the impact of $\sigma(x)$ is captured by $\tau^*$ defined in Equation (5.1); here whether a certain $\sigma$ achieves low prediction risk also depends on the teacher $\sigma^*$ (see Corollary 8 and Appendix D.2 for specific examples).
>
>
> 5. **Exact asymptotics after multiple gradient steps:**
> It is indeed possible to extend our precise characterization to multiple steps of feature learning, as long as the learning rate and the number of gradient steps remain finite.
> Intuitively, this is because when the learning rate is small, one may verify that the learned weight matrix after $t$ steps $\mathbf{W}_t$ also remains *close to initialization* (as defined in Equation (C.1)). Consequently, we may establish the Gaussian equivalence property and then compute the asymptotic prediction risk.
> We leave the rigorous treatment of this multi-step setting as future work.
>
>
> 6. **Extending to more general settings:**
> We agree with the reviewer that relaxing our assumptions on the teacher model and removing the data-splitting requirement are two important extensions to consider.
> We make the following remarks.
>
>     - For the data-splitting procedure, we added a new subsection in Appendix A.1 with empirical findings demonstrating that as $\psi_1=n/d$ becomes larger, our theoretical predictions (Theorem 5 and 7) can accurately capture the risk behavior in the absence of data splitting (i.e., $\mathbf{W}$ and $\mathbf{a}$ trained on the same samples). Please see our reply to reviewer 268n for more details.
>
>     - As commented on line 45 in the Introduction, we considered a single-index teacher primarily because this is a setting where the precise asymptotics of random features regression has been extensively studied; hence we believe that this is a good starting point to evaluate the benefit of feature learning. On the technical side, one reason is that our Gaussian equivalence proof builds upon [Hu and Lu 2020], which only handles the single-index setting. That said, we believe that a similar precise characterization can be obtained for more general settings such as certain multiple-index models (i.e., finite-width two-layer neural network).
>
> We would be happy to clarify any concerns or answer any questions that may come up during the discussion period.
>
> ___
> [Hu and Lu 2020] Universality laws for high-dimensional learning with random features.

---

### Author Response · Authors · 2022-08-02
**Summary of revision**

Dear Reviewers and Area Chair,

We deeply appreciate your continuing time and effort to provide detailed comments on our paper. To best respond to your comments, we revised our paper with additional clarifying content as suggested by the reviewers. Below is a short summary of the updates.

- We included a new subsection in Appendix A.1 to discuss the impact of data-splitting, i.e., $\mathbf{W}$ and $\mathbf{a}$ trained on different data. In short, we empirically observe qualitatively similar behavior with or without the data-splitting procedure; moreover, our asymptotic predictions (Theorem 5) can still accurately track the risk curve when $\psi_1=n/d$ is not too small.
- We improved the writing and fixed typos pointed out by the reviewers.

Please refer to our detailed response to each reviewer where we address all individual comments.

---

### Meta-Review · Area_Chair_oEhJ · 2022-08-31

**Recommendation:** Accept
**Confidence:** Certain

**Metareview:**

The paper studies a two layer neural network in the setting of one step of gradient descent on the first layer (after random init) and then freezing the first layer and training the last layer. The study is in proportinal asymptotic limits of parameters going to infinity. They identify two regimes depending on the learning rate for the single step on the first layer, in the first regime the step can improve over the random features model but stays below the best linear predictor and in the higher learning rate regime it can improve over the best linear predictor. The latter is established for a class of f* (student teacher model assumed). The meta takeaway is a neat study how feature learning can be happening. Although the precise setting is somewhat restricted, the results are strong and can lead to further research in this important area.

Overall all the reviewers felt that the paper is a strong contribution to Neurips community. I am happy to recommend acceptance.

**Award:**

No

---

### Decision · Program_Chairs · 2022-09-14

Accept